# 101 Geodynamic modelling: How to design, interpret, and communicate numerical studies of the solid Earth

Iris van Zelst[1,2,*], Fabio Crameri[3,4,*], Adina E. Pusok[5,*], Anne Glerum[6,*], Juliane Dannberg[7,*], and Cedric Thieulot[8,*]

[1]School of Earth and Environment, University of Leeds, Leeds, LS2 9JT, United Kingdom
[2]Now at: Institute of Planetary Research, German Aerospace Center (DLR), Berlin, Germany
[3]Undertone Design, Bern, Switzerland
[4]Centre for Earth Evolution and Dynamics (CEED), University of Oslo, Postbox 1028 Blindern, 0315 Oslo, Norway
[5]Department of Earth Sciences, University of Oxford, United Kingdom
[6]Helmholtz Centre Potsdam, GFZ German Research Centre for Geosciences, Potsdam, Germany
[7]Department of Geological Sciences, University of Florida, USA
[8]Department of Earth Sciences, Utrecht University, Utrecht, The Netherlands
[*]These authors contributed equally to this work.

**Correspondence:** Iris van Zelst (iris.vanzelst@dlr.de / iris.v.zelst@gmail.com)

**Abstract.** Geodynamic modelling provides a powerful tool to investigate processes in the Earth's crust, mantle, and core that are not directly observable. However, numerical models are inherently subject to the assumptions and simplifications on which they are based. In order to use and review numerical modelling studies appropriately, one needs to be aware of the limitations of geodynamic modelling as well as its advantages. Here, we present a comprehensive, yet concise overview of the geodynamic modelling process applied to the solid Earth, from the choice of governing equations to numerical methods, model setup, model interpretation, and the eventual communication of the model results. We highlight best practices and discuss their implementations including code verification, model validation, internal consistency checks, and software and data management. Thus, with this perspective, we encourage high-quality modelling studies, fair external interpretation, and sensible use of published work. We provide ample examples, from lithosphere and mantle dynamics specifically, and point out synergies with related fields such as seismology, tectonophysics, geology, mineral physics, planetary science and geodesy. We clarify and consolidate terminology across geodynamics and numerical modelling to set a standard for clear communication of modelling studies. All in all, this paper presents the basics of geodynamic modelling for first-time and experienced modellers, collaborators, and reviewers from diverse backgrounds to (re)gain a solid understanding of geodynamic modelling as a whole.

## 1 Introduction

The term **geodynamics** [1] combines the Greek word 'geo', meaning 'Earth', and the term 'dynamics' – a discipline of physics that concerns itself with forces acting on a body and the subsequent motions they produce (Forte, 2011). Hence, geodynamics is the study of forces acting on the Earth, and the subsequent motion and deformation occurring in the Earth. Studying geo-

---

[1]See the glossary in the Supplementary Material for definitions of the bold words that occur throughout the text

dynamics contributes to our understanding of the evolution and current state of the Earth and other planets, and the natural hazards and resources on Earth.

The broad definition of geodynamics typically results in a subdivision of disciplines relating to one of the Earth's spherical shells and specific spatial and temporal scales (Figure 1). So, geodynamics considers brittle and ductile deformation and plate tectonic processes on the crustal- and lithospheric scale as well as convection processes in the Earth's mantle and core. Lithospheric and crustal deformation operates on scales of tens to thousands of kilometres and thousands to millions of years to study both the folding of lithospheric plates and the faulting patterns accommodating them (Watts et al., 2013). **Mantle**

**convection** encompasses individual flow features like mantle plumes and subducted slabs on scales of hundreds of kilometres and hundred-thousands of years as well as whole mantle overturns on thousand kilometre and billion year scales (Androvandi et al., 2011; King, 2015). In the outer core, there are both small convection patterns on hundred-metre and hour scales and large-scale convection processes taking place over tens of kilometres and ten thousands of years (Bouffard et al., 2019). **Inner core dynamics** is believed to act on similar scales as mantle convection, although it is has not yet been established whether

or not its **Rayleigh number** allows for convection (Deguen and Cardin, 2011). In addition to the large-scale dynamics of the different spherical shells of the Earth, there are other processes that play a role and operate on yet other different spatial and temporal scales. For example, earthquakes and - on larger time scales earthquake cycles - facilitate the large-scale deformation of the lithosphere, but their nucleation process can occur on scales as little as milliseconds and millimetres (Stein and Wysession, 2009; Beroza and Kanamori, 2015). Similarly, surface processes such as erosion can play an important role in

lithosphere dynamics, although they take place on a smaller spatial and temporal scale (Pelletier, 2008). For lithosphere and mantle dynamics, processes including magma, fluid, volatile and grain dynamics have been shown to be important, which all have smaller spatial and temporal scales (Solomatov and Reese, 2008; Keller et al., 2013; McKenzie, 1984; Rüpke et al., 2004; Bercovici and Ricard, 2012; Keller and Katz, 2016; Katz, 2008; Montési and Behn, 2007). Hence, in order to fully understand geodynamics, many processes on different temporal and spatial scales need to be taken into account simultaneously. However,

this is difficult to achieve using only observational or experimental studies.

Indeed, since the study of geodynamics is predominantly occupied with processes occurring below the surface of the Earth, one of the challenges is the limited amount of direct observations in space and time. Engineering limitations are responsible for a lack of direct observations of processes at depth, with, for example, the deepest borehole on Earth being a mere 12 km (e.g., Ganchin et al., 1998), less than 0.2% of the radius of the Earth (6371 km). In addition, the available rock record grows

increasingly scarce when going back in time. Other disciplines, such as seismology, geology, and geodesy, relate to geodynamics as they study the (indirectly) observable expressions of geodynamic processes at the surface and at depth. Disciplines like mineral physics and rock mechanics relate to geodynamics by studying the physical properties and flow laws of rocks involved in geodynamic processes.

To compensate for the limited amount of data in geodynamics, many studies have turned towards modelling. Roughly

speaking, there are two main branches of modelling based on the tools that they use: **analogue modelling**, where real-world materials are used in a lab, and **physical or mathematical modelling**, where physical principles and computational methods are used. Analogue models have many advantages of which the most straightforward one is that they are automatically subject to all

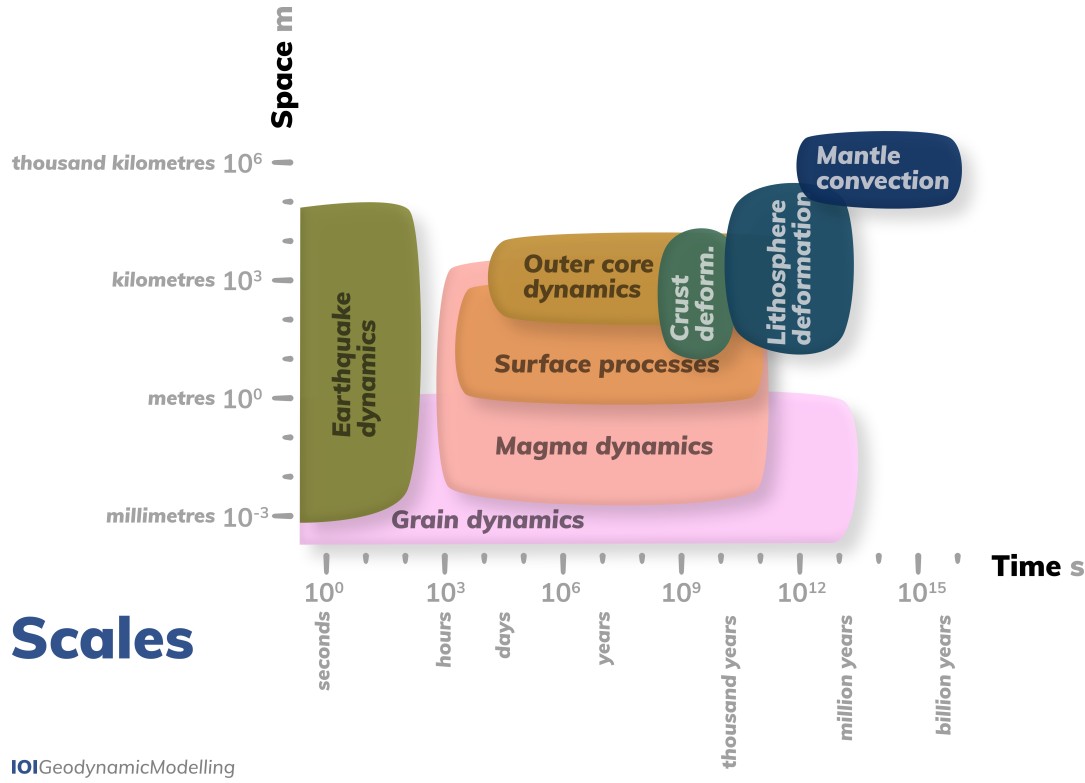

**Figure 1.** Spatial and temporal scales of common geodynamic processes. These processes occur over a wide range of time and length scales, and modellers have to take into account which of them can be included in any given model.

the physical laws on Earth. However, it is non-trivial to upscale the models to real-world applications that occur on a different temporal and spatial scale. Similarly, important aspects such as gravity are difficult to scale in a lab setting (e.g., Ramberg, 55   1967; Mart et al., 2005; Noble and Dixon, 2011). In addition, it is hard to determine variables such as stress and strain across the entire domain at any given time. Physical models aim to describe processes through equations. For relatively simple processes, analytical solutions or semi-analytical models can be used (e.g., Lamb, 1879; Samuel and Bercovici, 2006; Montési and Behn, 2007; Turcotte and Schubert, 2014; Ribe, 2018; McKenzie, 1969; Hager and O'Connell, 1981). However, when the equations and their dependencies become increasingly complex in their description of geodynamic processes, numerical methods become 60   necessary to solve the set of equations. Hence, **numerical modelling** is the use of numerical methods to solve a set of physical equations describing a physical process. Numerical models are powerful tools to study any desired geometry and time evolution with full control over the physics and access to all variables at any point in time and space in the model, but they come with their own set of limitations and caveats. For example, despite recent developments in numerical methods and modelling studies, it remains difficult to combine diverse spatial and temporal scales in one numerical model. This is due to the fact that numerical 65   models are increasingly more difficult to solve computationally when large contrasts are present in the physical properties (e.g., in space or time). Therefore, different modelling approaches have been developed to tackle the different natures of the

processes outlined in Figure 1. Ideally, these modelling approaches would be combined in the future. In this article, we focus on the continuously-growing branch of thermo-mechanical numerical modelling studies in geodynamics which mainly deal with lithosphere deformation and mantle convection, but we point out the links with other processes wherever relevant. Other disciplines, such as atmospheric dynamics and hydrology, use similar equations and assumptions, but are not discussed in detail here.

Since geodynamics has much in common with other Earth science disciplines, there is a frequent exchange of knowledge where, e.g., geodynamic studies use data from other disciplines to constrain their models. Vice versa, geodynamic models can inform other disciplines on the theoretically possible motions in the Earth. Therefore, scientists from widely diverging backgrounds are starting to incorporate geodynamic models into their own projects, applying the interpretation of geodynamic models to their own hypotheses, or are asked to review geodynamic articles. In order to correctly use, interpret, and review geodynamic modelling studies, it is important to be aware of the numerous assumptions that go into these models and how they affect the modelling results. Similarly, knowing the strengths and weaknesses of numerical modelling studies can help to narrow down whether or not a geodynamic model is the best way of answering a particular research question.

Here, we provide an overview of the whole geodynamic modelling process (Figure 2). We discuss how to develop the numerical code; set up the model from a geological problem; test and interpret the model; and how to communicate the results to the scientific community. We highlight the common and best practices in numerical thermo-mechanical modelling studies of the solid Earth and the assumptions that enter such models. We will focus on the Earth's crust and mantle, although the methods outlined here are similarly valid for other solid planetary bodies such as terrestrial planets and icy satellites, and the Earth's solid inner core. More specifically, we concentrate on and provide examples of lithosphere and mantle dynamics. We focus on recent advances in good modelling practices and clearly, unambiguously communicated, open science that make thermo-mechanical modelling studies reproducible and available for everyone.

## 1.1   What is a model?

We will use the word **model** to mean a simplified version of reality that can be used to isolate and investigate certain aspects of the real world. Therefore, by definition, a model never equals the complexity of the real world and is never a complete representation of the world, i.e., all models are inherently wrong, but some are useful (Box, 1976). We define a **physical model** as the set of equations that describes a physical or chemical process occurring in the Earth.

Computational geodynamics concerns itself with using numerical methods to solve a physical model. It is a relatively new discipline that took off with the discovery of plate tectonics and the rise of computers in the 1960s (Schubert et al., 2001; Gerya, 2019). Within computational geodynamics, a **numerical model** refers to a specific set of physical equations which are discretised and solved numerically with initial and boundary conditions. Here, we limit ourselves to thermo-mechanical numerical models of the Earth's crust, lithosphere, and mantle, where 'thermo-mechanical' refers to the specific set of equations the numerical code solves (Section 2).

It is important to realise that a numerical model is not equivalent to the **numerical code**. A numerical code can contain and solve one or more numerical models (i.e., different sets of discretised equations). Each numerical model can have a

different **model setup** with different dimensions, geometries, and initial and boundary conditions. For a specific model setup, the numerical model can then be executed multiple times by varying different aspects of the model setup to constitute multiple **model simulations** (also often called model runs).

While it can be tested if a numerical model solves the equations correctly through **analytical solutions**, it cannot be proven that the numerical model solves the appropriate physical model, i.e., equations. In addition, the results of simulations with complexity beyond analytical solutions can be compared against observations and results from other codes, but again cannot be proven to be correct or, indeed, the only possible model producing such results.

The models we are concerned with here are all forward models, where we obtain a model result by solving equations that describe the physics of a system. These results are a prediction of how the system behaves given its physical state which afterwards can be compared to observations. On the other hand, inverse models start from an existing data set of observations and aim to determine the conditions responsible for producing the observed data set. A well-known example of these kinds of models are the tomographic models of the Earth's mantle, which determine the 3-D seismic velocity structure of the mantle based on seismic data sets consisting of e.g., P-wave arrival times or full waveforms (e.g., Dziewonski, 1984; Bijwaard and Spakman, 2000; Fichtner et al., 2013). The possibility of incorporating data and estimating the model uncertainties in inverse models has recently led to an increasing amount of efforts combining both inverse and forward methods in geodynamic modelling. For example, adjoint methods (Liu and Gurnis, 2008; Burstedde et al., 2009; Reuber et al., 2018c, 2020a), pattern recognition (Atkins et al., 2016), and data assimilation (Bunge et al., 2002, 2003; Ismail-Zadeh et al., 2004; Majumder et al., 2005; Bocher et al., 2016, 2018) have been incorporated in lithosphere and mantle dynamics models. One of the most common ways of solving the inverse problem in geodynamics is to run many forward models and subsequently test which one fits the observations best (Baumann and Kaus, 2015). This is difficult, because individual forward models are computationally expensive which results in a limited amount of forward models that can be run. Additionally, there are too many parameters in the forward models to invert for. An alternative approach is to incorporate automatic parameter scaling routines or use adjoint methods to test parameter sensitivities in models (Reuber et al., 2018a, c), which could considerably reduce the amount of models required. However, whether it is forward or inverse models of geodynamics, a thorough understanding of the forward model is generally required, which we focus on in this work. The prospect of estimating uncertainties in geodynamic modelling and the increasing inclusion of data make the combination of inverse and forward modelling an exciting avenue in geodynamics.

## 1.2 What is modelling?

A scientific modelling study encompasses more than simply running a model, as is illustrated in Figure 2. First and foremost, the modeller needs to decide which natural process they want to study based on observations to increase the understanding of the Earth's system. From this, a hypothesis (i.e., a proposed explanation based on limited evidence) is formulated as a starting point to further investigate a knowledge gap driven by the modeller's curiosity. The modeller then needs to choose the equations that describe the physics of the processes they are interested in, i.e., the physical model (Section 2). Here, we confine ourselves to the conservation equations of mass, momentum and energy (Section 2.1). In order to solve the physical model, a numerical model is needed, which solves the discretised versions of the equations using a specific numerical scheme

(Section 3). Before applying the code to a particular problem, the code needs to be verified to ensure that it solves the equations correctly (Section 4). Once it has been established that the code works as intended, the to-be-tackled geological problem needs to be simplified into a model setup with appropriate geometry, material properties, initial and boundary conditions and the correct modelling philosophy (Section 5). After the model has successfully run, the model results should be validated in terms of robustness and against observations (Section 6). The results of the simulation then need to be interpreted, visualised, and diagnosed while taking into account the assumptions and limitations that entered the modelling study in previous steps (Section 7). The final step of a modelling study is to communicate the results to peers as openly, clearly, and reproducibly as possible (Sections 8, 9). It is important to note that geodynamic modelling is not a linear process as depicted in Figure 2. For example, modelling does not necessarily start with the formulation of a hypothesis, as this can also arise from the encounter of interesting, unexpected modelling results, resulting in the formulation of a hypothesis later on in the process. Similarly, observations from nature feed directly into all of the above-mentioned steps as illustrated by the dark grey arrow through all modelling steps (Figure 2). Indeed, it is important to evaluate and adjust the numerical modelling study throughout the entire process.

In the remaining parts of this paper, each of the above-mentioned steps of a modelling study correspond to individual sections, making this a comprehensive guide to geodynamic modelling studies.

## 2  The physical model

From seismology, we know that on short time scales the Earth predominantly deforms like an elastic medium. Our own experience tells us that when strong forces are applied to rocks, they will break (brittle failure). But from the observation that continents have moved over the history of the Earth, and from the postglacial rebound of regions like Scandinavia, we know that on long time scales the mantle strongly deforms internally. In geodynamic models, this ductile deformation of rocks is usually approximated as the flow of a highly **viscous fluid** (Schubert et al., 2001; Gerya, 2019). The transition from short to long time scales that controls which deformation behaviour is dominant is marked by the **Maxwell relaxation time** (Ricard, 2015), which is on the order of 450 years for the Earth's sublithospheric mantle (Schubert et al., 2001). The rocks in the Earth's interior are not the only material that deforms predominantly in this way. In fact, many other materials show the same behaviour: solid ice, for example, also flows, causing the constant motion of glaciers (Ricard, 2015).

In the following, we will focus on problems that occur on large time scales on the order of thousands or millions of years (i.e., $\gg 450$ years, the Maxwell time, see Figure 1). Accordingly, we will treat the Earth's interior as a highly viscous fluid. We further discuss this assumption and how well it approximates rocks in the Earth in Section 2.2.1. We will also treat Earth materials as a continuum, i.e., we assume that the material is continuously distributed and completely fills the space it occupies, and that material properties are averaged over any unit volume. Thus, we ignore that the material is made up of individual atoms (Helena, 2017). This implies that, on a macroscopic scale, there are no mass-free voids or gaps (Gerya, 2019). Under these assumptions and keeping the large uncertainties of Earth's materials in mind, we can apply the concepts of fluid dynamics to model the Earth's interior.

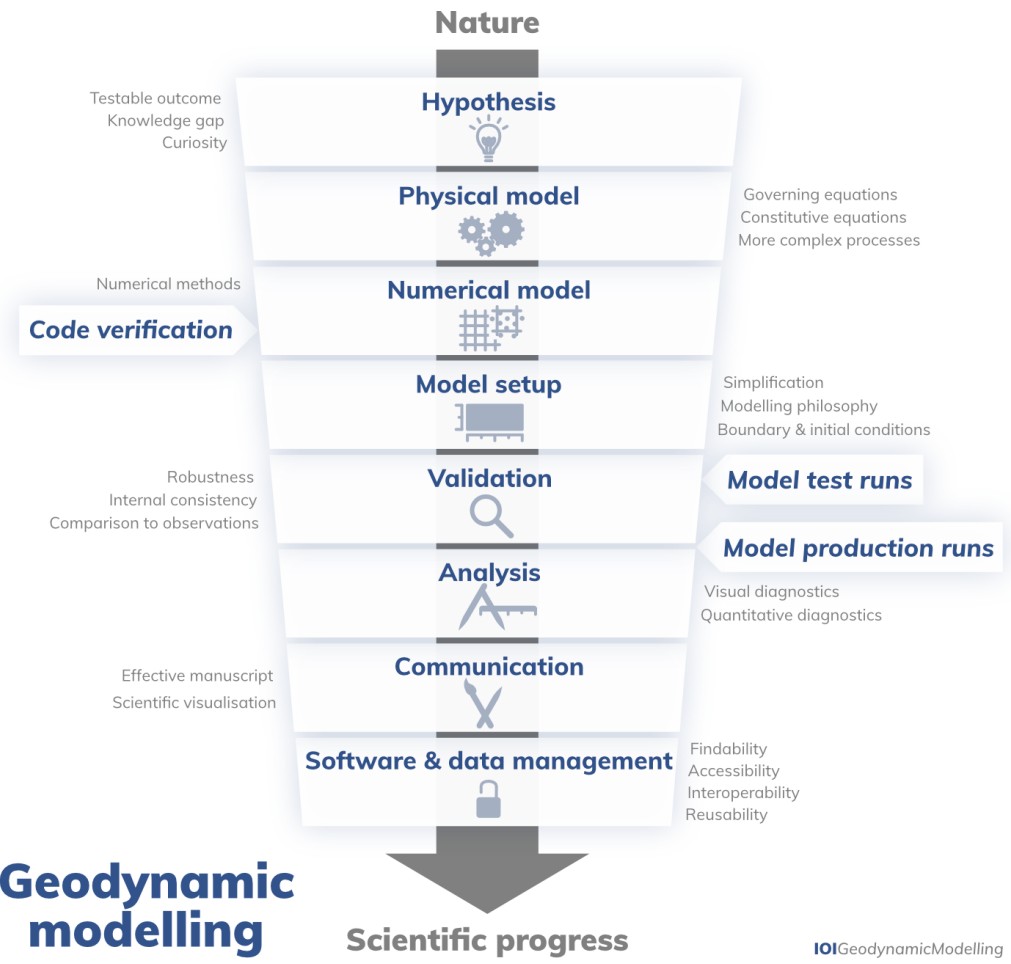

**Figure 2.** The geodynamic modelling procedure. A modelling study encompasses everything from the assemblage of both a physical (Section 2) and a numerical model (Section 3) based on a verified numerical code (Section 4), to the design of a simplified model setup based on a certain modelling philosophy (Section 5), the validation of the model through careful testing (Section 6), the unbiased analysis of the produced model output (Section 7), the oral, written, and graphical communication of the modelling approach and results (Section 8), and the management of both software and data (Section 9). Note that constant (re-)evaluation and potential subsequent adjustments of previous steps are key, and indeed necessary, throughout this process.

## 2.1 Basic equations

We have discussed above that setting up a model includes choosing the equations that describe the physical process one wants to study. In a fluid dynamics model, these governing equations usually consist of a mass balance equation, a force balance or momentum conservation equation, and an energy conservation equation. The solution to these equations states how the values of the unknown variables such as the material velocity, pressure, and temperature (i.e. the dependent variables) change in space, and how they evolve over time (i.e. when one or more of the known/independent variables change). Even though these governing equations are conservation equations, geodynamic models often consider them in a local rather than a global context, i.e., material and energy flow in or out of a system, or an external force is applied. Additionally, the equations only consider specific types of energy, i.e., thermal energy, and not, for example the potential energy related to nuclear forces. This means that for any system considered — or, in other words, within a given volume — a property can change due to the transport of that property into or out of the system, and thermal energy may be generated or consumed through a conversion into other types of energy, e.g., through radioactive decay. This can be expressed using the basic principle:

$$\text{Accumulation} = \text{Influx} - \text{Outflux} + \text{Generation} - \text{Consumption}. \tag{1}$$

This statement suggests that accumulation, or rate of change of a quantity within a system, is balanced by net flux through the system boundaries (influx/outflux), and net production within the system (generation/consumption). Depending on which physical processes are relevant and/or of interest, different terms may be included in the equations, which is a source of differences between geodynamic studies. Since each term describes a different phenomenon and may imply characteristic time and length scales, being aware of these relations is not only important for setting up the model, but also for interpreting the model results and comparing them to observations. Mathematical tools like **scaling analysis** or **linear stability analysis** are helpful to determine the relative importance of each of the terms (see also Section 7.2), and for determining which terms should be included in any given model. In solid Earth geodynamic modelling, we are usually interested in problems occurring on large time scales on the order of thousands or millions of years (Figure 1). Accordingly, we usually model the Earth's interior as a **viscous fluid**. The corresponding equations are the **Stokes equations**, describing the motion of a highly viscous fluid driven in a gravitational field, and the energy balance written as a function of temperature (Figure 3). These equations are partial differential equations, describing the relation between the unknowns (velocity, pressure, and temperature) and their partial derivatives with respect to space and time. Solving them requires additional relations, the so-called constitutive equations (Section 2.2), that describe the material properties appearing as parameters in the Stokes equations and energy balance (such as the density) and their dependence on velocity, temperature and pressure. We look at each equation in more detail in the following sections, but we do not provide their derivations, and instead refer the interested reader to Schubert et al. (2001).

### 2.1.1 Mass conservation

The first equation describes the conservation of mass:

$$\frac{\partial \rho}{\partial t} + \boldsymbol{\nabla} \cdot (\rho \boldsymbol{v}) = 0, \tag{2}$$

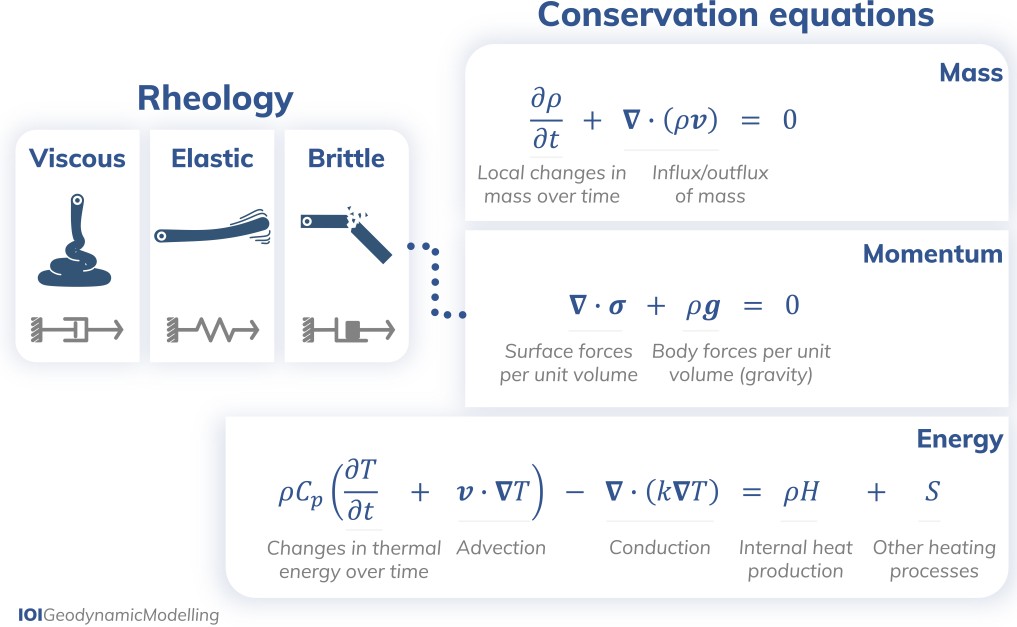

**Figure 3.** The governing equations: conservation of mass (Eq. 2, Section 2.1.1), conservation of momentum (Eq. 5, Section 2.1.2), and conservation of energy (Eq. 6, Section 2.1.3) with different types of rheology (Section 2.2.1). In these equations, $\rho$ is the density, $t$ is time, $\boldsymbol{v}$ is the velocity vector, $\boldsymbol{\sigma}$ is the **stress tensor**, $\boldsymbol{g}$ is the gravitational acceleration vector, $C_p$ is the heat capacity, $T$ is the temperature, $k$ is the thermal conductivity, $H$ is a volumetric heat production term (e.g., due to radioactive decay) and the term $S = S_1 + S_2 + S_3$ accounts for viscous dissipation, adiabatic heating, and the release or consumption of latent heat (e.g., associated with phase changes), respectively. Many geodynamic models include additional constitutive and evolution equations. Note that the brittle rheology depicted here is approached by plasticity in geodynamic models, as depicted by the plastic slider in dark grey underneath the breaking bar. See Section 2.2.4 for more details.

where $\rho$ is the mass density, $t$ is time, and $\boldsymbol{v}$ is the velocity vector. The conservation of mass (2) states that in any given volume of material, local changes in mass $\partial\rho/\partial t$ can only occur when they are compensated by an equal net influx or outflux of mass $\boldsymbol{\nabla} \cdot (\rho\boldsymbol{v})$. These local changes in mass are caused by the expansion or contraction of material, which implies a change in density $\rho$. This usually happens as a response to a change in external conditions, such as a change in temperature (thermal expansion/contraction) or pressure, and is described by the equation of state (see Section 2.2.5). Other specific examples in the Earth are phase transitions or chemical reactions.

Because the first term explicitly includes a time-dependence, it introduces a characteristic time scale into the model due to viscous (Curbelo et al., 2019) and elastic forces (Patočka et al., 2019). For viscous forces, this is called the **viscous isentropic relaxation time scale** and is on the order of a few hundred years for the upper mantle to a few tens of thousands of years for the lower mantle (Curbelo et al., 2019). For visco-elastic deformation, this time scale is the Maxwell time (see above). When we consider the Earth as a visco-elastic body (see Section 2.2.1), the relaxation time is dominated by elastic forces.

The time scale of viscous relaxation is usually shorter than that of convective processes, and is often shorter than the time scales a model is focused on. In addition, these local changes in mass are often quite small compared to the overall mass flux in the Earth. Accordingly, many geodynamic models do not include this term (see also Jarvis and McKenzie, 1980; Glatzmaier, 1988; Bercovici et al., 1992), and instead Eq. (2) is replaced by

$$\boldsymbol{\nabla} \cdot (\rho \boldsymbol{v}) \;=\; 0. \tag{3}$$

This means that the net influx and outflux of mass in any given volume of material is zero. The density can still change, e.g., if material is advected into a region of higher or lower pressure (i.e., downwards or upwards within the Earth), but these changes in density are always associated with the motion of material to a new location. Using this approximation still takes into account the largest density changes. For example, for the Earth's mantle, density increases by approximately 65% from the Earth's surface to the core-mantle boundary.

In some geodynamic models, particularly the ones that span only a small depth range, even this kind of density change is small. For example, within the Earth's upper mantle, the average density changes by less than 20%. This is the basis for another approximation: assuming that material is incompressible (i.e., its density $\rho$ is constant). In this case, Eq. (2) becomes

$$\boldsymbol{\nabla} \cdot \boldsymbol{v} \;=\; 0. \tag{4}$$

Assuming incompressibility also has implications for another material property in the model, the Poisson's ratio $\nu$, which is defined as $\nu = \frac{\lambda}{2(\lambda+G)}$ for a homogeneous isotropic linear elastic material, where $\lambda$ is Lamé's first parameter and $G$ is the shear modulus. The Poisson's ratio is a measure frequently used in seismology to describe the volume change of a material in response to loading, or in other words, how much a material under compression expands in the direction perpendicular to the direction of compression. The assumption of incompressibility implies a Poisson's ratio of $\nu = 0.5$. When converting material properties from geodynamic models to wave speeds for seismological applications, incompressible models result in unrealistic infinite P-wave speeds unless a different value for the Poisson's ratio is assumed during the conversion (van Zelst et al., 2019).

### 2.1.2 Momentum conservation

Eq. (5) describes the conservation of momentum, or, in the way we use it here, a force balance:

$$\boldsymbol{\nabla} \cdot \boldsymbol{\sigma} + \rho \boldsymbol{g} = \boldsymbol{0}, \tag{5}$$

where $\boldsymbol{\sigma}$ is the stress tensor, $\rho$ is the density, and $\boldsymbol{g}$ is gravitational acceleration vector. The conservation of momentum states that the sum of all forces acting on any parcel of material is zero. Specifically, the first term $\boldsymbol{\nabla} \cdot \boldsymbol{\sigma}$ represents the net surface forces and the second term the net body forces. In mantle convection and long-term tectonics models, the gravity force $\rho \boldsymbol{g}$ is generally the only body force considered, and the gravitational acceleration $\boldsymbol{g}$ is often taken as a constant of 10 m/s$^2$ for global studies, because it varies by less than 10% over the whole mantle depth (Dziewonski and Anderson, 1981), whereas for lithospheric-scale studies $\boldsymbol{g}$ is taken to be 9.8 m/s$^2$. Other body forces like electromagnetic or Coriolis forces are negligibly

small compared to gravity. Conversely, these forces are very important for modelling the magnetic field in the outer core. Hence, depending on the geodynamic problem, additional terms become relevant in the force balance.

The surface forces are expressed as the spatial derivatives of the stress $\boldsymbol{\nabla} \cdot \boldsymbol{\sigma}$. Stresses can be perpendicular to the surface of a given parcel of material like the pressure; they can act parallel to the surface; or they can point in any other arbitrary direction. In addition, the forces acting on a surface may be different for each surface of a parcel of material. Accordingly, the stress can be expressed as a $3 \times 3$ tensor in 3 dimensions (3D), where each entry corresponds to the component of the force acting in one of the three coordinate directions $(x, y, z)$ on a surface oriented in any one of the three coordinate directions, giving a total of $3 \times 3 = 9$ entries. One of the most complex choices in the design of a geodynamic model is the relation between these stresses and the deformation (rate) of rocks, i.e., the rheology (Section 2.2.1).

Under the assumption that deformation is dominantly viscous, Eqs. (2) and (5) are often referred to as the **Stokes equations**. In their more general form, called the **Navier–Stokes equations**, Eq. (5) contains an additional term $\rho\, D\boldsymbol{v}/Dt$ that governs inertia effects, describing the acceleration of the material. In the Earth's mantle, material moves so slowly that inertia (and, consequently, the momentum of the material) does not have a significant influence on the motion of the material. Consequently, this term is very small and usually neglected in geodynamic models. In other words, we look at the behaviour of the material on such a long time scale that from our point of view, material accelerates/decelerates almost immediately to its steady-state velocity. Conversely, for models of seismic wave propagation (which cover a much shorter time scale, and consequently use a different form of the momentum equation because deformation is predominantly elastic; Stein and Wysession (2009)), it is crucial to include the inertia term because it introduces the physical behaviour that allows for the formation of seismic waves.

Dropping the inertia term means that the **Stokes equations** (2) and (5) describe an instantaneous problem if the mass conservation is solved using one of the common approximations (3) or (4). The time does not appear explicitly in these equations, so the solution for velocity and pressure at any point in time is independent of the history unless this is explicitly incorporated in the material properties (Section 2.2). The model evolution in time is incorporated through Eq. (6), which describes the conservation of energy. For more details on the Stokes equations and a complete derivation, we refer the reader to Schubert et al. (2001).

### 2.1.3 Energy conservation

Eq. (6) describes the conservation of thermal energy, expressed as $\rho C_p T$:

$$\rho C_p \left( \frac{\partial T}{\partial t} + \boldsymbol{v} \cdot \boldsymbol{\nabla} T \right) - \boldsymbol{\nabla} \cdot (k \boldsymbol{\nabla} T) = \rho H + S, \tag{6}$$

where $\rho$ is density, $C_p$ is the heat capacity, $T$ is the temperature, $t$ is time, $\boldsymbol{v}$ is the velocity vector, $k$ is the thermal conductivity, $H$ is a volumetric heat production term, and $S = S_1 + S_2 + S_3$ are other sources of heat. Changes in thermal energy over time $(\rho C_p \partial T / \partial t)$ can be caused by several different processes. The term $\rho C_p \boldsymbol{v} \cdot \boldsymbol{\nabla} T$ corresponds to advective heat transport, i.e., the transport of thermal energy with the motion of material. Consequently, it depends on the velocity $\boldsymbol{v}$ the material moves with, and on how much the temperature, and with that the thermal energy, changes in space, which can be expressed through

the temperature gradient $\boldsymbol{\nabla} T$. Heat is also transported by conduction, which is represented by the term $\boldsymbol{\nabla} \cdot (k\boldsymbol{\nabla} T)$. When the thermal conductivity $k$ is larger, or the temperature variation as expressed by the gradient $\boldsymbol{\nabla} T$ becomes steeper, more heat is diffused. If the model includes no additional heating processes, Eq. (6) can be simplified by dividing it by $\rho C_p$, assuming that they are constants:

$$\frac{\partial T}{\partial t} + \boldsymbol{v} \cdot \boldsymbol{\nabla} T = \boldsymbol{\nabla} \cdot (\kappa \boldsymbol{\nabla} T) \tag{7}$$

which introduces the **thermal diffusivity** $\kappa = k/(\rho C_p)$.

The terms on the right-hand side of Eq. (6) describe other heating processes that can be significant in a geodynamic model. **Internal heating** can be expressed as the product of the density $\rho$ and a volumetric heat production rate $H$. Its most common source is the decay of radiogenic elements. Accordingly, this process is important in regions where the concentration of heat-producing elements is high, such as the continental crust. **Viscous dissipation**, also called shear heating, describes the amount of energy that is released as heat when material is deformed viscously and/or plastically. The more stress required to deform the material, and the larger the amount of deformation, the more shear heating is produced. This can be expressed as $S_1 = \boldsymbol{\sigma}' : \dot{\boldsymbol{\varepsilon}}'$, where $\boldsymbol{\sigma}'$ is the deviatoric stress, i.e. the part of the stress not related to the hydrostatic pressure, $\dot{\boldsymbol{\varepsilon}}'$ is the non-recoverable, that is non-elastic, deviatoric strain rate (see also equations (8) and (9) in Section 2.2.1) and : is the matrix inner product, so $\boldsymbol{\sigma}' : \dot{\boldsymbol{\varepsilon}}' = \sum_i \sum_j \sigma'_{ij} \dot{\varepsilon}'_{ij}$. **Adiabatic heating** describes how the temperature of the material changes when it is compressed or when it expands due to changes in pressure, assuming that no energy is exchanged with the surrounding material during this process. Hence, the work being done to compress the material is released as heat. This temperature change depends on the thermal expansivity $\alpha$, which expresses how much the material expands for a given increase in temperature, and on the changes in pressure over time: $S_2 = \alpha T Dp/Dt$. Here, $Dp/Dt = \partial p/\partial t + \boldsymbol{v} \cdot \nabla p$ is the material (Lagrangian) derivative of the pressure. Since the dominant pressure variation in the Earth's interior is the effect of the lithostatic pressure increase with depth, the usual assumption is that pressure only changes due to the vertical motion of material with velocity $v_z$ along a (lithostatic) pressure gradient $\boldsymbol{\nabla} p = \rho g \hat{\boldsymbol{z}}$ (with $\hat{\boldsymbol{z}}$ the unit vector in the vertical direction), resulting in $S_2 = \alpha T \rho g v_z$.

When material undergoes phase changes, thermal energy is consumed (endothermic reaction) or released (exothermic reaction) as latent heat. This happens both for solid-state phase transitions, such as from olivine to wadsleyite, and for melting of mantle rocks. For phase transitions that occur over a narrow pressure/temperature range, this can lead to abrupt changes in mantle temperature where phase transitions are located, such as around 410 km depth. The amount of energy released or consumed is proportional to the density $\rho$, the temperature $T$, and the change in the thermodynamic quantity entropy across the phase transitions $\Delta S$, i.e., $S_3 = \rho T \Delta S DX/Dt$ (Schubert et al., 2001, Chapter 4.1). Here, $X$ is the fraction of the material that has already undergone the transition, sometimes also called the phase function. For mantle melting, $X$ would be the melt fraction. The terms discussed above are generally the most important heating processes in the Earth's interior. While there may be other sources of heat or heat transport, they are assumed to be so small that they can be neglected in most cases, such as heat transport by radiation, or the energy required for reducing the mineral grain size and creating new boundaries between the grains.

### 2.1.4 Approximations of the governing equations

In the previous sections on the mass, momentum, and energy equations, we have already seen that there are different ways these equations can be simplified and that there is a choice in which physical processes to include. Based on an analysis of the equations, there are a number of different approximations that are commonly used in geodynamics (see also Gassmöller et al., 2020; Schubert et al., 2001). They all make choices about how the density is approximated, so it is important to understand the magnitude of density variations in a model (see also Section 2.2.5). In the Earth's mantle, the primary density variations that drive flow are usually caused by temperature, and are on the order of 1% of the total density (assuming a thermal expansivity of $10^{-5}$ K$^{-1}$ and temperature variations of about 1000 K). Density variations due to chemical heterogeneities have a similar magnitude. Density jumps at phase transitions can reach 500 kg/m$^3$, which is on the order of 10% of the total density. The increase in lithostatic pressure from the surface to the core-mantle boundary causes a density increase of 65% (Schubert et al., 2001), but these density changes do not drive the flow. Density variations caused by the dynamic pressure are much smaller, on the order of 0.1% of the total density (assuming dynamic pressures of up to 500 MPa). Note that these are estimates for the Earth's mantle, and these values may change for other applications. For example, a model of salt domes has a much larger compositional density contrast between the salt and the rock, and certain dehydration reaction can lead to volume changes larger than 10%. Before deciding to use any of the following approximations in a geodynamic model, it is therefore important to make sure that the underlying assumptions about the density variations are justified for that particular application.

The **Anelastic Liquid Approximation** (ALA, Jarvis and McKenzie, 1980) assumes that lateral density variations are small relative to a reference density profile and can be neglected in the mass (3) and energy conservation equations (6), which instead use the density of the depth-dependent reference profile. Only the buoyancy term in the momentum equation (5) uses a temperature- and pressure-dependent density, which is approximated using a Taylor expansion (Appendix A1). This approximation is commonly used in whole-mantle convection models, where the density changes substantially from the surface to the core-mantle boundary, and compressibility is an important effect. Its use is appropriate as long as deviations from the reference profile remain small or are not important for the interpretation of the model. For example, substantial expansion or contraction caused by temperature changes that are not on the reference profile, such as cooling or heating in the thermal boundary layers caused by conduction, and the stresses caused by this process, cannot be modelled with this approximation. Phase transitions of different types of materials can also cause large density or temperature changes, since only one type of material can be represented by the reference profile (see also Gassmöller et al., 2020).

The **Truncated Anelastic Liquid Approximation** (TALA, Jarvis and McKenzie, 1980; Ita and King, 1994) is based on the same assumptions as the ALA, but further assumes that the variation of the density due to pressure variations can be neglected in the buoyancy term as well. In the Earth's mantle, these density variations are typically an order of magnitude smaller than the density changes caused by temperature. The TALA has similar applications as ALA, but introduces an imbalance between energy dissipation calculated from the Stokes equation and heat dissipation in the energy equation. Consequently, it should be avoided when energy dissipation is important in the model (Leng and Zhong, 2008; Alboussière and Ricard, 2013).

The **Boussinesq approximation** (BA, Oberbeck, 1879; Boussinesq, 1903; Rayleigh, 1916) assumes that density variations are so small that they can be neglected everywhere except in the buoyancy term in the momentum equation (5), which is equivalent to using a constant reference density profile. This implies incompressibility, i.e., the use of Eq. (4) for mass conservation. In addition, adiabatic and shear heating are not considered in the energy equation (6). This approximation is valid as long as density variations are small (see also Section 2.1.1) and the modelled processes would cause no substantial shear or adiabatic heating. The Boussinesq approximation is often used in lithosphere-scale models. Due to its simplicity, the approximation of incompressibility is sometimes also adopted for whole-mantle convection models, where it is only approximately valid and it has been shown that compressibility can have a large effect on the pattern of convective flow (e.g. Tackley, 1996).

The **extended Boussinesq approximation** (EBA, Christensen and Yuen, 1985; Oxburgh and Turcotte, 1978) is based on the same assumptions as the BA, but does consider adiabatic and shear heating. Since it includes adiabatic heating, but not the associated volume and density changes, it can lead to artificial changes of energy in the model, i.e., material is being heated or cooled based on the assumption that it is compressed or it expands, but the mechanical work that causes compression or expansion is not done. Consequently, the extended Boussinesq approximation should only be used in models without substantial adiabatic temperature changes.

For a comparison between some of these approximations using benchmark models, see e.g., Steinbach et al. (1989); Leng and Zhong (2008); King et al. (2010); Gassmöller et al. (2020). In addition, the choice of approximation may also be limited by the numerical methods being employed (for example, the accuracy of the solution for the variables that affect the density). Also note that technically, these approximations are all internally inconsistent to varying degrees (Section 6.2), since they do not fulfil the definitions of thermodynamic variables, but use linearised versions instead, and use different density formulations in the different equations. Nevertheless, many of them are generally accepted and widely used in geodynamic modelling studies, as they allow for simpler equations and more easily obtained solutions.

## 2.2 The constitutive equations: Material properties

Using the equations discussed in the previous section 2.1 to model how Earth materials move under the influence of gravity requires some assumptions about how Earth materials respond to external forces or conditions. These relations are often called constitutive equations, and they relate the material properties to the solution variables of the conservation equations, like temperature and pressure. Which of these relations is most important for the model evolution depends on the application. In regional models that impose the plate driving forces externally, the relation between stress and deformation, i.e., the rheology (Section 2.2.1) can be the most important parameter choice in the model, and can control most of the model evolution. Since buoyancy is one of the main forces that drive the flow of Earth materials, it is often most important to take into account how the density depends on other variables, and to include this dependency in the buoyancy term, i.e., the right-hand side of Eq. (5). This is particularly important for mantle convection models, and is described by the **equation of state** (Section 2.2.5) of the material. In models of the lithosphere and crust, stress and state of deformation become more important. The constitutive equations can also include time-dependent terms, like in cases where the strength of a rock depends on its deformation history, requiring additional equations to be solved (see also Section 2.3).

### 2.2.1  Rheology of Earth materials

The **rheology** describes how materials deform when they are exposed to stresses. This relation between stress and deformation
enters the momentum equation (5) in the term $\nabla \cdot \boldsymbol{\sigma}$, which represents the surface forces, and is also used to compute the amount
of shear heating $\boldsymbol{\sigma}' : \dot{\boldsymbol{\varepsilon}}'$ in the energy balance. In many cases, the material reacts differently to shear stresses (acting parallel
to the surface) and normal stresses (acting normal to the surface and leading to volumetric deformation). Correspondingly, the
resistance to these different types of deformation is expressed by different material parameters: the shear viscosity or shear
modulus, and the bulk viscosity or bulk modulus.

Rocks deform in different ways, necessitating different rheologies. For example, rocks can deform elastically or by brittle
failure. On long time scales their inelastic deformation is usually modelled as that of a highly viscous fluid. Based on this latter
assumption, we use the Stokes equations to describe the deformation. This physical model adequately explains many processes
in the Earth's interior related to mantle convection and observations. However, some observations such as plate tectonics, which
involves strain localisation and strong hysteresis, and the existence of ancient geochemical heterogeneity revealed by isotopic
studies are behaviours not commonly associated with fluids. Indeed, in fluids, different chemical components are well-mixed in
a convective flow, and the flow usually does not depend on the deformation history and has no material memory. Geodynamic
models still aim to reproduce such processes using complex rheologies that go beyond the basic assumption of a viscous fluid
and include plastic yielding, strain- or strain-rate weakening or hardening, and other friction laws (see below). Hence, resolving
how to go forward with the assumption of a viscous fluid in the face of complex deformation behaviour of rocks remains an
important challenge for the geodynamic modelling community.

In the following, we will discuss some common rheological behaviours being used in geodynamic models.

### 2.2.2  Viscous deformation

We start by discussing a very simple type of rheology with the following features: (i) only viscous (rather than elastic or brittle)
deformation, (ii) deformation behaviour that does not depend on the direction of deformation (corresponding to an isotropic
material), (iii) a linear relation between the stress and the rate of deformation (corresponding to a **Newtonian fluid**), and
(iv) a low amount of energy dissipation under compaction or expansion (corresponding to a negligible bulk viscosity). In this
case, the (symmetric) stress tensor $\boldsymbol{\sigma}$ can be written as:

$$\boldsymbol{\sigma} = -p\boldsymbol{I} + \boldsymbol{\sigma}' = -p\boldsymbol{I} + 2\eta\,\dot{\boldsymbol{\varepsilon}}'(\boldsymbol{v}), \tag{8}$$

where $p$ is the pressure, $\boldsymbol{I}$ is the unit tensor, $\boldsymbol{\sigma}'$ is the deviatoric stress tensor, $\eta$ is the (dynamic) shear viscosity and $\dot{\boldsymbol{\varepsilon}}'(\boldsymbol{v})$ is
the deviatoric rate of deformation tensor (often called deviatoric strain rate tensor), which is directly related to the velocity
gradients in the fluid. The total strain rate tensor is defined as

$$\dot{\boldsymbol{\varepsilon}}(\boldsymbol{v}) = \frac{1}{2}(\nabla\boldsymbol{v} + \nabla\boldsymbol{v}^T). \tag{9}$$

It describes both volume changes of the material (volumetric) and changes in shape (deviatoric), and can be written as the sum
of these two deformation components: $\dot{\boldsymbol{\varepsilon}}(\boldsymbol{v}) = \frac{1}{3}(\nabla \cdot \boldsymbol{v})\boldsymbol{I} + \dot{\boldsymbol{\varepsilon}}'(\boldsymbol{v})$. The prime denotes the deviatoric part of the tensor, i.e., the

part that refers to the change in shape. It follows that the case of incompressible flows, where the density is assumed to be constant and $\nabla \cdot \boldsymbol{v} = 0$ (Eq. 4), also implies $\dot{\boldsymbol{\varepsilon}}(\boldsymbol{v})' = \dot{\boldsymbol{\varepsilon}}(\boldsymbol{v})$. Hence, incompressibility implies that there is no volume change of the material, so this part of the strain rate tensor is not taken into account.

In the Earth's interior, viscous deformation is the dominant rock deformation process on long time scales, if temperatures are not too far from the rocks' melting temperature. Under these conditions, imperfections in the crystal lattice move through mineral grains and contribute to large-scale deformation over time. The physical process that is thought to most closely resemble this idealised case of a Newtonian fluid is **diffusion creep**, where single atoms or vacancies, i.e., the places in the crystal lattice where an atom is missing, migrate through the crystal. Diffusion creep is assumed to be the dominant deformation mechanism in the lower mantle. Other creep processes exhibit different behaviour. **Dislocation creep**, which is considered to be important in the upper mantle, is enabled by defects in the crystal lattice that are not just a single atom, but an entire line of atoms. Dislocation creep is highly sensitive to the stress applied to a rock, which means that the relation between stress and strain rate is no longer linear, but a power law. **Peierls creep**, which has an exponential relation between stress and rate of deformation, becomes important at even higher stresses (Gerya, 2019). **Grain boundary sliding** describes deformation due to grains sliding against each other along grain boundaries, which has to be accommodated by another mechanism that allows the grains themselves to deform (e.g., diffusion or dislocation creep). It becomes important at high temperatures and high stresses (Hansen et al., 2011; Ohuchi et al., 2015).

Consequently, the viscosity of rocks strongly depends on e.g., temperature, pressure, stress, size of the mineral grains, deformation history, and the presence of melt or water and it varies by several orders of magnitude, often over small length scales. These experimentally-obtained flow laws can be expressed in a generalised form using the relation (Hirth and Kohlstedt, 2003):

$$\dot{\varepsilon} = A\sigma_d{}^n d^{-q} f_{\mathrm{H_2O}}^r \exp(\alpha^\star \phi) \exp\left(-\frac{E + pV}{RT}\right), \tag{10}$$

where $\dot{\varepsilon}$ is the strain rate, $\sigma_d$ is the applied differential stress, $d$ is the grain size, $f_{\mathrm{H_2O}}$ is the water fugacity and relates to the presence of water, $\phi$ is the melt fraction, and $A$, $n$, $q$, $r$, $\alpha^\star$, $E$, $V$ and $R$ are constants. In geodynamic modelling, this flow law is usually recast in terms of the square root of the second invariant of the deviatoric strain rate or stress tensors to obtain the effective viscosity (see Chapter 6 of Gerya, 2019). Each of these parameters (except for $R$, which is the gas constant) varies depending on the mineral and creep mechanism. Which of these dependencies is most important depends on the type of model. For example, diffusion creep exhibits a linear relation between stress and strain rate, so $n = 1$, whereas dislocation creep strongly depends on stress (typically $n \sim 3-4$), but there is no dependence on grain size (i.e., $q = 0$). Accordingly, temperature is probably the most important control in the lower mantle, while in the upper mantle, the stress-dependence plays a dominant role. Grain size $d$ is another potentially strong influence (e.g. Hirth and Kohlstedt, 2003; King, 2016; Jain et al., 2018), but since the grain size in the mantle is not well-constrained it is hard to include this effect in geodynamic models (Dannberg et al., 2017). While the presence of melt can also lead to substantial weakening of rocks, the exact dependence on melt fraction is less well-constrained than other dependencies, and additional equations for the motion of melt are usually required to model a

realistic melt distribution (see also Section 2.3). Generally, several deformation mechanisms with different flow laws (Eq. 10) can be active at the same time, and their individual strain rates add up to the total 'effective' rate of deformation.

 ### 2.2.3 Elastic deformation

On shorter time scales, the elastic behaviour of rocks becomes important in addition to viscous flow. In the case of elasticity, the stress tensor is related to the strain tensor through the generalised Hooke's law $\boldsymbol{\sigma} = \boldsymbol{C} : \boldsymbol{\varepsilon}(\boldsymbol{u})$ where $\boldsymbol{C}$ is the fourth-order elastic tensor and $\varepsilon$ is the strain tensor which depends on the displacement vector $\boldsymbol{u}$ as follows:

$$\boldsymbol{\varepsilon} = \frac{1}{2}(\boldsymbol{\nabla}\boldsymbol{u} + \boldsymbol{\nabla}\boldsymbol{u}^T). \tag{11}$$

Hence, for elastic deformation, the stress is proportional to the amount of deformation rather than the rate of deformation, as is the case for viscous processes. Due to the inherent symmetries of $\boldsymbol{\sigma}$, $\boldsymbol{\varepsilon}$, and $\boldsymbol{C}$, the tensor $\boldsymbol{C}$ is reduced to two numbers for homogeneous isotropic media, and the stress–strain relation becomes:

$$\boldsymbol{\sigma} = \lambda(\boldsymbol{\nabla} \cdot \boldsymbol{u})\boldsymbol{I} + 2\mu\boldsymbol{\varepsilon} \tag{12}$$

where $\lambda$ is the first Lamé parameter and $\mu$ is the second Lamé parameter (also referred to as shear modulus), which describes
how much the material deforms elastically under a given shear stress. These two moduli together define the Poisson's ratio $\nu$ (also see Section 2.1.1).

Elastic deformation is often included in geodynamic codes that solve the Stokes equations by taking the time derivative of Eq. (12), which introduces the velocity and strain-rate. The term $\dot{\boldsymbol{\sigma}}$ is then approximated by a first order Taylor expansion (see Appendix A) which ultimately amounts to adding terms to the right hand side of the momentum equation (Eq. (5); see for
example von Tscharner and Schmalholz (2015)).

### 2.2.4 Brittle deformation

For strong deformation and large stresses on short time scales, such as occurring in the crust and lithosphere, brittle deformation becomes the controlling mechanism. In this case, a fault or a network of faults (which can range from the atomic to the kilometre-scale) accommodates deformation. The relative motion of the two discrete sides of the fault is limited by the friction
on the interface. However, in a continuum formulation, discontinuous faults cannot be represented and hence the deformation in geodynamics models typically localises in so-called **shear bands** of finite width, which can be interpreted as faults on crustal and lithospheric scales (van Zelst et al., 2019). We approximate the brittle behaviour by so-called **plasticity**, characterised by a yield criterion that determines the yield stress (also called yield strength) that the maximum stress must satisfy. When locally built-up stresses reach this yield stress, the rock is permanently deformed and the plastic strain rate is no longer zero. Note
that it is difficult to compare a tensor (the stress) and a scalar (the yield stress). This is particularly important as the difference between the standard yield criteria is partly based on the way that this comparison is made, i.e., the stress invariant for the von Mises and Drucker-Prager yield criteria, which leads to a smooth yield envelope, versus shear stress for the Tresca and

Mohr-Coulomb yield criteria, which leads to a segmented yield envelope. Also, in other Earth science communities the terms 'plastic deformation' and 'plasticity' are used to describe all non-recoverable, time-dependent deformation (e.g. solid-state creep, like dislocation creep) that is commonly also referred to as ductile or viscous deformation (e.g., Karato, 2008; Fossen, 2016). Here we use 'plasticity' to refer to non-recoverable, nearly instantaneous yielding at high stresses and fracturing.

One of the most well-known yield criteria is the Mohr-Coulomb criterion (Handin, 1969; Kachanov, 2004), where yielding occurs when:

$$\tau_m = C \cos \phi_f + \sigma_m \sin \phi_f. \tag{13}$$

where $\tau_m = \frac{\sigma_1 - \sigma_3}{2}$ is the maximum shear stress, $\sigma_m = \frac{\sigma_1 + \sigma_3}{2}$ is the normal stress, $\sigma_1$ and $\sigma_3$ are the maximum and minimum principle stresses, respectively, $C$ is the cohesion and $\phi_f$ is the angle of friction. The strength of rocks is primarily characterised by the two parameters $C$ and $\phi_f$ that are obtained from laboratory measurements.

Often Eq. (13) is rewritten as $F = \tau_m - C \cos \phi_f - \sigma_m \sin \phi_f = 0$ where $F$ is the yield criterion. $F > 0$ is not allowed as the stresses would then exceed the strength of the rock. This function can be formulated as a function of the pressure and the second and third deviatoric stress invariants (see Appendix B of Thieulot, 2011, for the invariants expression). For each location in the model domain, if $F < 0$ then deformation is (elasto-)viscous, otherwise plastic deformation occurs and a solution satisfying $F = 0$ is found.

In a Mohr's circle diagram of shear stress $\tau$ versus normal stress $\sigma_n$, the Mohr-Coulomb yield function or envelope is given by

$$\tau = C + \mu_f \, \sigma_n \tag{14}$$

where $\mu_f = \tan \phi_f$ represents the **friction coefficient** of the rock. It is clear from this equation that the effective strength of the rock increases with the normal stress. At $F = 0$, the Mohr's circle touches the yield function.

Experimentally, Byerlee (1978) measured that $\tau = \mu_f \, \sigma_n = 0.85\sigma_n$ for low stresses up to approximately 200 MPa and $\tau = C + \mu_f \, \sigma_n = 50 + 0.6\sigma_n$ (in MPa units) for higher normal stresses. This relationship between the shear stress and the normal stress is commonly referred to as **Byerlee's law**.

The **numerical implementation** of the Mohr-Coulomb yield criterion is convoluted, because of the shape of the yield envelope in the principle stress space $(\sigma_1, \sigma_2, \sigma_3)$. As a consequence, the Drucker-Prager yield criterion (Drucker and Prager, 1952) is often preferred. In the case of an incompressible, two-dimensional (2-D) model with a **plane strain assumption**, the Drucker-Prager and Mohr-Coulomb yield criteria are equivalent for consistently chosen parameters. However, in general the Drucker-Prager and Mohr-Coulomb criteria are not equivalent and the resulting strength of the rocks can differ by as much as 300% between them (Wojciechowski, 2018) for the same friction angle and cohesion parameters. In contrast to the above Drucker-Prager and Mohr-Coulomb criteria, the simpler von Mises and Tresca yield criteria do not consider a normal stress or pressure-dependence of the yield stress.

A common simplification of the Drucker-Prager and Mohr-Coulomb yield criteria is to use the lithostatic pressure $p_{\text{lith}} = \rho g z$ (where $z$ is depth) instead of the total pressure, thereby ignoring contributions to the pressure by, e.g., the dynamic pressure and

pore fluid pressure. This assumption effectively makes the yield criterion purely depth-dependent and is sometimes referred to as the depth-dependent von Mises criterion (Spiegelman et al., 2016). If only lithostatic pressure is used in the yield criterion, shear bands are always orientated at 45° with regards to the principal stress directions under both extension and compression. This assumption is allowed for the mantle, where the total pressure is close to the lithostatic pressure. However, pressure can deviate strongly from the lithostatic pressure in the lithosphere, which can have major effects on the results.

In nature, the strength of rocks can change over time and depends on the deformation history. Examples are the evolution of the mineral grain size, formation of a fault gouge, and percolation of fluids, which alter the strength of the rock. To account for this variation in strength over time on tectonic time scales, the cohesion and friction coefficient of the rock can be made dependent on the strain or strain-rate in what is called strain- or strain-rate-weakening (also called softening): when the strain or strain rate increases, the strength of the rock is lowered. Similarly, strain- or strain-rate-hardening can be applied (e.g., Massmeyer et al., 2013). When geodynamic models are used to study fault and earthquake processes, they focus on a shorter time scale of rock strength variations (Figure 1) and a more sophisticated way of strength variations needs to be used. Experiments have shown that the strength of the rock is better represented as depending on slip velocity (i.e., the velocity with which the two sides of a fault slide past each other). In hybrid seismo-thermo-mechanical models, it is often assumed that the slip velocity is equivalent to the strain-rate apart from a multiplication with an assumed constant fault width (van Dinther et al., 2013a). Complex friction laws based on the parameterisation of lab experiments have been developed to describe the change in friction coefficient $\mu_f$ (Eq. 14) of rocks during an earthquake varying from simple linear slip weakening approximations (Ida, 1973) to complex rate- (van Dinther et al., 2013a; Dal Zilio et al., 2018; van Zelst et al., 2019) and rate-and-state dependent friction laws (Lapusta et al., 2000; Sobolev and Muldashev, 2017; Herrendörfer et al., 2018).

In the Earth, the relation between stress and deformation can be very complex, and deformation is a combination of elastic and viscous behaviour and brittle failure. In general, both the viscosity and the elastic moduli can depend on temperature, pressure, chemical composition, the presence of melt and fluids, the size and orientation of mineral grains, the rate of deformation and the deformation history of the material. Consequently, Earth materials are usually not isotropic, and the strength of the material depends on the direction of deformation. To incorporate this behaviour into geodynamic models, the viscosity and elastic moduli have to be expressed as tensors, and cannot be reduced to one or two parameters (Mühlhaus et al., 2002; Lev and Hager, 2008). This complexity is not taken into account in most geodynamic models. This is on the one hand because it is computationally expensive and requires substantial effort to implement in a model, but also because there is substantial uncertainty in how, specifically, rocks deform anisotropically. For models that include anisotropy, see for example Mühlhaus et al. (2002); Kocher et al. (2006); Lev and Hager (2008); Heister et al. (2017); Faccenda (2014); Perry-Houts and Karlstrom (2018); Király et al. (2020a).

### 2.2.5 The equation of state

The relation between density, temperature, pressure, and sometimes other variables like chemical composition is often called the **equation of state**. It describes the state of the material (density) under a set of conditions (temperature, pressure, etc.).

It incorporates material properties such as the thermal expansivity $\alpha$, which describes how much the rock expands when the temperature is increased, and the compressibility, which describes how much the volume of a rock decreases when it is exposed to higher pressures. These relations should be chosen in a way that is thermodynamically consistent (see Section 6.2).

Depending on the model application, there are many different equations of state that can be used. For models that aim to capture the first-order effects of a given process, analyse the influence of a material property on the model evolution, or
540 develop a scaling law (see section 5.2.2), it is often appropriate to simplify the relationships between solution variables and material properties. In mantle convection models, the temperature usually has the largest influence on density variations that affect buoyancy. So in the simplest case, the equation of state may be a linear relationship between density and temperature. For example, it could take the form $\rho = \rho_0(1 - \alpha\Delta T)$ with $\rho_0$ being the constant reference density, and $\Delta T = T - T_0$ with $T_0$ being the constant reference temperature at which the density equals $\rho_0$. This relationship is commonly used in the Boussinesq
approximation. However, the most important variables depend on the application. For example, chemical composition plays an important role in both the outer core, where temperature variations are very small, and in lithospheric deformation.

On the other end of the spectrum, there are models designed to fit existing observations (e.g., seismic wave speeds, see also Section 5.2.1). In such a scenario, it is often desirable to include the material properties in the most realistic way that our current knowledge allows. This requires knowledge about mineral physics and thermodynamics, which is often handled by
550 external thermodynamics programs. Software packages such as Perple_X (Connolly, 1990), HeFESTo (Stixrude and Lithgow-Bertelloni, 2005) and BurnMan (Cottaar et al., 2014) in connection with a mineral physics database, include the complex thermodynamic relations that determine how mineral assemblages evolve under Earth conditions. They compute material properties in dependence of the solution variables of the geodynamic model (usually temperature and pressure, but alternatively, specific entropy and volume, or a combination of these variables, can be used; see Connolly, 2009). Two-way coupling
of geodynamic and thermodynamic models is challenging, and there are ongoing community efforts to improve model inter-operability (e.g. ENKI, http://enki-portal.org). On the other hand, one-way coupling, where pre-computed look-up tables from thermodynamic software for a fixed composition and pressure–temperature range are used to determine the material properties in geodynamic models has become a common approach (Gerya et al., 2004; Nakagawa et al., 2009; Faccenda and Dal Zilio, 2017; Rummel et al., 2020).

## 2.3 More complex processes

The mass, momentum, and energy conservation equations are visibly coupled since velocity (derivatives) and pressure enter the stress tensor, and thermal energy transport due to advection depends on the velocity. More importantly, the previous sections have highlighted how material properties such as $\alpha$, $\rho$, and $C_p$ depend on pressure, temperature and other quantities such as composition and that deformation is partitioned between various mechanisms. Their pressure-, temperature- and strain rate-
565 dependence makes the exact partitioning complex and renders the conservation equations (2), (5) and (6) very **nonlinear** with regards to the primary variables $v$, $p$ and $T$. Hence, the equations usually contain nonlinear terms that imply a relationship between the solution variables that is no longer linear.

In addition to temperature, pressure, and velocity there may be other conditions in the model that change over time and are important for the model evolution, but not directly related to changes in temperature, pressure and velocity. A common example is the chemical composition of the material (which can refer to the major element composition, but may also relate to the water content for example). In this case, a transport equation is required for every additional quantity that should be tracked in the model and moves with the material flow:

$$\frac{\partial c_i}{\partial t} + \boldsymbol{v} \cdot \nabla c_i = 0 \qquad i = 1, \ldots n_c \tag{15}$$

where $c_i$ is the to-be-tracked quantity, $t$ is time, $\boldsymbol{v}$ is velocity, and $n_c$ is the number of compositions, fields, or materials present. This advection equation assumes that at any given location, changes in composition over time $\partial c_i / \partial t$ are caused by the transport of material with the velocity $\boldsymbol{v}$ of the flow. The stronger the spatial variations in composition, expressed by the gradient of the composition $\nabla c_i$, and the faster the flow, the bigger the changes in composition over time. Another way to think about this is that this equation describes the mass conservation of each composition (compare to Eq. 2). Consequently, if there are other processes that influence the composition, like chemical reactions or diffusion, the corresponding terms would have to be included in the equation.

Other physical processes may require additional terms or additional equations. Examples are the generation of the magnetic field in the outer core (Jones, 2011), two-phase (McKenzie, 1984; Bercovici et al., 2001) or multi-phase flow (Oliveira et al., 2018; Keller and Suckale, 2019), disequilibrium melting (Rudge et al., 2011), complex magma dynamics in the crust (Keller et al., 2013), reactive melt transport (Aharonov et al., 1995; Keller and Katz, 2016), (de)hydration reactions and water transport (Faccenda et al., 2012; Magni et al., 2014; Quinquis and Buiter, 2014; Wilson et al., 2014a; Nakagawa et al., 2015), the evolution of the mineral grain size (Hall and Parmentier, 2003; Solomatov and Reese, 2008; Bercovici and Ricard, 2012; Cerpa et al., 2017; Mulyukova and Bercovici, 2019), the fluid dynamics and thermodynamics of a magma ocean (Labrosse et al., 2007; Solomatov, 2000; Ulvrová et al., 2012), the interaction of tectonic processes with erosion and other surface processes (Burov and Cloetingh, 1997; Roe et al., 2006; Thieulot et al., 2014; Ueda et al., 2015; Sternai, 2020), anisotropic fabric (Mühlhaus et al., 2002; Lev and Hager, 2008; Heister et al., 2017; Faccenda, 2014; Perry-Houts and Karlstrom, 2018; Király et al., 2020a), phase transformation kinetics (Bina et al., 2001; Tetzlaff and Schmeling, 2009; Quinteros and Sobolev, 2012; Agrusta et al., 2014) and inertial processes and seismic cycles (van Dinther et al., 2013b, a; Sobolev and Muldashev, 2017; Tong et al., 2019). Since the aim of this contribution is to introduce the general concepts of geodynamic modelling, we will not discuss these more complex effects here, and will instead focus on the main conservation equations presented above.

In the end, the partial differential equations (2), (5), (6), and potentially (15), as well as the constitutive equations must be supplemented by the geometry of the domain on which they are to be solved (e.g., 2-D/3-D, Cartesian, cylindrical, spherical; Section 5), a set of initial conditions for the time-dependent temperature and compositions and a set of boundary conditions (Section 5.3).

## 3 The numerical model

The conservation equations of Section 2.1 generally cannot be solved analytically. However, computers are capable of finding approximate solutions to equations (2), (5) and (6). To this end, we must define a **numerical model**, i.e. the mathematical description of the **physical model** in a computer language, to solve numerically. As computers cannot by construction represent a continuum, the above equations must be discretised and then solved on a finite domain divided into a finite number of **grid points**, which together form the mesh of the computation. This process requires specific numerical methods (Section 3.1) and choices (Section 3.2-3.7) and verification of the resulting numerical code (Section 4). For a recent in-depth analysis and outlook of numerical methods in geodynamics, we refer the reader to Morra et al. (2020).

### 3.1 Numerical methods

The solutions of the continuum equations described in Section 2.1 need to be computed on a finite number of moments in time and locations in space. Rewriting the equations of Section 2.1 for a discrete number of points is called discretisation. The three main methods of discretisation in geodynamics are the **finite element method** (FEM), the **finite difference method** (FDM) and the **finite volume method** (FVM). The last two are equivalent in some instances, such as in the case of the commonly-used staggered grid finite difference discretisation in geodynamic codes. Often combinations of these methods are used to deal with time and space discretisation separately. All three discretisation methods convert Ordinary Differential Equations (ODEs) or Partial Differential Equations (PDEs), which may be **nonlinear**, into a system of linear equations that can be solved by matrix algebra techniques. None is intrinsically better than the other, although there are differences that make a certain method more suitable for certain types of science questions. For instance the finite element and finite volume methods are capable of dealing with non-Cartesian geometries, such as the spherical shape of the Earth, or topography at the surface of a model, while the finite difference method is not. On the other hand, the finite difference method is more intuitive in its design than the other two. An overview of the basic principles of these numerical methods is available in Ismail-Zadeh and Tackley (2010) and in Zhong et al. (2015) and an in-depth exposé of the finite difference method in geodynamics is available in Gerya (2019). It is worth noting that **spectral methods** are also encountered in mantle dynamics modelling (Ismail-Zadeh and Tackley, 2010; Petrunin et al., 2013), as is the **boundary element method** (BEM) (Morra et al., 2009, 2010) and the Lattice Boltzmann Method (Mora and Yuen, 2017). Meshless methods such as the **discrete element method** (DEM, Buiter et al., 2016), the **smoothed particle hydrodynamics method** (SPH, Golabek et al., 2018) and **radial basis function method** (RBF, Arrial et al., 2014) have also been used in geodynamic modelling. In what follows we focus on the most popular methods, i.e. finite differences, finite elements, and finite volumes.

### 3.2 Discretisation

The discretisation concept for all three main methods (FEM, FDM, FVM) is identical. The domain is subdivided in cells or elements, as shown in Figure 4. In essence, the methods look for the solution of the equations of Section 2.1 in the form of combinations of polynomial expressions (also called shape functions in FEM terminology) defined on each element or cell.

To illustrate this concept, we provide a small example here for the conservation of energy using the finite difference method, which is based on a Taylor expansion keeping only first and second order terms (see Appendix A for the complete example). In one dimension, and under the assumptions that there is no advection or heat sources, and that the coefficients are all constant in space, Eq. (6) becomes what is commonly called the heat equation:

$$\frac{\partial T}{\partial t} = \kappa \frac{\partial^2 T}{\partial x^2}. \tag{16}$$

This is to be solved for $T(x,t)$, for $x$ between 0 and $L_x$, and $t$ between 0 and $t_{\text{final}}$. The time discretisation describes how to use a temperature distribution that is known at time $t_n$ to compute a new temperature at time $t_{n+1} = t_n + \Delta t$, with $\Delta t$ being the time step size. The discretisation in space means that this temperature distribution is computed at a finite number $np$ of points. The simplest way of choosing the position of these points is such that $x_i = ih$, with $h = L_x/(np-1)$ being the distance between points (which is often referred to as the resolution or grid size of the model) and the point indices running from $0 \leq i \leq np-1$. As shown in Appendix A, there are different ways of discretising Eq. (16), but the so-called explicit version writes

$$\frac{T_i^{n+1} - T_i^n}{\Delta t} = \kappa \frac{T_{i+1}^n - 2T_i^n + T_{i-1}^n}{h^2}, \tag{17}$$

where the subscripts refer to space indices and the superscripts refer to time indices, i.e., $T_i^n = T(x_i, t_n)$. This also means that $T_i^0$ is the initial temperature condition which is needed to start the calculation. In this example, we know all temperatures at time step $n$ and we can easily compute the temperatures at time step $n+1$ at all locations $i$ inside the domain, while the temperature on the boundary $T(x_{i=0})$ and $T(x_{i=np-1})$ is prescribed by the boundary conditions. It also illustrates what the name of the finite difference method means: when going from the continuum to a finite number of grid points, derivatives are approximated by differences in temperature between these points. In addition, 'finite' refers to the mathematical definition of a derivative as a limit where $h \to 0$ is replaced by a formula in which $h$ remains finite (see Appendix A).

While finite difference codes only need to specify the order of the approximation they rely on (and the associated **stencil**), finite element codes often specify the type of element they use (i.e., the specific polynomial expressions for the shape functions). This also controls how the partial differential equations are solved on the grid, with finite difference methods solving the equations pointwise and finite element methods averaging the equations per element. Earlier codes such as Citcom(S/CU) (Moresi and Solomatov, 1995), ConMan (King et al., 1990) and SOPALE (Fullsack, 1995) relied on the computationally cheap (i.e. yielding the smallest possible linear system size) $Q_1 \times P_0$ element. This finite element pair is known to be unstable and this manifests itself via so-called pressure modes (see Section 6.1). Stable pairs are to be preferred and modern codes such as ASPECT (Kronbichler et al., 2012) and pTatin3D (May et al., 2015) rely on the more accurate $Q_2 \times Q_1$ and $Q_2 \times P_{-1}$ (sometimes denoted $Q_2 \times P_1^{\text{disc}}$) elements, respectively. For quadrilaterals/hexahedra, the designation $Q_m \times Q_n$ means that each component of the velocity is approximated by a continuous piecewise polynomial of degree $m$ in each direction on the element and likewise for pressure, except that the polynomial is of degree $n$. Again for the same families, $Q_m \times P_{-n}$ indicates the same velocity approximation with a pressure approximation that is a discontinuous complete piecewise polynomial of degree $n$ (not of degree $n$ in each direction ) (Donea and Huerta, 2003; Gresho and Sani, 2000). Stable elements are typically characterised by $m > n$.

For all methods, the discretisation process results in a linear system of equations with its size being the number of unknowns, i.e., a multiple of the number of nodes/elements. This system of equations is written as

$$\boldsymbol{A} \cdot \boldsymbol{X} = \boldsymbol{b} \tag{18}$$

where $\boldsymbol{A}$ is a large and very **sparse matrix**, $\boldsymbol{X}$ is the vector of unknowns, typically consisting of velocity components, pressure, and temperature on the grid, and $\boldsymbol{b}$ is the known right-hand side vector. In each time step, the system is solved, new solution
fields are obtained, post-processed, and analysed. If the model evolves in time, the domain and therefore the mesh may evolve, compositions are transported and a new system is formed.

### 3.3 Kinematical description

After the discretisation step, a kinematical description must be chosen to define how the material is going to move in the model. There are several widely-used options, i.e., the Eulerian, Lagrangian and Arbitrary Lagrangian-Eulerian (ALE) formulations
(Figure 5).

  Eulerian codes have a fixed mesh through which material flows. Since the evolution of the top boundary of the model is often of prime importance in geodynamical studies as it accounts for the generated topography, a feature that is directly and easily observable on Earth, the air above the crust must be modelled as well to allow for the formation of topography. This air layer has been coined 'sticky air' (Crameri et al., 2012) because of its much higher viscosity ($\sim 10^{18}$ Pa s) than real air
($\sim 10^{-5}$ Pa s). This higher viscosity is necessary to avoid numerically difficult viscosity contrasts in the domain of more than 20 orders of magnitude. Although the sticky air viscosity seems high, it remains very low compared to the effective viscosity of the crust ($\sim 10^{24}$ Pa s) bordering the sticky air, so that negligible shear forces are exerted on the crust surface by the air layer. Hence, topography can be developed. Finite difference codes typically use the Eulerian kinematical description and use the sticky air approach. Problems associated with the use of sticky air and a possible solution are presented in Duretz et al.
(2016).

  In contrast to a Eulerian kinematical description, the mesh of Lagrangian codes deforms with the computed flow and therefore does not require the use of sticky air to model topography. This limits Lagrangian codes to small deformation. For example, subduction processes would quickly deform the mesh to such a point that it would not be suitable for accurate calculations. PyLith (Aagaard et al., 2013), a finite element code for dynamic and quasistatic simulations of crustal deformation, uses a
690 Lagrangian kinematical description.

  Finally, as its name implies, the Arbitrary Lagrangian-Eulerian (ALE) method, part of the semi-Lagrangian class of methods, is a kinematical description that combines features of both the Lagrangian and Eulerian formulations. In geodynamical codes, it often amounts to the mesh conforming vertically or radially to the free surface, while retaining its structure in the horizontal or tangential direction (Figure 5). This approach forms the basis of the SOPALE (Fullsack, 1995), FANTOM (Thieulot, 2011),
pTatin3D (May et al., 2015), and ASPECT (Rose et al., 2017) codes.

### 3.4 Solving the linear system

The discretisation process outlined in section 3.2 leads to a linear system as given in Eq. (18). There are many ways to solve these large linear systems of equations, which can be split into two families: **direct methods** and **iterative methods**. **Direct methods** exploit the sparsity of the matrix (i.e., the ratio of non-zero terms to the number of terms in the matrix is very small) and arrive at the desired solution within a predictable number of operations. Because of the memory requirements of the direct solvers to store and process the matrix terms, these tend to be used for 2-D applications (or low resolution 3-D problems) only. Examples of direct solver packages frequently used in geodynamics (e.g. Braun et al., 2008; Quinquis et al., 2011; Thieulot, 2011) are MUMPS (Amestoy et al., 2019), Pardiso (Petra et al., 2014) and WSMP (Gupta et al., May 1997).

**Iterative methods** start with an initial solution (a guess) and incrementally improve on it until a convergence criterion is met, i.e. the remaining error (residual) is small. Common iterative methods in geodynamic codes are the Conjugate Gradient (CG) method (Shewchuk et al., 1994) and the GMRES (Generalized Minimal RESidual) method (Saad, 2003; Langer and Neumüller, 2018), which are used in conjunction with multigrid methods to accelerate their convergence. Note that the choice of iterative solving method is intrinsically tied to the properties of the matrix $A$. The most popular iterative solvers packages are PETSc (Balay et al., 1997) (used in pTatin3D, May et al., 2015) and Trilinos (Heroux et al., 2005) (used in ASPECT, Kronbichler et al., 2012). Recently, iterative pseudo-transient solvers have been used to solve coupled sets of equations (Räss et al., 2019; Reuber et al., 2020b). These methods introduce a physics-based transient-term (a time derivative) to a steady-state equation, in order to iterate towards the steady-state solution. The matrix-free, finite difference pseudo-transient schemes of Räss et al. (2019) are well-suited for GPU (Graphical Processing Unit) accelerated systems (Räss et al., 2020).

### 3.5 Computer architectures & parallelisation

Early computing architectures of the 1970s were quite limited by today's standards and predominantly relied on sequential programming where one task is performed after the other (Figure 6). Hence, early models in the 1970s were confined to a few thousands of unknowns at most (Stephansson and Berner, 1971; Beaumont and Lambert, 1972). The computer architectures and processor speeds on which calculations are performed have since vastly increased in the past decades following Moore's law (Eijkhout, 2013) resulting in models with several hundreds of millions of unknowns now possible (Alisic et al., 2012). Accommodating such enormous computational loads was only made possible by the use of parallel architectures (Figure 6) and their growing availability. Such architectures can be multi-core **processor**-based commodity-grade desktop computers, home-made (so-called Beowulf clusters) or super calculators counting up to a few millions of processors as listed in the TOP 500. The majority of the state-of-the-art geodynamics codes run in parallel on distributed memory systems and rely on the **MPI** (Message Passing Interface) standard (Figure 6). In MPI, domain decomposition is used, i.e. each computer thread of tasks only 'knows' about a subset of the mesh for which the solution is obtained. This allows for a virtually unlimited number of threads but it requires that the whole code is built with parallelism in mind to achieve maximum efficiency (Saad, 2003).

Other codes, such as the surface processes code FastScape (Braun and Willett, 2013) or the geodynamics code I2/I3ELVIS (Gerya and Yuen, 2007), rely on a different approach based on **OpenMP** which is limited to running on **shared memory**

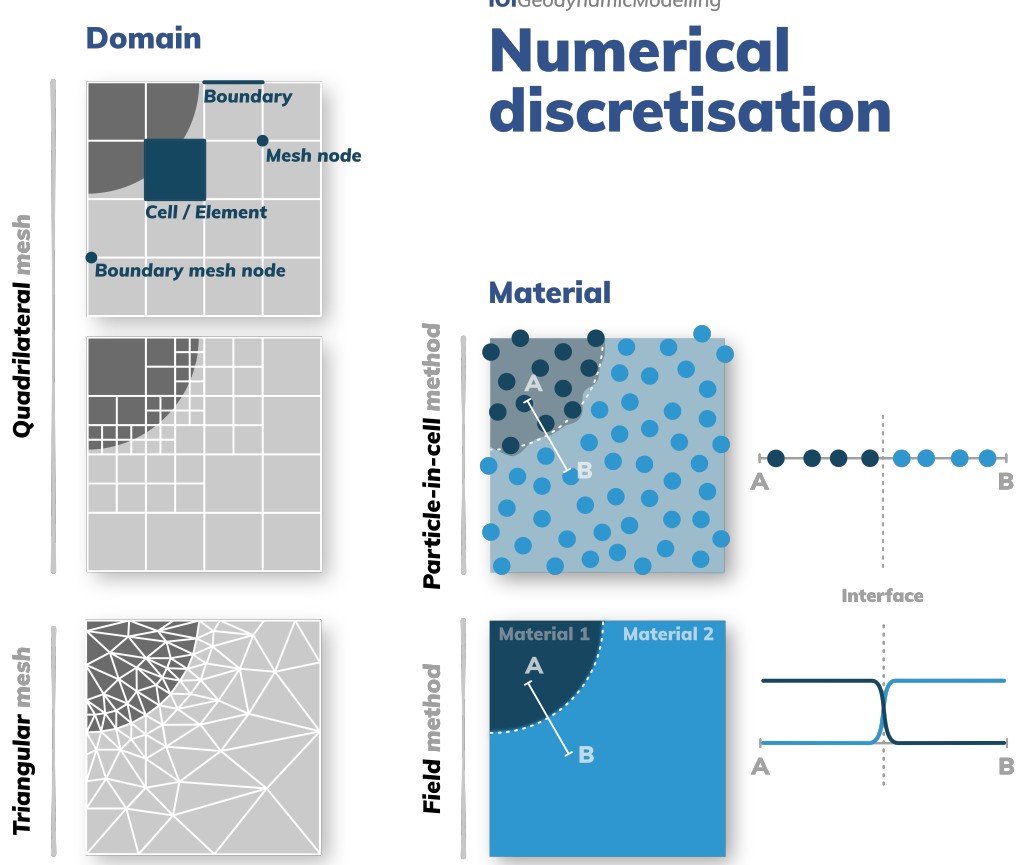

**Figure 4.** Examples of two-dimensional domain and material discretisation. The domain discretisation in the left-hand side column illustrates different types of meshes. The top left mesh is built on a **quadtree** and also shown with several levels of mesh refinement (middle left) so as to better capture the circular interface. The bottom left panel shows an unstructured triangular mesh built so that element edges are aligned with the (quarter) circle perimeter. Note that non-rectangle quadrilateral elements can also be used to conform to an interface. The material discretisation is illustrated by different methods of material tracking in the right-hand side column based on either the **particle-in-cell** method (top right) or grid-based advection (bottom right) for the material contrasts indicated by the blueish colours.

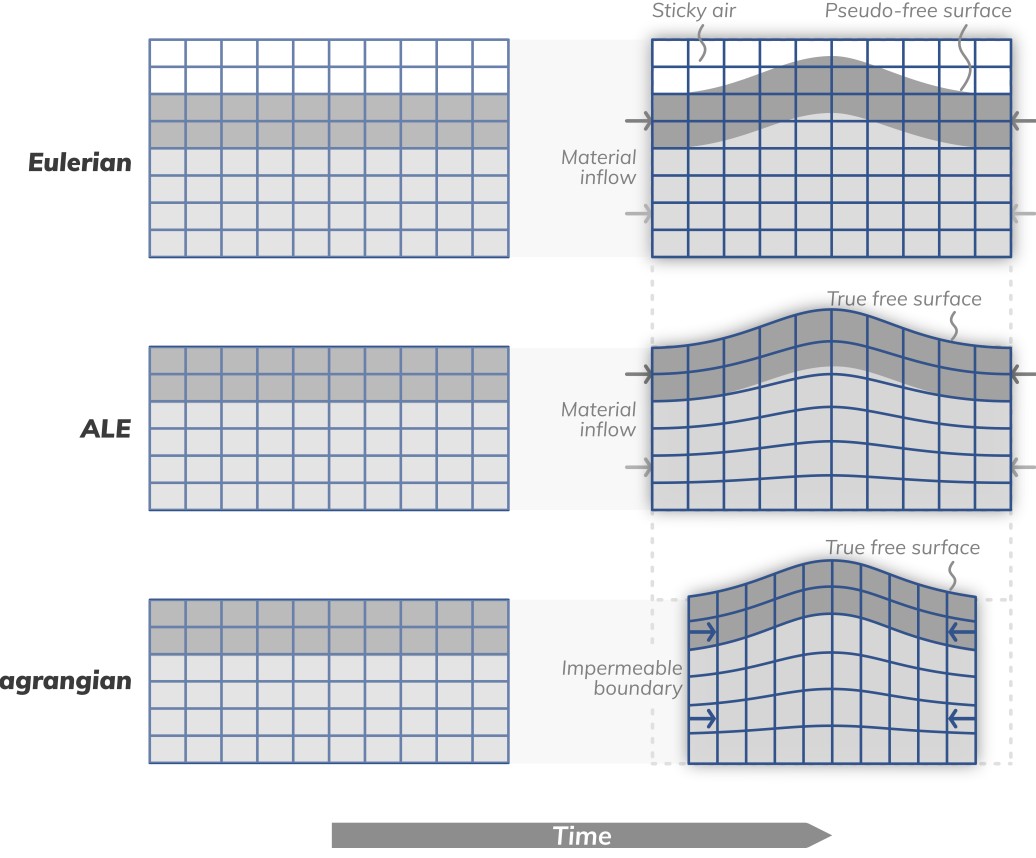

**Figure 5.** Kinematical descriptions for a compressed upper-mantle model setup. The left column shows the undeformed, initial model setups and the right column shows the deformed model after a certain amount of model time has passed. In the Eulerian kinematical description the computational mesh is fixed and the generated positive topography is accommodated by implementing a layer of sticky air above the crust. When an Arbitrary Lagrangian-Eulerian approach is used, the domain width is often kept constant in geodynamic applications, such that the mesh only deforms vertically to accommodate the topography. In the Lagrangian formulation, the mesh deforms with the velocity computed on its nodes.

# Computing

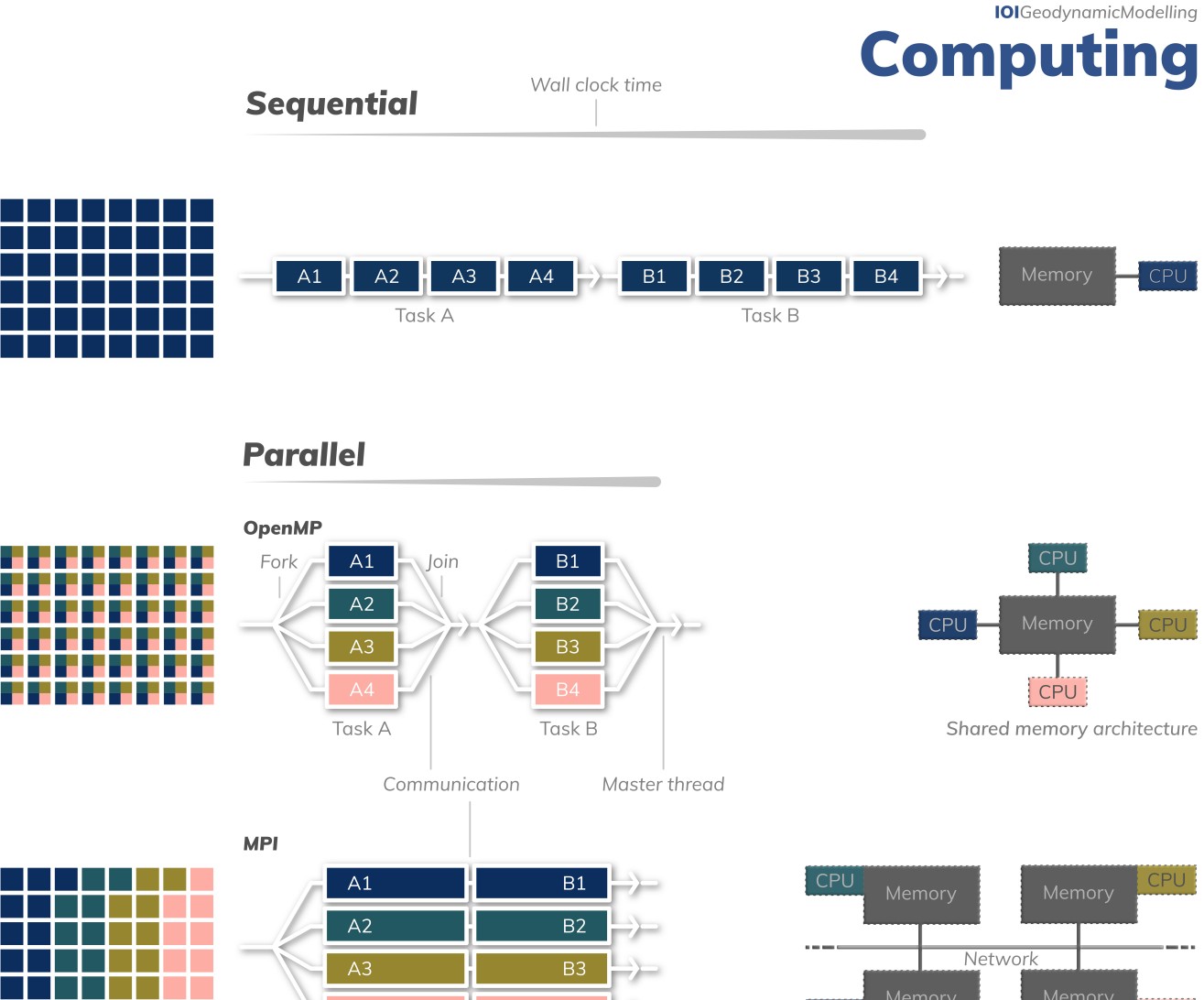

**Figure 6.** Computation paradigms. Top: Sequential programming with the discretised domain shown on the left. The code performs two tasks A and B in a sequential manner, on a single thread which has access to all of the computer's memory. Middle: The same code executed in parallel relying on OpenMP. Each **processor** of the computer concurrently carries out a part of tasks A and B so that the compute wall clock time is smaller. Bottom: If relying on MPI-based parallelisation the domain is usually broken up so that each thread 'knows' only a part of the domain. Tasks A and B are also executed in parallel by all the CPUs but now there is a distributed architecture of processors and memory interlinked by a dedicated network.

**systems**. OpenMP can be added later on to an already existing sequential code and targets areas of the code which take the most time. Although appealing at first, this level of parallelism is limited by the number of CPUs attached to the single memory (typically a few dozen at the maximum). It is worth noticing that many codes or the linear algebra libraries that they link to use a combination of both MPI and OpenMP.

When documenting the parallel performance of a code, one often talks of strong and weak scaling (Kronbichler et al., 2012; May et al., 2015; Kaus et al., 2016). **Strong scaling** is defined as how the solution time varies with the number of processors for a fixed problem size. Optimal strong scaling is achieved if the solution time is inversely proportional to the number of processors. Conversely, when looking at **weak scaling**, both the number of processors and the problem size are increased by the same factor. This also results in a constant workload per processor and one therefore expects a constant solution time for optimal weak scaling. In practice, all geodynamic modelling software have different limits for their strong and weak scaling, thereby limiting the size of the model they can effectively compute.

## 3.6 Dealing with nonlinearities

As mentioned in Section 2.3, the set of partial differential equations to be solved is likely to be **nonlinear**, i.e. the coefficients of the equations depend on the solution. For example, the viscosity may depend on the velocity via the strain rate tensor, which makes the term of the momentum equation that is the product of viscosity and strain rate a nonlinear term. Special techniques must then be employed to solve this system and so-called nonlinear iterations are carried out on the linearised equations until a converged solution is obtained. Note that these nonlinear iterations are distinct from the iterations taking place in the iterative method employed to solve the system. One type of nonlinear iterations are the **fixed-point ('Picard') iterations**, where a guess of the solution is made and it is used to compute the solution-dependent coefficients. The linear system is solved, and the coefficients are updated with the new solution. This process is repeated until the left-hand side and the right-hand side of the linear system match up to a given tolerance. Then, the iterative process has reached convergence. This technique is sub-optimal in its convergence rate as it often requires dozens or hundreds of iterations for strongly nonlinear problems, i.e., the linear system must be solved as many times (Spiegelman et al., 2016; Glerum et al., 2018).

State-of-the-art codes now all rely on some form of **Newton iterations** based on Newton-Raphson's method, which guarantee quadratic convergence towards the solution in contrast to the linear convergence of the Picard iteration, provided the initial guess is close enough to the solution of the nonlinear problem (May et al., 2015; Kaus et al., 2016; Rudi et al., 2015; Spiegelman et al., 2016). It is worth noting that the implementation of Newton's method is substantially more complex and therefore difficult than Picard iterations (Fraters et al., 2019a).

## 3.7 Tracking materials

Aside from the conservation of mass, momentum, and energy, special care must be taken when solving the advection equation (15), which accounts for the tracking of the different fluids, chemical species or materials in a broad sense. Many techniques have been devised in the field of computational fluid dynamics and are used in geodynamic codes. The methods differ in whether they are designed to treat smoothly-varying fields such as temperature, or discontinuous fields such as those that

represent lithologies. Many methods have been developed to advect smooth fields while minimising artefacts such as numerical diffusion and dispersion (i.e., SUPG (Streamline Upwind/Petrov-Galerkin), flux limiters, TVD (Total Variation Diminishing), FCT (Flux-Corrected Transport), and MPDATA (Multidimensional Positive Definite Advection Transport Algorithm) (Gerya, 2019; Ismail-Zadeh and Tackley, 2010)). Other employed techniques for tracking discontinuous fields are the **level-set method** (Samuel and Evonuk, 2010; Hillebrand et al., 2014; Braun et al., 2008), grid-based advection methods (sometimes referred to as **compositional fields methods**, Figure 4, Gerya (2019)), the **marker-chain method** (van Keken et al., 1997), and the **volume-of-fluid method** (Robey and Puckett, 2019; Louis-Napoléon et al., 2020). A popular advection method, especially for discontinuities, is the **particle-in-cell** (sometimes called marker-in-cell) technique (Figure 4), where a set of markers is placed throughout the domain at the beginning of the simulation with an assigned material identity, e.g., crust, mantle lithosphere, or sediments. These properties are then projected onto the mesh on which they are needed for solving the partial differential equations. Once a new velocity field has been computed, these markers are advected with the flow velocity field, and the process is repeated at each timestep. This method is found in finite element (Tackley and King, 2003; Deubelbeiss and Kaus, 2008; Thielmann et al., 2014; Gassmöller et al., 2018, 2019) and finite difference codes alike (Gerya and Yuen, 2007; Pusok et al., 2016).

It is important to mention that there is no single best recipe for advection, and oftentimes methods are tested against each other (van Keken et al., 1997; Tackley and King, 2003). Each method comes with its own strengths and weaknesses and needs to be assessed in the context of the problem at hand. For example, particle-based methods are relatively easy to implement, not subject to numerical diffusion (a form of undesirable smearing of sharp gradients), and can also be used to track properties like deformation history, water content, composition, and phase transitions. However, they take up considerable memory, introduce artificial noise, and may violate conservation laws of the advected quantities when averaged to the grid. On the other hand, field-based methods are straightforward to parallelise and much lighter in terms of memory usage, but more challenging to advect and require numerical stabilisation. As such, choosing a field-based method for tracking a single smooth property is generally warranted, while particle methods serve well for tracking multiple properties that vary throughout the model domain.

## 3.8 Multiphysics

On Earth, the lithosphere interacts with the cryosphere, biosphere, atmosphere, hydrosphere, magnetosphere, and other systems, and the deformation of the lithosphere is related to many natural hazards such as earthquakes, volcanic eruptions, landslides and tsunamis. These systems are often inherently multi-scale with different processes occurring on vastly different time and/or length scales (Figure 1) and/or multi-physics where the coupling of different physical processes is important. Such multiphysics or multidisciplinary research often sees researchers coupling a geodynamics code with another existing one, be it for surface processes (Braun and Yamato, 2010; Thieulot et al., 2014; Collignon et al., 2014; Ueda et al., 2015; Beucher and Huismans, 2020), atmosphere evolution (Gillmann and Tackley, 2014), planetary impact (Golabek et al., 2018), plate reconstruction modelling (Bull et al., 2014; Gassmöller et al., 2016), elastic flexure (Naliboff et al., 2009), or dynamic rupture models and tsunami models (van Zelst et al., 2019; Madden et al., 2020). This coupling often takes place in the boundary conditions of the geodynamics code at the surface, or at the bottom of the lithosphere or mantle. This coupling can be one-sided when, e.g.,

plate velocities are prescribed. The coupling can also be two-sided when, e.g., the uplift rate and a current topography of a geodynamic model are used in a surface process model which returns an eroded surface which forms the new top boundary of a geodynamic model. Coupling codes and the clever use of boundary conditions (Section 5.3.1) are promising avenues to incorporate geodynamic models in a host of multidisciplinary research.

However, some multi-physics problems are so closely intertwined that solving the coupled system of equations requires different numerical methods than solving the problems individually (for example, coupled magma/mantle dynamics). In these cases, coupling requires the development of new code that tightly integrates the different physical processes. One recent approach in building numerical applications for multi-physics problems that could be used for this makes use of Application Programming Interfaces (APIs) instead of readily-available community codes. APIs can be a collection of routines that are

optimised to perform certain operations such as assembling vectors and matrices, solving systems of equations in parallel (i.e., PETSc - Balay et al. (2021), ParallelStencil for Julia), or create and solve geodynamic-oriented objects such as tensors or even partial differential equations systems (i.e., FEniCS, MILAMIN - Dabrowski et al. (2008), Räss et al. (2017); Pusok et al. (2020)). These APIs are not necessarily stand alone codes, but are instead building blocks that require the user to set up their model (i.e., program their code application using available routines). Therefore, they provide the user with more flexibility in

creating and coupling models with different physics, and can be more robustly tested as the APIs usually come with their own suite of tests. This approach also allows scientists to build simple, experimental numerical applications, which can then be scaled to large, parallelised production codes in a relatively short time span.

## 4    Code verification

At this point, we have described the model in terms of governing and constitutive equations, and discretised and solved the

system using appropriate numerical methods. Hence, an application code has been obtained at this point. However, before any scientific study can be performed with confidence, the code must be tested to ensure it does what it is intended to do. This process has two components: **verification** and **validation** (Section 6). Verification means 'solving the equations right', while validation means 'solving the right equations' (Roache, 1997). In this section, we will briefly focus on code verification, which involves using **tests** and **benchmarks**.

In the software engineering community, the importance of tests is well acknowledged: 'Without tests every change is a possible bug' (Martin, 2008). While there are only a few equations to solve in geodynamics (Section 2; Lynch (2005)), the coefficients and the constitutive equations (i.e., rock rheology, multi-component processes) to describe the physical system often turn them into complex nonlinear problems, which can be difficult to test. Tests should verify specific parts or the whole functionality of the code. It is also desirable to test as many features of the code as possible. Of course, one can think of an

infinite number of tests for even a simple geodynamics application, but because code verification can be time-consuming, a balance has to be found between test coverage and production. Without a doubt, verification is an absolutely necessary stage of code development as its purpose is to test the robustness of the code in a broad range of situations and applications.

The recommended approach is to implement only what is needed, test it, and then refactor and expand the system to implement new features later on, and test again. Implementing an automatic testing framework can speed up these steps. Good tests also follow the FIRST acronym: Fast (run quickly), Independent (do not depend on each other), Repeatable (in any environment), and Self-validating (have a boolean output, pass or fail) Tests (Martin, 2008). Without proper testing, model results may look beautiful, but are probably wrong; and having only simple tests may not catch bugs in complex geodynamic applications (Gerya, 2019). Moreover, just inspecting the output and saying 'it looks right' is not sufficient. Instead, checking error norms and expected convergence rates or other numerical diagnostics is more robust. Tests should verify the code on a broad range of challenging cases relevant to the scientific question. For example, in lithosphere dynamics modelling, the code should correctly handle time-dependent problems with a variable viscosity that depends on temperature and strain rate, as well as complex geometries with a free surface, and the transport of chemical heterogeneities.

The types of tests one can do to benchmark codes relate to **numerical implementation** and functionality. In general, this means comparing the numerical solution obtained from solving the system of equations to analytical solutions, results of analogue experiments, numerical results from other codes and general physical considerations. However, one can also make smaller **unit tests** that verify correct functionality of individual routines, i.e., not necessarily routines that solve for physics, and make sure they do not change when modifications are done somewhere else in the code. For example, parallelisation is not a physical process but poses new problems that can be tested with unit testing. In a parallel code, one can verify that the parallel communication routines work correctly with a simple **hello world** test. Recently, unit tests can be automatically incorporated in the development process through platforms like GitHub, Gitlab, BitBucket, and Travis CI.

**Analytical solutions** are available for systems of equations that are well defined (e.g. steady-state diffusion, linear single-flow models) (Turcotte and Schubert, 2014). Using analytical solutions for testing is powerful because it verifies the whole code functionality, including boundary conditions and the solutions often have a physical meaning. However, the disadvantage is that they are mostly restricted to simple models with linear coefficients. Analytical solutions for problems relevant to solid-earth physics include the onset of mantle convection, topography relaxation for free surface, the corner flow (applied to mantle wedges), viscous folding (Schmalholz and Podladchikov, 1999), half-space cooling, the bending of an elastic beam, and the indentation into a visco-plastic material. More examples can be found in Turcotte and Schubert (2014), Gerya (2019) and references therein, and numerical benchmarks for some analytical solutions can be found in the ASPECT manual (Bangerth et al., 2019).

Analytical solutions for code verification can also be used in the form of **method of manufactured solutions** (MMS) (Roache, 1997, 2002). In principle, the user manufactures an exact solution or coefficients, without being concerned about their physical meaning. The method of manufactured solutions primarily verifies correct implementation and the convergence order of a particular numerical method, i.e., the accuracy of the solution with increasing spatial and temporal resolution. Even if the method does not focus on the physical meaning of the solution, some physical considerations still need to be taken into account when using manufactured solutions, e.g., coefficients corresponding to physical parameters need to be positive. The method of manufactured solutions is a powerful technique as it can be applied to arbitrarily complex partial differential equations (e.g., linear, nonlinear, time dependent) and to arbitrarily complex domains and boundary conditions. Recent computational advances

make this method easy to use by creating manufactured solutions using symbolic algebra packages such as Maple, MATLAB, Mathematica or SymPy. The method of manufactured solutions is not as commonly used in geodynamics as in engineering or applied mathematics, but it provides potential for more robust testing frameworks, including nonlinear problems (May et al., 2013; Donea and Huerta, 2003; Burstedde et al., 2013; Kramer et al., 2021).

When analytical solutions are not possible, numerical experiments of the same model setup (i.e., equations, boundary conditions, geometry, and parameters) can be tested with a number of different codes within the community. These are called **community benchmarks** and allow for testing complex problems by comparing similar diagnostics (e.g., non-dimensionless parameters such as the Nusselt number; Section 7.2) across codes. The results are then compiled, and the best average behaviour among codes is taken as the benchmark for the test. Examples of community benchmarks include thermo-mechanical convection (Blankenbach et al., 1989; Travis et al., 1990; Busse et al., 1994; Tackley and King, 2003; Zhong et al., 2008; King et al., 2010; Arrial et al., 2014; Tosi et al., 2015), subduction (Schmeling et al., 2008; van Keken et al., 2008), Rayleigh-Taylor instabilities (van Keken, 1997; Louis-Napoléon et al., 2020), and motion of the free surface (Crameri et al., 2012). It is important to note that community benchmarks are not bulletproof against bugs and do not necessarily provide insights into the numerical or physical behaviour of the system. Moreover, successfully developing and executing a community benchmark is a long process that typically takes years.

Comparison with analogue experiments is important for calibrating numerical models for complex processes, required for numerical modelling of large-scale tectonic processes (e.g., plastic failure). They can be used for both verification and validation of numerical models. For example, modelling sandbox experiments (Buck and Sokoutis, 1994; Buiter et al., 2006, 2016) poses significant computational challenges because the numerical code must be able to calculate large strains along narrow shear zones. Other complex processes in which analogue experiments can be insightful include frictional and free surface boundaries, complex rheology involving both viscous and frictional/plastic materials, or reactive processes. Numerical studies comparing to analogue experiments include gelatine wedge seismic experiments (van Dinther et al., 2013a), plume dynamics (Davaille, 1999; Davaille et al., 2011), indenter block experiment (Tapponnier et al., 1982; Peltzer and Tapponnier, 1988), and subduction dynamics (Schellart, 2005; Duarte et al., 2013; Király et al., 2020b). Since there are fundamental differences between numerical and laboratory experiments, the model results are often not identical, and instead certain characteristic features of the solutions need to be compared.

As the importance of testing is revealed, more software engineering practices are required to keep codes clean, testable, and robust (see Section 9). Designing a suite of complete tests is just as important as building efficient, fast, complex, high-resolution numerical 3-D codes. There are many excellent books on analytical solutions (Turcotte and Schubert, 2014), keeping codes clean (Martin, 2008; Hunt and Thomas, 1999) and building robust testing frameworks for geodynamics (Ismail-Zadeh and Tackley, 2010; Simpson, 2017; Morra, 2018). As a final note on code verification, complex systems of equations (i.e., multi-phase, multi-component, nonlinear rheology) can still be badly posed and are difficult to test. For this reason, verification of simpler systems is important, and complex solutions should be validated using **scaling analysis** and against natural observations (see Section 6).

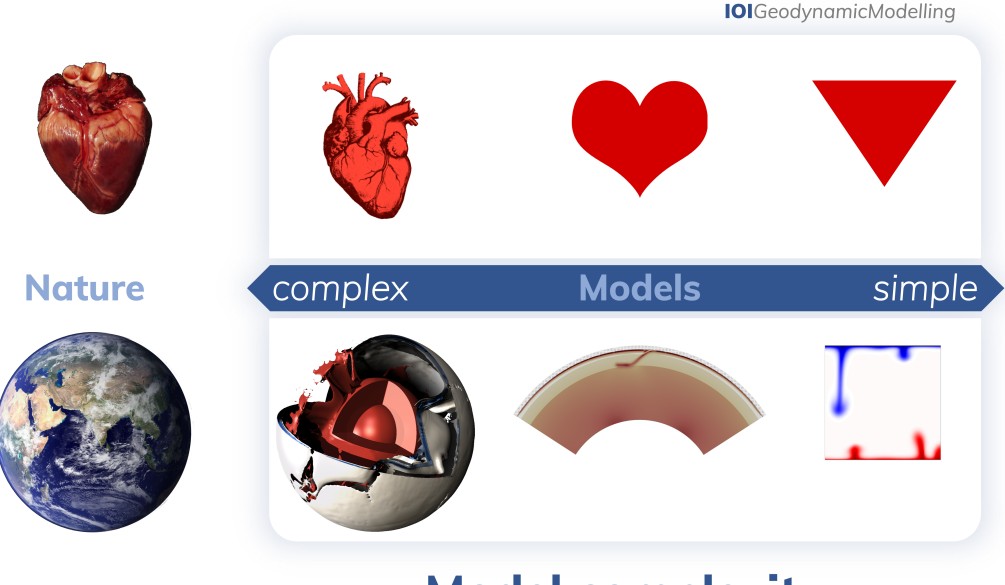

**Model complexity**

**Figure 7.** Different model complexities for the heart (top) and the Earth (bottom). A simpler model can be more useful: the basic shape of the heart has likely become the most successful model, indeed a true icon, only because it was neither too complex (it can be reproduced easily), nor too simple (its characteristic shape is still recognisable). Finding the right level of complexity is challenging and must repeatedly be considered carefully by the modeller for each new modelling task at hand.

## 5   From the real world to a model setup

Designing a model is not straightforward. Before starting to design a model, it is important to understand the code, the model, and the difference between the two. While the code's purpose is of a general nature (e.g., to allow for creating models to investigate some geodynamic problems), the purpose of the model is very specific and, in most cases, indeed unique. This unique purpose is reflected in the complex nature of the model, which has to be set up with care. A model is the sum of an underlying (modelling) philosophy, one or more geodynamic concepts and hypotheses, its physical and numerical construct, and initial and boundary conditions. Even though the purpose of a geodynamic model is usually unique, its outcome never is. The same result of a spatially or temporally restricted model of nature can always be recovered by multiple different models. Therefore, a geodynamic model cannot be verified, in contrast to the code.

How to design a simplified — but not oversimplified — geodynamic model that is based on a certain modelling philosophy and applies suitable initial and boundary conditions, is therefore outlined in this section.

## 5.1 Simplifying without oversimplifying

A model is, by definition, a simplified representation of a more complex natural phenomenon. This is a simple and obvious truth that is easily forgotten when geodynamic models are interpreted, presented, and reviewed. It is the modeller's responsibility to not only constantly remind themselves, but also others about this key underlying fact.

The complexity of the planet Earth as a whole is vast. It is therefore challenging to reconcile such true complexity with a desired simplicity. A model can easily become too complex and, just as easily, oversimplified (Figure 7). A geodynamic modeller should always strive to meet the appropriate middle ground in between a too complex and too simple model. This is one of the first major tasks of a modeller when setting up a new modelling approach to answer a certain hypothesis: everything should be made as simple as possible, but not simpler.

So, how simple should a model optimally be? The answer to this question is not an easy one, as it strongly depends on the purpose of the model, the capabilities to diagnose and understand it, and the hypothesis that it will test. It is clear though that a more complex model does not necessarily mean a better model. In fact, a simpler model is often better than a more complex model. A simpler model is clearer to understand, clearer to communicate, and, by making fewer assumptions, more likely right (Occam's razor; Thorburn, 1915), although not necessarily directly applicable to the real world.

There are various ways to reduce model complexity (Figure 8). Simplifying the physical model is one of them. Any given physical complexity of the natural phenomenon in question has to be evaluated and a decision has to be made to either reproduce it, parameterise it (i.e., mimic the phenomenon with a simplified approach), or neglect it in the model. This decision is based upon the model's purpose and the spatial and temporal extent of the process under investigation.

Further model simplification is achieved through numerical adjustments. For example, all the following studies model plate tectonics, but the geometry of the model can be complex (e.g., a 3-D spherical domain like Arnould et al. (2020)) or simple (e.g., a 1-D or 2-D domain like Bercovici and Ricard (2014)). One can choose the temporal extent of the model to be more complex (e.g., time-dependent with a long evolution as in Crameri et al. (2012)) or simply instantaneous as done by Stadler et al. (2010). For the same model, such simplifying choices make for a generally simpler model. However, a model with simpler geometry (or time evolution) can, of course, also feature more complicated processes than one with a more complex geometry. Indeed, a simpler model geometry (or time evolution) often enables the modelling of more complex physical processes.

The numerical model complexity can also be adjusted by changing the initial and boundary conditions (heterogeneous as in Conrad and Gurnis (2003) or homogeneous as in Foley and Becker (2009)) and the imposed forcing (space and time dependent forcing like Quere and Forte (2006) or self-consistent like Rolf and Tackley (2011)). Overall, complexity should be decided based on the scientific question addressed and the focus and scope of the study.

## 5.2 Modelling philosophies

There are two overarching geodynamic modelling philosophies: **specific modelling** and **generic modelling** (Figure 9). The first philosophy attempts to reproduce the specific state of a certain geodynamic system (e.g., based on a specific observation) with a specific model to better understand the system's specific state. In contrast, generic modelling attempts to produce different

| complex | Model | simple |
|---|---|---|
| **Physical complexities** Reproduce | *Parameterise | Neglect |
| **Multiphysics** Coupling | One-way coupling | None |
| **Constitutive equations** Non-linear | Linearised | Constant |
| **Domain geometry** 3-D Spherical Wide | 2-D | 1-D Cartesian Narrow |
| **Model duration** Time-dependent | | Instantaneous |
| **Initial conditions** Heterogeneous | | Homogeneous |
| **Boundary conditions** Free | | Fixed |
| **External forcing** Self-consistent | | Imposed |

*Mimic a physical complexity with a simplified approach

**Simplifying**

**IOI**GeodynamicModelling

**Figure 8.** Potential options for geodynamic model simplification. Note that we mean 'multiphysics' beyond the already coupled system described in Section 2 (see Section 3.8).

regimes of behaviour of a certain geodynamic system (e.g., based on a general observation) to better understand the system's general behaviour.

Both overarching modelling philosophies can either fulfil or reject a hypothesis. Most results published to date fulfil a hypothesis, even though positive modelling results only hint at a certain phenomenon being responsible for an observation. Modelling results that reject a hypothesis (often called 'failed models') are of course more abundant, but, also, much clearer as they indeed serve as proof that a certain situation does not lead to a specific observation.

Furthermore, both overarching modelling philosophies can result in instantaneous and time-dependent studies (Tackley et al., 2005). Instantaneous models are focused on resolving a certain state and usually rely heavily on a comparison with measured geophysical, geological, and geochemical data. Time-dependent models tend to focus more on the evolution or the natural state of a system. For general modelling, time-dependent studies can be performed either focusing on the transient evolution from a known initial state or the statistical (quasi-)steady state.

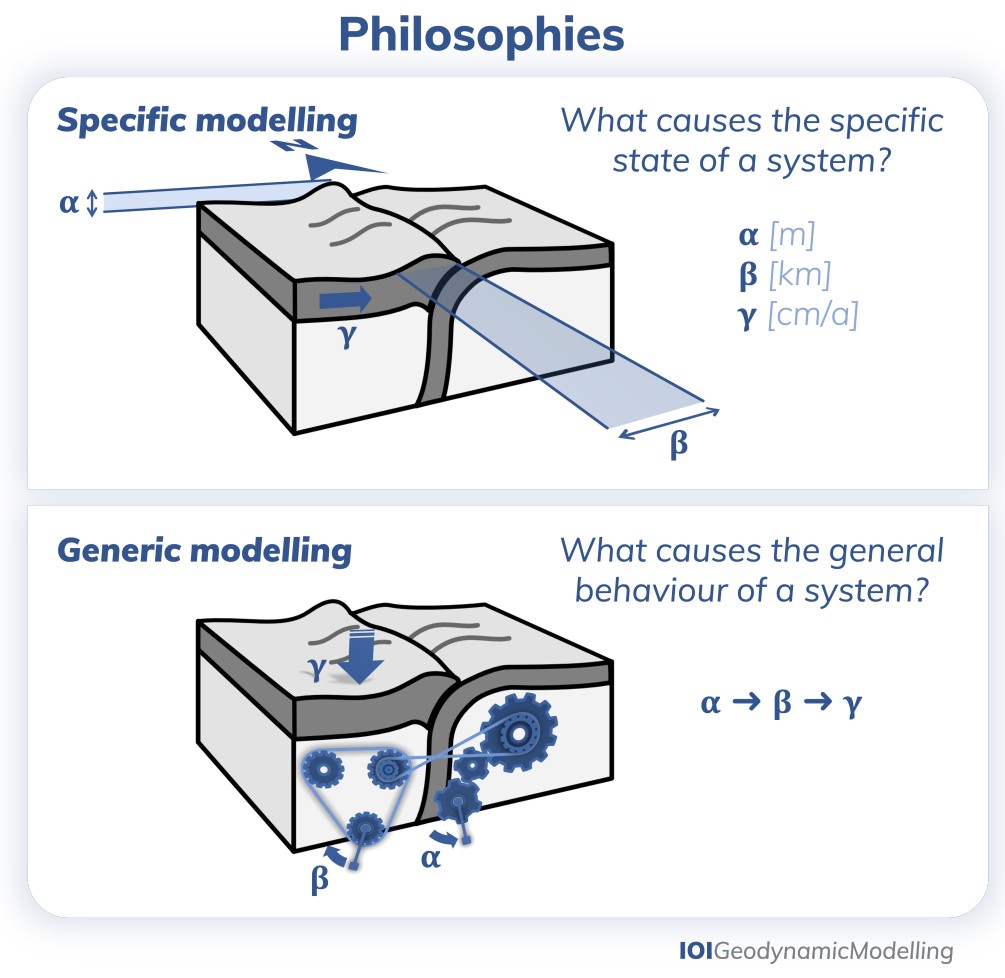

**Figure 9.** The two overarching modelling philosophies. (a) Specific modelling and (b) generic modelling have different scientific goals and need to be used, communicated, and reviewed differently.

### 5.2.1 Specific modelling procedure

Modelling aimed to compare and understand a specific state of a geodynamic system necessitates the following procedure. Firstly, a specific observation (in a certain region) has to be defined. Secondly, a hypothesis about the control mechanism(s) has to be outlined. Thirdly, a model setup needs to be designed considering three key aspects. The model needs to be able to produce the observed feature, include the hypothetical control mechanism(s), and physically link the control mechanism to the observed feature. Lastly, the model has to be simplified to be easily understandable without being oversimplified. For specific modelling in particular, the modeller needs to keep in mind that there is no guarantee that the suspected control mechanism is the actual, nor the only, controlling mechanism (see Section 1.2).

A specific modelling philosophy is often used to understand the circumstances that facilitated natural hazard events, like earthquakes, in order to improve hazard analyses. For example, Smith and Sandwell (2006) optimise the fit between geodetic velocities of the San Andreas Fault and their model predictions of the seismic history of the fault. From the modelled present-day stress state, regions of high seismic hazard are then inferred. Another example are investigations into the specific surface topographic history of a certain subduction zone. These investigations first build the best informed model of a subduction zone (e.g., the Cocos subduction zone in Southwestern Mexico, or the Southern Alaskan subduction zone in North America) and then use it to reproduce and study how this subduction zone impacts the topographic evolution of the adjacent topographic heights (e.g., the Trans-Mexican Volcanic Belt) as done in Gérault et al. (2015) or intra-continental shear zones (e.g., the Denali fault) as done in Haynie and Jadamec (2017). A global specific modelling example is the global lithosphere-asthenosphere modelling study of Osei Tutu et al. (2018), which obtains plate boundary friction coefficients and the minimum asthenospheric viscosity cutoff value by optimising the fit with observed global plate motions.

### 5.2.2 Generic modelling procedure

Modelling aimed at understanding the general behaviour of a geodynamic system necessitates the following procedure. First, a general, first-order observation has to be defined. Second, a hypothesis about the controlling parameters and their possible range has to be outlined. Third, a model setup needs to be designed considering two key aspects. The model needs to include the proposed control mechanism(s), and it needs to be built on a set of assumptions for simplification. For generic modelling in particular, the assumptions that go into designing the geodynamic model are key and need to be specified and described clearly (Section 8.1).

When a generic modelling philosophy is applied, a general geodynamic feature is investigated via a **parameter study**, where a certain **parameter space** is mapped out and can be represented by a so-called **regime diagram** (e.g., Lourenço et al., 2020; Citron et al., 2020; Gülcher et al., 2020; Neuharth et al., 2021; Pusok and Kaus, 2015). For example, surface elevation changes due to plate subduction can be investigated without specifically relating to one particular subduction zone, but to subduction zones in general. One way of doing that is to model subduction and its surface response, and vary all key subduction parameters over their individual Earth-like ranges as done in e.g., Crameri et al. (2017). Other generic modelling examples are quantifying crustal thickness at mid-ocean ridges for various spreading velocities (e.g., Katz, 2008), reproducing general magma dynamics

(e.g., McKenzie, 1984; Spiegelman, 1993; Sim et al., 2020) or magma transport behaviour (e.g., Yamato et al., 2012), the onset of convection in a planetary mantle (Turcotte and Schubert, 2014, section 6.19), testing for what Rayleigh numbers and Clapeyron slopes a phase transition induces layered convection (e.g., Christensen and Yuen, 1985), quantifying the amount of entrainment of a dense layer into mantle plumes (e.g., Lin and van Keken, 2006a, b; Jones et al., 2016), investigating the plate tectonic regimes of a planet, which might range from a stagnant lid with only one plate to a mobile lid, similar to modern plate

tectonics (e.g., Petersen et al., 2017; Lourenço et al., 2016; Lourenço et al., 2020), investigating under which conditions the flow in the Earth's outer core would cause an Earth-like dynamo (Christensen, 2011; Christensen and Wicht, 2015; Wicht and Sanchez, 2019), and mapping out the dominance of inner-core convection, rotation, or translation depending on its viscosity and conductivity (e.g., Deguen et al., 2013). Non-dimensional numbers often make up axes and boundaries in regime diagrams, although other diagnostic quantities can also be used (Section 7.2).

The mapping of a parameter space is often done through manual variation of a single model parameter and comparison of the resulting model predictions. However, recent developments allow for scaling laws between the model solution and the model parameters to be computed automatically through adjoint methods. Besides solving inverse problems (e.g. Ismail-Zadeh et al., 2003; Ghelichkhan and Bunge, 2016; Colli et al., 2018), adjoint methods can efficiently compute the scaling exponent for all model parameters with one linear solve (for a specific model time step) (Reuber et al., 2018c). These scaling exponents (that

are based on the derivative of the solution parameter to model parameter) indicate which parameters control the model solution and which have a lesser effect (e.g. Reuber et al., 2018a; Crawford et al., 2018). Knowledge of the relative importance of each model parameter can help decrease the parameter space that is to be investigated (see also Section 7.3).

### 5.3 Boundary and initial conditions

After choosing the equations that will be solved and the model geometry, both initial and boundary conditions are needed to

1005 solve the numerical model. The solution of the numerical model will depend on the used initial and boundary conditions, so it is important to choose them carefully (Quinquis et al., 2011; Chertova et al., 2012).

### 5.3.1 Boundary conditions

The **boundary conditions** describe (part of) the solution variables (e.g., velocity) at the boundaries of the model domain necessary to solve the system of equations. They can vary in time and space. Mathematically, there are five main types of

1010 boundary conditions. (1) **Dirichlet boundary conditions** (also called first-type or fixed) specify the value of the solution of an equation at the boundary. (2) **Neumann boundary conditions** (also called second-type) specify the value of the derivative of the solution. (3) **Robin boundary conditions** (also called third-type) are linear combinations of the values and the derivatives of the solution. (4) **Mixed boundary conditions** indicate that across one boundary Dirichlet, Neumann, and Robin boundary conditions are applied to specific parts of that boundary. For example, in lithosphere dynamics, there could be both a litho-

1015 spheric plate and the mantle at the vertical boundary of a Cartesian model. Mixed boundary conditions applied here could be a constant velocity (Dirichlet) applied to the lithospheric plate, while the mantle has an open boundary condition (Neumann). The last type of boundary conditions is (5) the **Cauchy boundary condition** where both the Dirichlet and Neumann boundary

condition are applied to a boundary simultaneously by specifying both the solution and its normal derivative. Note that this differs from Robin boundary conditions as there is no linear combination and the normal derivative is specifically prescribed for Cauchy boundary conditions.

For the thermo-mechanical models considered here, we need to prescribe boundary conditions for the conservation of mass, momentum, and energy equations in order to solve them. For the **Stokes equations**, typical mechanical boundary conditions include (1) the **free surface** (Neumann), where there is no shear (parallel to the boundary) or normal (perpendicular to the boundary) stress acting on the boundary, and which can therefore freely deform according to the velocity solution. It is most commonly used in models where topography is important. Specifying the stress is a Neumann boundary condition, because the relations between stress and velocity defined in the constitutive equations (Section 2.2.1, Eqs. (8) and (9)) imply that fixing the stress prescribes the velocity derivatives. (2) For the **free slip** boundary condition, the component of the velocity normal to the boundary is set to 0 (Dirichlet), and there are no stresses acting parallel to the boundary (Neumann). This results in material flowing freely along the boundary such as the core-mantle boundary for global convection models. Note that the free slip boundary condition is neither a Robin nor a Cauchy boundary condition even though it combines prescribing the solution and its derivative. (3) **Prescribed velocities** (Dirichlet), also called kinematic or in-/outflow boundary conditions, are often applied at the sides of plates in lithospheric models or at the top of mantle convection models to mimic plate tectonics (Bull et al., 2014). (4) The **no slip** boundary condition (Dirichlet) is a special case of prescribed velocities where the velocity is zero at the boundary. This is typically used to mimic the 660 discontinuity at the bottom boundary of asthenospheric-scale models. (5) **Prescribed stresses** are Neumann boundary conditions as the stresses relate to the velocities through the derivatives. They can be used to mimic plate push and topographic loads. (6) An **open boundary** (Neumann) is a special case of prescribed stresses, where material can freely leave the domain (Chertova et al., 2012). In addition, infinity-like or 'external' boundary conditions can be applied where a boundary is modelled as being far away (Gerya, 2019). This is typically applied in lithospheric-scale models, where an external free or no slip boundary is applied to the bottom boundary, mimicking a free or no slip boundary at a distance $\Delta L$ from the actual boundary. Similar to this, a Winkler boundary condition assumes an isostatic equilibrium at the bottom boundary, analogous to applying a free surface boundary condition at the bottom of a lithospheric- or crustal-scale model (Burov et al., 2001; Yamato et al., 2008). In combination with a free slip boundary condition, a layer of less dense ($\rho \approx 1 \, \text{kg/m}^3$) '**sticky air**' (or 'sticky water') material is often used to model topography in methods that use an undeformable mesh and can therefore not employ a free surface boundary condition (Section 3.3, Figure 4) (Schmeling et al., 2008; Crameri et al., 2012).

Another type of commonly used boundary conditions in geodynamics are the **periodic boundary conditions**. They are different in nature from the purely mathematical boundary conditions listed above, as they do not explicitly prescribe any part of the solution. Instead, they 'link' boundaries together to approximate a larger (or infinite) system of which the model setup is merely a part: any materials or flows passing through one boundary interface re-enter the model domain through the opposite boundary interface. This makes periodic boundary conditions the natural choice in global mantle convection models (Gurnis, 1988; Lowman et al., 2001). This technique is also widely used in lithosphere dynamics, where often smaller-scale model setups are used, but they are modelled as part of the wider mantle convection process through periodic boundary conditions.

For example, mantle flow leaving the domain at the right-hand side of the model setup will re-enter the model domain on the left-hand side of the model domain, thereby effectively creating a theoretical closed mantle convection cell. Another example is the use of periodic conditions on the boundaries normal to the rift trend in 3-D models of continental rifting, effectively creating an infinitely long rift of which only a segment is modelled.

Boundary conditions can be used to drive the system by, e.g., prescribing the velocities of plates resulting in lithospheric extension or convergence. Hence, the modeller could assimilate data on plate motions from the geologic record into the model through the boundary conditions to improve the predictive power of the model (Colli et al., 2015). Similarly, stress boundary conditions can be implemented to simulate a load on the model, such as an icesheet. The data assimilation into the models via boundary conditions is typically used for specific modelling studies (Section 5.2).

Considering boundary conditions for the energy equation for models of the Earth's mantle and crust, it is common practice to prescribe fixed temperatures (Dirichlet) at the Earth's surface and at the core-mantle boundary, and prescribed heat fluxes (Neumann) for boundaries within the mantle. Fixing the heat flux fixes the amount of energy within the domain. When using fixed temperatures, the amount of energy can freely evolve but the temperature variations are fixed. However, this might not always be applicable. For example, a model of a mantle plume in the upper mantle will need a prescribed inflowing plume temperature at the bottom of the model at the upper mantle. Similarly, although the outer core can be assumed to be a reservoir with a constant temperature for mantle models, for models of the outer core heat flux boundary conditions at the core-mantle boundary are more appropriate (Wicht and Sanchez, 2019). Coupled modelling methods could also result in different temperature boundary conditions (Section 3).

The boundary conditions of both the Stokes equations and the energy equations can be related in the model. For example, if the model has an open boundary, there could be both inflow and outflow along different parts of that open boundary. On the inflow part of the boundary, it is useful to prescribe the temperature (e.g., slab age), whereas on the outflow part of the boundary, insulating Neumann boundary conditions can be used.

It is also possible to constrain degrees of freedom inside the domain. For example, in lithospheric-scale models, a velocity prescribed within the slab (i.e., not at the boundary) is an example of an internal driving force to prescribe subduction in the absence of initial slab pull (e.g., van Dinther et al., 2013b).

The choice of boundary conditions can alter the modelling results with, e.g., different mechanical boundary conditions on the sidewalls of the mantle affecting the resulting subduction behaviour (Chertova et al., 2012; Quinquis et al., 2011). It is also important to choose boundary conditions consistent with the rest of the model (Section 6.2). Hence, the modeller should be careful when selecting the boundary conditions and keep in mind how they affect the produced model results.

### 5.3.2 Initial conditions

**Initial conditions** are required for time-dependent equations. Together with boundary conditions, they control how the model evolves over time and are required to set up and drive the model. Since the conservation of energy (6) contains a time derivative $\partial T/\partial t$, we need to define an initial temperature field. This is difficult, since we have even less knowledge about what the Earth looked like at depth in the past than the present-day. Luckily, there are some useful strategies to come up with reasonable

initial conditions. In the case of oceanic lithosphere, the modeller can choose the age of the lithosphere and calculate the corresponding temperature profile according to well-established cooling models, such as the **half-space cooling model** or the **plate cooling model** (Turcotte and Schubert, 2014). For the continental lithosphere, an initially linear temperature profile or slightly more complex geotherm can be prescribed (Chapman, 1986). In the regions of the mantle that are not part of the thermal boundary layers, it is often a reasonable assumption to start from an adiabatic temperature profile, which is the temperature path along which material in a convecting mantle moves if it does not exchange energy with its surroundings (see also **adiabatic heating**). As the conservation equations of mass and momentum do not contain time derivatives of the velocity or pressure, we do not need to provide an initial velocity or pressure field. However, from a numerical point of view, a reasonable initial guess for the pressure or velocity can reduce the amount of iterations needed in iterative solvers and hence speed up computations. This becomes particularly relevant when using pressure- and strain-rate dependent rheologies, and is critical in cases when deformation mechanisms included in a model are strain-rate-dependent (such as pure dislocation creep).

The initial conditions also include the initial compositional geometry and material layout and/or history in the model, since the transport equation (15) contains a time derivative as well. For general parametric studies of the dynamics of a system, simple geometric blocks could be used to set up the initial geometry, representing for example the lithosphere. For more complex models, the initial conditions could be inferred from (regional) tomographic models. For complex models of specific regions, it is often difficult to manually create detailed geometries that correspond to geologic or tomographic observations. Therefore, several tools have recently been developed, such as geomIO (Bauville and Baumann, 2019), the SlabGenerator (Jadamec and Billen, 2012), and the Geodynamic World Builder (Fraters et al., 2019b), to automate and simplify setting up models with complex, 3-D geometries. Another choice the modeller has to make concerns the initial chemical heterogeneity present in the Earth's mantle. The simplest choice is to assume that the mantle is homogeneous or has been mixed so well that heterogeneities are on such a small length scale that they do not influence mantle dynamics. However, we know from geochemical data that the mantle and crust are chemically heterogeneous. In addition, subduction zones continuously introduce new chemical heterogeneities into the mantle. Hence, another option is to initialise the model with chemical heterogeneities on a given length scale, for example representing a model like the marble-cake mantle (Allègre and Turcotte, 1986), or to include distinct chemical reservoirs in the model setup. The initial state of deformation in terms of accumulated plastic or viscous strain or the state variable in rate-and-state friction laws can also be prescribed as initial conditions. Such initial strain can represent preexisting heterogeneity in the crust or lithosphere formed through deformation prior to the model start time. The top of the model can be used to assimilate topographic data into the model for a certain region or from another model (Section 3.3).

To initially drive the model in the absence of driving boundary conditions, specific initial conditions inside the model domain, so-called '**initial perturbations**', can be applied. A frequently used example in mantle modelling is a density or temperature perturbation in the middle of, or distributed throughout, the mantle. Such a perturbation ensures that subsequent deformation starts immediately, rather than after the accumulation of numerical rounding errors over time, and it localises the deformation in the region the modeller is interested in, e.g., a plume rises in the middle of the model.

Another common example of initial conditions is the so-called **weak seed** in numerical models: a small zone with artificially lowered strength used to localise the deformation in the model in the desired region. Without a weak seed, numerical inaccu-

racies will determine the location of instabilities and subsequent deformation, typically at the boundaries of a model. This is undesirable, because this results in irreproducible, random models and deformation influenced directly by boundary conditions. Hence, weak seeds are necessary in models and can represent naturally occurring heterogeneity in the Earth such as a previous fault zone or a region of melting (Buck et al., 1999). Weak seeds are commonly used in lithospheric- or crustal-scale models of rifting (Huismans et al., 2005; Allken et al., 2011), strike-slip (Schierjott et al., 2020), subduction (Erdos et al., 2015), and continental collision to localise the deformation and force the model to behave in such a way that the relevant process can be studied (Gray and Pysklywec, 2012; Yang et al., 2020). They can take numerous shapes and sizes and the lower strength of the weak seed can be achieved through many different methods, including a mechanically weaker seed (i.e., lower friction coefficient (Allken et al., 2012; Erdős et al., 2019)), initial strain or a temperature anomaly such as a locally raised thermal lithosphere-asthenosphere boundary (Brune et al., 2017). The choice of weak seed affects the numerical results, as it could for instance lead to different modes of rifting (Dyksterhuis et al., 2007). However, most studies argue that the weak seed does not significantly alter their conclusions, like Jammes and Lavier (2016) who consider random noise in the plastic strain of the crust in order to demonstrate that the weak seed does not control the geometry of the margin and the mechanism of deformation. The effect of weak seeds and robustness of the models can vary between codes.

The modeller should always keep in mind that the initial conditions can often determine the model outcome. That is, after all, their purpose, since otherwise there would be no localised deformation or initial drivers. Efficiently starting the model is solely at the discretion of the modeller, who aims to artificially mimic a process they are interested in. Since these initial conditions, in combination with the boundary conditions, are critical for the model development, the choices the modeller makes are sometimes referred to as the **hand of god** that helps the models along in the beginning. It is therefore important that initial conditions and their effects on the model are acknowledged and described alongside other important details of the modelling approach (Section 8.1).

## 6  Validation of the geodynamic model

After the code has been successfully tested and benchmarked (Section 4), every individual model setup with its particular modelling strategy (Section 5.2) and initial and boundary conditions (Section 5.3) should be carefully validated to make sure that it contains no detectable flaws, is internally consistent, and represents the geological problem to the best of our knowledge.

### 6.1  Common numerical problems

The construction of a specific model setup to investigate a particular problem or hypothesis can give rise to numerical issues, despite successful code verification. During the model validation process, these issues are identified and addressed. They can usually be spotted through monitoring solver convergence behaviour and visual inspection of the solution throughout the model evolution, with model breakdown (i.e. a crash of the program) and unexpected behaviour being the most obvious red flags. In this section, we describe a number of common problems and their potential solutions.

A **resolution test** should be standard to check whether a certain model, or model aspect, is mesh-dependent or not. So, the modeller should check the change in model results with higher mesh resolution. In the ideal case, from a certain resolution onward the solution no longer changes significantly or the spatial discretisation error becomes smaller than other errors like that resulting from time discretisation, i.e., the numerical solution has converged. It is desirable to use a grid spacing where, for example, the thermal boundary layers or crustal compositions are well resolved and the resolution does therefore not affect the model evolution anymore. However, it is not always possible to completely avoid grid-dependency. For example, most implementations of brittle (plastic) deformation do not include an internal length scale and are therefore grid-dependent (de Borst and Duretz (2020); Figure 10c). This grid dependency causes shear zone width to keep decreasing with increased resolution. There is an active research effort to include internal length scales that can limit grid-dependency (Lavier et al., 2000; Choi and Petersen, 2015; Duretz et al., 2018, 2019, 2020). The practical solution is to have one fixed resolution for the affected domain throughout the modelling study after assessment of changes in the overall model behaviour with resolution.

When using a free surface, fast increasing model velocities, a corresponding increase in solver iterations and a sloshing movement of the surface are indicative of the **drunken sailor** effect (Figure 10a). It occurs for models with a free surface where the time step is chosen too large to accurately reproduce changes in surface boundary elevation. The interface then over-shoots slightly in one time step, which causes it to overshoot in the other direction even further the next time step. The positive feedback of this numerical instability deriving from the stark density contrasts at the surface usually leads to the program crashing. The stark density contrasts (approximately 1.2 kg/m$^3$ versus 2830 kg/m$^3$) lead to much larger stress perturbations from topographic changes compared to similar topography variations at a typical density contrast inside the Earth (e.g. the density jump at the continental crust-mantle boundary is ∼280 kg/m$^3$ (Martinec, 1994)). Solutions to the drunken sailor problem involve using either smaller time steps, or a stabilisation algorithm (e.g. Duretz et al., 2011; Kaus et al., 2010; Quinquis et al., 2011; Kramer et al., 2012; Rose et al., 2017) to enable the model to run stably with larger time steps.

Another problem that can occur when mesh deformation is allowed in finite elements (i.e., in Lagrangian and ALE methods; Figure 5) is distortion of the mesh elements. The quality of the mesh decreases when element aspect ratios change too much from 1 for triangular as well as quadrilateral elements (e.g., the ratio of the width and height of a 2-D quadrilateral element (Figure 4), and similarly in 3D), and the accuracy of the computations therefore decreases. When the element distortion is too large, the computation will crash or at the very least become extremely inaccurate. To avoid such a distortion of the elements, local or global **remeshing** (i.e., mesh regeneration) or **mesh smoothing** can be applied (e.g., Zienkiewicz et al. (1995); Yamato et al. (2007); Steer et al. (2011); Thielmann et al. (2015); Duretz et al. (2016)). Diffusion of the surface topography can also help to stabilise the model by smoothing high gradient topography. (Nonlinear) diffusion can be implemented atop the model mimicking erosional processes (e.g. Burov and Cloetingh, 1997; Sternai, 2020), but remeshing of the free surface also introduces some numerical diffusion (Jammes and Huismans, 2012) that could help remedy instabilities. However, adding diffusion could possibly change the physics included in the model.

Visual inspection of the modelling results can uncover other issues. For one, smaller features (e.g. a subduction interface) can be seen to spread out over time and disappear; this can be due to diffusion or smearing of the advected field in the grid-based advection method. On the other hand, steep gradients of advected fields can lead to oscillations of these fields normal to the

# Problems

## a. *"Drunken sailor"*

## b. Chequerboard pattern

## c. Mesh dependence

**Figure 10.** Common numerical modelling problems. (a) Rayleigh-Taylor instability problem demonstrating the drunken sailor instability arising from a numerical time step that is too large (e.g. Kaus et al., 2010; Rose et al., 2017) for the stress perturbations deriving from surface topography due to the typical crust-air density difference being much larger than density differences inside the Earth. The large time step size leads to a fast sloshing of the surface, as seen from the velocity vectors. Note that the vectors in the model without stabilisation are scaled down by one order of magnitude. The high velocities also lead to overshooting of the advected compositional field, i.e., values exceed 1. When stabilisation is applied, a time step of 5000 years leads to a stable uprising in the centre of the domain. (b) The lid-driven cavity model (e.g. Erturk et al., 2005; Erturk, 2009; Thieulot, 2014) demonstrates the need for smoothing the pressure field when using $Q_1 \times P_0$ elements in the finite element method (in this case $32 \times 32$ elements). Using an LBB stable element, like $Q_2 \times Q_1$ or $Q_2 \times P_{-1}$, avoids this problem (Donea and Huerta, 2003). (c) Extension of a visco-plastic medium with shear bands forming at a viscous weak seed along the bottom (e.g. Lemiale et al., 2008; Kaus, 2010; Spiegelman et al., 2016; Glerum et al., 2018). The angle and thickness of the shear bands is dependent on the mesh resolution. Regularised plasticity implementations and sufficient resolution are required to achieve convergence with resolution (e.g. Duretz et al., 2020).

gradients. Mitigating such under- and overshooting requires more diffusion or different stabilisation algorithms of advection (e.g. Lenardic and Kaula, 1993; Brooks and Hughes, 1982; Guermond and Pasquetti, 2011).

Unstable yet very popular finite element pairs such as the $Q_1 \times P_0$ element (Section 3.2) are prone to spurious oscillations and an element-wise **chequerboard pressure pattern** (Figure 10b as shown in Figure 18 of Thieulot et al. (2008) or in Donea and Huerta (2003)). Before this pressure can be used in rheological expressions it must be postprocessed and smoothed so as to remove the chequerboard mode (Lee et al., 1979). Stable elements, which fulfil the Ladyzhenskaya-Babuška-Brezzi compatibility condition (LBB or inf-sup condition), do not exhibit pressure artefacts (Donea and Huerta, 2003) and are preferable (see Bathe (2014) for examples of such elements). Moreover, the required number of outer iterations does not increase significantly with mesh resolution compared to the $Q_1 \times P_0$ element (Thieulot and Bangerth, 2021).

## 6.2 Internal consistency of the model

Internal inconsistencies can arise from disagreements in the modeller's choices in terms of boundary conditions, density formulations in the different governing equations, and the equations' coefficients. Not all inconsistencies are easily detectable or manifest themselves as numerical problems. For example, when the net prescribed in- and outflow through the model boundaries is not (close to) zero, while a model is assumed incompressible, volume is no longer conserved and the solver for the Stokes equations might crash or return a nonsensical solution. When a free surface is used, this problem might be overseen, as the surface can rise or fall in response to a net increase or decrease of volume, respectively. This physical inconsistency is also harder to detect in compressible models. Another example is prescribed surface velocities based on, for example, plate reconstructions models, which can add unrealistic amounts of energy into the modelled system.

Care should also be taken that the assumptions made to simplify the treatment of density in the governing equations (see Section 2.1 and specifically 2.1.4) agree with one another. The simplest accepted combination of simplifications is the Boussinesq Approximation.

The thermodynamics of Earth materials are very complex, especially in multi-phase, multi-material systems, hence they are often simplified in the numerical model. For example, in nature the thermal expansivity varies both smoothly and abruptly with depth (e.g. Figure 7 of Stixrude and Lithgow-Bertelloni (2007)), but in models it is often taken as constant or merely smoothly increasing with depth. At first, the material properties described in Section 2.2.5 may appear to be independent. However, the definition of properties like density, thermal expansivity, specific heat, and compressibility need to satisfy thermodynamic relations in order for them to be consistent. These thermodynamic relations can be derived through **thermodynamic potentials**. For example, the thermodynamic potential of the thermal expansivity is defined as (Schubert et al., 2001):

$$\alpha = -\frac{1}{\rho}\left(\frac{\partial \rho}{\partial T}\right)_p, \tag{19}$$

where $\alpha$ is thermal expansivity. It defines the fractional increase in volume of a material per degree of temperature increase at constant pressure.

The thermodynamic potential of the isothermal compressibility is defined as (Schubert et al., 2001):

$$\beta_T = \frac{1}{\rho}\left(\frac{\partial \rho}{\partial p}\right)_T, \tag{20}$$

where $\beta_T$ is the isothermal compressibility and the potential describes the percentage increase in density per unit change in pressure at constant temperature. For the isobaric heat capacity, the thermodynamic potential is defined as (Schubert et al., 2001):

$$C_p = \left(\frac{\partial q}{\partial T}\right)_p, \tag{21}$$

where $C_p$ is the isobaric heat capacity and the potential is defined as the ratio of the increment of heat $\delta q$ added to the material to the corresponding change in temperature $\delta T$. The thermodynamic potentials also imply that:

$$\left(\frac{\partial C_p}{\partial p}\right)_T = -T\left(\frac{\partial \alpha/\rho}{\partial T}\right)_p. \tag{22}$$

All these relations have to be fulfilled at all temperatures and pressures. Consequently, it is often not immediately apparent if a given material description is thermodynamically consistent or not, and equations of state used in the geodynamic literature do not always take this into account (Schubert et al., 2001, Section 6.10). For example, it would be thermodynamically inconsistent to take thermal expansivity, compressibility at constant entropy (along an adiabat) and isobaric specific heat all as constants, if the temperature varies in the model, because, from Eq. 22, any temperature dependence in $\alpha/\rho$ leads to a pressure dependence of the specific heat.

## 6.3 Testing the model against observations

After the steps described in the previous section, checking for potential numerical issues and the internal consistency of the model setup, it is time to test whether the model results are consistent with our understanding of geodynamic processes. In a broad sense, does the model evolution stay within the bounds of what we know to be possible from geological and geophysical observations? More specifically, do the velocities obtained make sense? For example, does the sinking velocity of a slab lie within the estimated bounds from reconstructions and mantle tomography (e.g., van Der Meer et al., 2010; Butterworth et al., 2014)? Does the plume rise at acceptable speeds (e.g., Steinberger and Antretter, 2006)? Do the plate motions agree with plate reconstruction models (e.g., van Der Meer et al., 2010; Doubrovine et al., 2012; Matthews et al., 2016; Müller et al., 2019, for global reconstructions)? Does the surface heat flow lie within observed values (e.g., Davies, 2013; Lucazeau, 2019)? Does the density structure agree with 1-D average profiles like PREM (Dziewonski and Anderson, 1981)?

Note that deviations of the model results from generic observations do not necessarily mean that the results are wrong. In fact, a model of a natural system like the geodynamic models described here can never truly be validated (Oreskes et al., 1994). This is because natural systems are never truly closed. For example, neither the rocky planetary surface, nor the core-mantle boundary are true closed boundaries devoid of any temperature, compositional, or mechanical exchange with the outside world. As such, all model results are always non-unique and the model cannot be validated even if compared to a natural observation.

Considering this lack of experimental control from both the real world and analogue models from the lab (see Section 4), techniques such as uncertainty quantification and cross-validation of the models beyond the typical testing and predicting of hypotheses become increasingly important to assess how well models capture the real world.

## 7   Model analysis

After ensuring the accuracy, consistency and applicability of the model results, these can now be used to address the hypothesis the modeller set out to test according to a particular modelling philosophy (Section 5.2). The raw model output therefore requires analysis. While analysing the geodynamic model results, the modeller has to keep in mind all simplifications made during the setup of the physical model (e.g., what forces and processes were included), the initial and boundary conditions (e.g., whether subducting plates are free to move laterally or are attached to the boundaries), the resolution (e.g., whether the resolution is high enough to resolve a certain process), and all other numerical and physical model assumptions and uncertainties. Most importantly, the model cannot be mistaken for the real Earth (or any other real planet).

Model analysis includes visual (qualitative) diagnostics and quantitative diagnostics. These two important, partly overlapping, aspects are discussed in detail below.

### 7.1   Visual model diagnostics

Visualising the model output allows us to test, analyse, diagnose, and communicate the model results. Figures can describe and summarise the enormous amounts of data that numerical modelling can produce and highlight important features that support the initial hypothesis. Depending on the complexity of the data and the objective of the figure, visualisation methods differ widely and range from a graph of root-mean-square velocity (a quantitative model diagnostic, Section 7.2) over time to a complete 4D animation of a certain solution variable like the velocity (see also Section 8.2).

To cover the wide range of potential visualisation products, a multitude of visualisation programs is available. Some of the commonly used software packages are gnuplot for creating 2-D and 3-D graphs, GMT for making maps and plotting data on maps (Wessel and Luis, 2017), MatLab for scripted analysis and plotting of model data, matplotlib for static and interactive plotting in Python, ParaView for looking at 2-D and 3-D data sets both interactively and in batch jobs (Ahrens et al., 2005; Ayachit, 2015), and similarly VisIt (Childs et al., 2012). There are even fully immersive visualisation options like CAVE (Billen et al., 2008) and virtual reality display technologies (Kreylos and Kellogg, 2017) that overcome the necessity to project 3-D data to 2-D surfaces for visual inspection. However, no matter the software, a modeller has to pay careful attention to circumvent the most common **visualisation pitfalls** for effective graphics as is outlined in Section 8.2.

### 7.2   Quantitative model diagnostics

Mere visual inspection of the model results is not sufficient to analyse and interpret the outcome of the simulations; a quantitative analysis of the results is also required. Deciding what specific post-processing is to be done can be a time-consuming process. There are a range of non-dimensional numbers that can be calculated to characterise the flow of fluids in a range of

geodynamic environments. If multiple physical processes influence the behaviour of a system, the non-dimensional numbers derived from the governing equations (Section 2.1) can be used to analyse the relative importance of each of these processes. Examples are the **Rayleigh number**, *Ra*, the **Knudsen number**, *Kn*, the **Mach number**, *M*, **Argand number**, *Ar*, the **Ekman number**, *Ek*, the **Reynolds number**, *Re*, the **Peclet number**, *Pe*, the **Prandtl number**, *Pr*, and the **Nusselt number**, *Nu*. The Rayleigh number, in particular, is a widely used non-dimensional number characterising the vigour of buoyancy-driven flow. Many of these non-dimensional numbers can be defined differently for different environments, but they can all be insightful diagnostic metrics. With increasing model complexity, newer diagnostic numbers have been defined, such as plateness (Tackley, 2000; Alisic et al., 2012; Stein et al., 2014).

Further analysing and diagnosing a model then varies with the modelling approach that has been taken (see Section 5.2). For **specific modelling** (i.e., model approaches directly comparing to an observation to understand the origin of a specific state of a system), a modeller should diagnose whether the model predictions match the observations to a satisfying degree. To test the match between the model prediction and the natural observation, a modeller can perform a visual comparison, statistical analysis like a misfit measure, and check for a comparable dynamic model behaviour. Secondly, it should be diagnosed whether the hypothesised mechanism is actually responsible for creating the physical complexity of interest. To this end, the model sensitivity to the used (and optimally also unused) parameters needs to be systematically tested. It should be clear whether a variation of a parameter over its uncertainty range affects a model outcome or not, and if it does, by how much. Similarly, it should be clarified what impact currently neglected parameters would have on the model outcome, for example by discussing the results of other modelling studies that did include these parameters. After such diagnostics, the model can provide insight into what causes a specific state of the system.

For **generic modelling** (i.e., models used to reproduce basic fluid dynamics to understand how a specific system works), a modeller should diagnose whether there are individual models that exhibit similar behaviour within the model parameter space and whether there are controlling parameters that can, when changed, cause a switch from one regime to another. In general, often-used diagnostic quantities to define regimes are the root-mean-square (RMS) and maximum values of certain model parameters like velocity or temperature. For example, the root-mean-square velocity over time can be used to check whether a model has reached steady-state or shows signs of periodicity (e.g. Tosi et al., 2015). For subduction models, the slab dip, slab tip depth, slab sinking velocity, plate age at the trench, and the interaction of the slab with the mantle transition zone are often monitored (e.g. Garel et al., 2014; Crameri and Lithgow-Bertelloni, 2018). Specific isotherms, fault length and offset, rift symmetry, and the migration of fault activity are often tracked in extensional models (e.g. Huismans and Beaumont, 2003; Naliboff et al., 2020). From general convection models, the degree of entrainment or mixing, the convective regime (e.g., stagnant lid versus mobile lid), and the degree of layering are often extracted (Heyn et al., 2018; Foley, 2018; O'Neill et al., 2016).

### 7.3 Automated model diagnostics

While some model analysis can be done by hand, the more elaborate post-processing that becomes increasingly popular nowadays needs to be automated using open-source, testable, and extendable algorithms and shared as user-friendly soft-

ware (Greene and Thirumalai, 2019). To achieve such next-generation post-processing, like plate boundary tracking, extraction of lithosphere thickness, or computing the dynamic or isostatic topography (Crameri, 2017b), the output of geodynamic codes should ideally use a standard, widely-accepted format and include metadata that can also be accessed by machines. Accessing individual subsets of the data, like individual time steps or parameter fields, should be straightforward. Model output should therefore be independent from computational details, like the number of computational cores the results have been produced on.

A few software packages that allow for automated post-processing and diagnosis of geodynamic models are available to support geodynamicists with analysing their increasingly complex models, and the large datasets originating from them. However, such tools are rare, because while most individual researchers spend a large amount of time in coding post-processing scripts, they often do not share those scripts with the geodynamics community. Moreover, scripts that are shared in the context of repeatability and transparency are not necessarily applicable or relevant to other software output. Making their own post-processing scripts more generically applicable can also not be required of individual scientists. Contributing to post-processing tools as part of a community software project is a great step forward, reducing the duplication of work while providing author recognition. Unfortunately, not all available post-processing tools supplied with community software can be applied to results from other codes. Defining a set of interfacing functions, like the Basic Model Interface established by the Community Surface Dynamics Modeling System (CSDMS), facilitates communication between different modelling and post-processing tools.

Generic, open-access geodynamic diagnostic tools are Adopt (Mallard et al., 2017) to detect and outline surface plates, StagPy to diagnose plate boundaries, and StagLab (Crameri, 2017b, 2018c) to perform a large set of geodynamic diagnostics. StagLab in particular offers an extensive set of robust 2-D and 3-D diagnostics. It works across multiple model domain geometries, resolutions and model setups and produces publication-ready, fully annotated figures and movies using exclusively Scientific colour maps (Section 8.2). Moreover, StagLab is versioned and extendable, and it can process output of more than one mantle convection code, while also offering low-effort compatibility for any other machine readable output.

Other recent developments include the automated comparison of observations to model predictions to find the smallest misfit between the two. Such statistical and probabilistic inversion methods help determine the model parameters, e.g., mantle viscosity or crustal density, that result in the best fit of the model solution with the observed quantity through forward geodynamic modelling (e.g. Baumann et al., 2014; Baumann and Kaus, 2015; Li et al., 2017; Reuber et al., 2018b; Morishige and Kuwatani, 2020; Ortega-Gelabert et al., 2020).

## 8 Communicating modelling results

Scientific results are only of value if they are communicated to the wider scientific community. No matter whether they are spoken or written, the first aspects to get right when communicating science concerns letters, words, and phrases. Since geodynamic modellers, like most other life forms, tend to learn most effectively by observing and copying other fellows, it is no surprise that we tend to speak and write in a similar way to our mentors, peers, and friends. While there is generally nothing wrong with that process, it does, however, make for an excellent breeding ground for problems related to semantics that can

lead to serious miscommunication (e.g., Crameri, 2018a). Written and spoken communication therefore needs to be held dear and handled carefully by a geodynamic modeller.

The semantics behind a modelling publication or presentation need to be in tune with the approaches taken in the modelling itself. If a modelling study is suitably communicated, there will be less misunderstanding about what the presented model stands for, what it does not stand for, and what the drawn conclusions mean.

Because geodynamic models are per definition simplifications of a natural system (Section 1.1), their individual features should not be mistaken for an exact replica of their natural counterparts. When communicating modelling aspects, semantic differentiation between the feature in the model and in nature helps to avoid confusion. For example, one should refer to 'the modelled slab' instead of 'the slab' or 'the subducted Nazca plate'. On a similar note, one should distinguish between 'thermo-chemical piles', which are collections of material with different thermal and chemical properties as the surrounding mantle, and 'LLSVP's', which are observed regions of low shear wave velocity along the core-mantle boundary.

In addition, care has to be taken with absolute statements, like 'X on the Earth is due to Y', when drawing conclusions from the model results. As discussed in Section 5.2, models can only demonstrate a certain likelihood of an hypothesis, and in particular specific modelling studies should acknowledge that there is no guarantee that the suspected control mechanism is the actual, nor the only, controlling mechanism. Statements like 'X on the Earth is likely due to Y', 'Y is a potential explanation for X', and 'if our assumptions A, B, and C are fulfilled, then XYZ will probably happen in the Earth' are more correct and prevent misconceptions.

Communicating a geodynamic modelling study, however, goes beyond semantics. The suitable words and phrases are most effective when combined with an appropriate manuscript structure and effective still and even motion graphics. Combined, these forms of communication make a new scientific insight accessible.

## 8.1   Structure of a geodynamic modelling manuscript

Peer-reviewed scientific papers are essential to disseminate relevant information and research findings. In particular, it is important to make results understandable and reproducible in the methods and results sections. Reviewers will criticise incomplete or incorrect method descriptions and may recommend rejection because these sections are critical in the process of making the results **reproducible** and **replicable**. In this section, we briefly review the structure of a manuscript and highlight the parts required in a geodynamic numerical modelling study (Pusok, 2019).

While there are many ways of writing a paper, the main purpose of a scientific paper is to convey information. Historically, the structure of scientific papers evolved from mainly single-author letters and descriptive experimental reports to a modern-day comprehensive organisation of the manuscript known as 'theory-experiment-discussion' (Audisio et al., 2009; Liumbruno et al., 2013). The formal **IMRAD** structure (i.e., Introduction, Methods, Results, And Discussion/conclusions) was adopted in the 1980s, and at present, is the format most widely used and encouraged by scientific journals. The IMRAD structure facilitates modular reading and locating of specific information in pre-established sections of an article (Kronick, 1976). In geodynamics, the general structure of a manuscript follows the IMRAD structure (Figure 11), although journals can place different emphasis on the individual components through reordering and formatting.

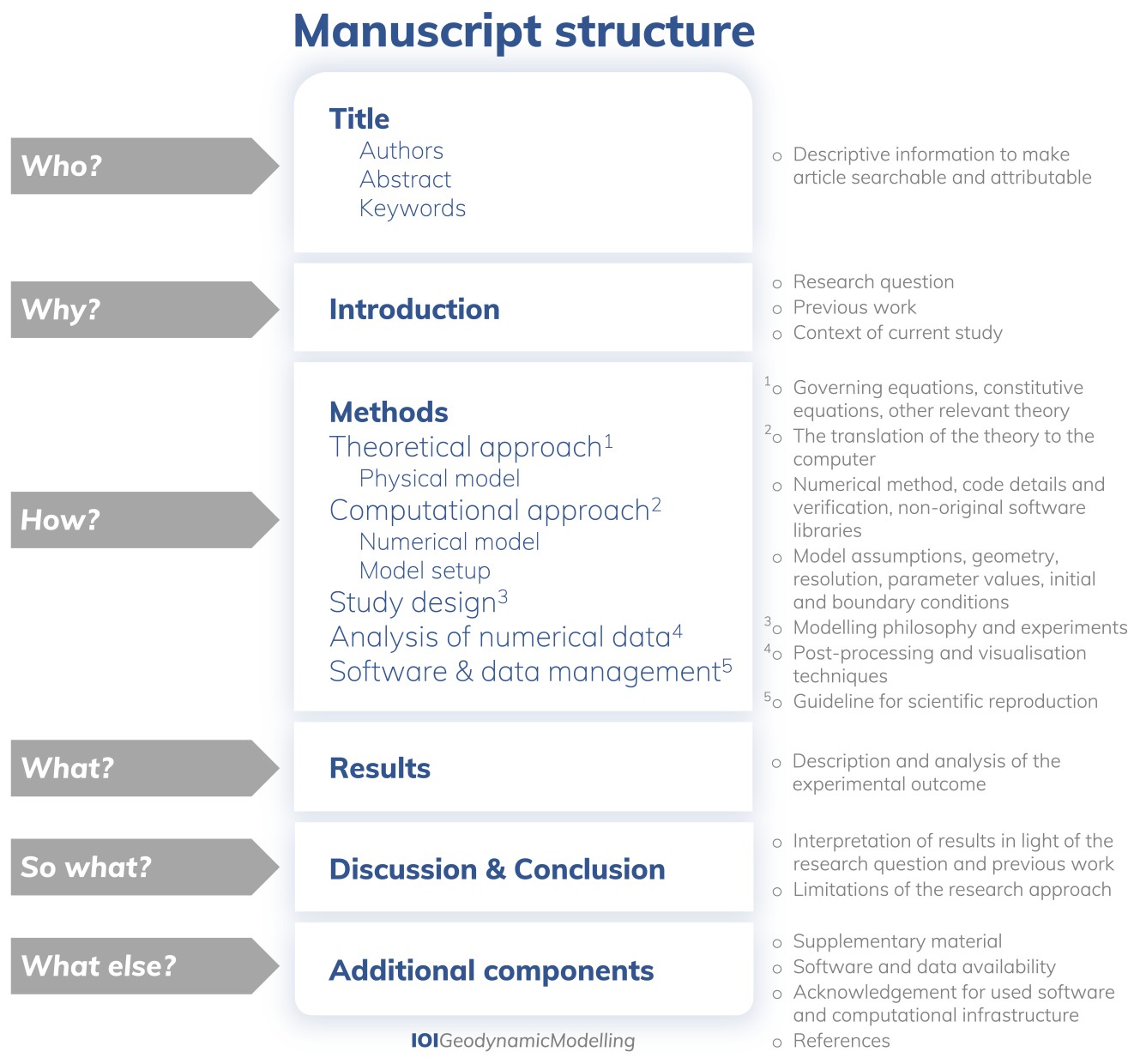

**Figure 11.** Manuscript structure for a geodynamic numerical modelling study following the IMRAD structure. In particular, the methods section should include a description of the physical and numerical model, the design of the study, and of any techniques used to visualise and analyse the numerical data.

A good introduction should answer the following questions: what is the problem to be solved? What previous work has been done? What is its main limitation? What do you hope to achieve? How do you set up your investigation? One major mistake is to attempt to do an extensive literature review in the introduction, which often goes off-topic. The introduction serves as the stage to lay out the motivation for the study, and any background reading should focus on the question being addressed.

The methods section is an important part of any scientific manuscript (Kallet, 2004). A good methods section allows other scientists to verify results and conclusions, understand whether the design of the experiment is relevant for the scientific question (validity), and to build on the work presented (reproducibility and replicability) by assessing alternative methods that might produce differing results. Thus, the major goals of the methods section are to verify the experiment layout and allow others to reproduce the results. Here, we outline standards for reporting the methods in numerical geodynamic modelling.

First, the methods should be plain and simple, objective, and logically described and should be thought of as a report of what was done in the study. Unstructured and incomplete methods can make the manuscript cumbersome to read or even lead the reader to question the validity of the research. Generally, journals have guidelines on how the methods should be formatted, but not necessarily what they should contain because they vary from field to field. The 'who, what, when, where, how, and why' order proposed by Annesley (2010) breaks the methods section down in the following questions: who performed the experiment? (not directly applicable to geodynamics, although one might mention here the specific cluster or supercomputer on which the simulations were run) What was done to answer the research question? When and where was the experiment undertaken? (i.e., what computational resources and which software versions were used?) How was the experiment done, and how were the results analysed? Why were specific procedures chosen? The answers to these questions should be adapted to every field (i.e., in geodynamics, 'results were obtained using code X on cluster Y'). Here, we focus on methods that have primarily theoretical (mathematical and physical) and numerical (computational) components, but geodynamic studies may have other aspects such as a data component (e.g., collection and post-processing of data) or an analogue component (e.g., laboratory experiments).

Figure 11 shows a breakdown of the most important elements of a manuscript and of the methods section in particular. The methods should start with a brief outline (one paragraph) describing the study design and the main steps taken to answer the scientific question posed in the introduction. The outline should be logical and go from theoretical elements, to numerical aspects, to analysis and post-processing. First to be described is the theoretical framework. This includes the mathematical and physical concepts used in the study including the governing equations, constitutive equations, initial and boundary conditions, and any other relevant theory describing the model setup.

This is followed by a section on the computational approach explaining how the theory and the model are translated into computer language (Section 3). This includes details on numerical methods (discretisation and other numerical techniques, code details, solvers and software libraries, etc.), and model setup. The model setup subsection should include details on the current experiment such as model geometry, resolution (numerical and physical), parameter values, and initial and boundary conditions. Any necessary figures of model geometry and tables of parameter values should be provided. More importantly, the choice of parameters should be motivated to explain their relevance to addressing the scientific question.

After the model setup has been explained, the methods should contain a section describing the design or layout of the study in detail. What is being tested/varied? How many simulations were performed in terms of model and parameter space? For example, one can use different model setups (i.e., lithosphere-scale and mantle-scale subduction models) with varying parameters in the same study. Why perform those simulations and vary those parameters? A summary table is handy to indicate all simulations that were run and which parameter was varied in which run. Additionally, it is important to include information on all input parameters and their values and units, as well as possible conversions within the code to enable reproducibility of the study (see Section 9) and to foster transparency. This information ideally takes the form of an extensive table including the description, symbol, value, and SI units of the parameters. Errors may still be introduced during manuscript writing, and published values of input parameters may differ to values actually used in the numerical model. Automatic routines to print input parameters in publishable format directly from the code can avoid these mistakes in models, and increase transparency and replicability. As an example, laboratory-derived creep laws are often listed haphazardly in tables for publications. However, published laboratory data may need conversion due to unit change (i.e., $MPa^{-n}s^{-1}$ to $Pa^{-n}s^{-1}$, for $n > 1$ in creep laws) or correction due to the type of experiment (uniaxial, simple shear) in order to be directly usable in geodynamic models (see Gerya (2019)). This example also demonstrates the importance of consistently listing the units of all parameters used in a study, preferably in SI units.

Analysis, visualisation and post-processing techniques of numerical data should also be described in the methods section. This is a step generally ignored, but it is important to be open about it, e.g., 'visualisation was performed in ParaView/Matlab, and post-processing scripts were developed in python/matlab/unicorn language by the author'. If the post-processing methods are more complex, the author can provide more details (i.e., statistical methods used for data analysis). It is also good practice to provide these post-processing scripts for peer-reviewing and community engagement (Section 9).

Information should also be given on code and data availability. This has originally been part of the methods section, but recently journals have introduced data management requirements (Section 9) and this information may have a designated location in the manuscript template. However, it is good practice to write this information in the methods section. The authors should indicate code availability, code verification, and input files or other data necessary to reproduce the simulation results (e.g., installation guides). Additional questions to be answered in the methods section are: where were the simulations performed, and on how many cores? What was the average runtime of a simulation? Can the model results be reproduced on a laptop/desktop computer or is access to a cluster required?

Before moving to other sections, model assumptions need to be stated clearly, in either the description of the theory or the numerical approach. Geodynamics is a field in which we take a complex system like the Earth or another planetary body and simplify it to a level from where we can extract some understanding (Section 5). In doing so, we rely on a physically consistent set of assumptions. It is important to bear in mind that this set of assumptions may not always be obvious to the reader. As long as assumptions are explicit and consistent (i.e., through clear and honest communication), the reviewers and readers will find fewer flaws in the study. It is good practice to write a complete methods section for every manuscript, such as the one described here. However, some journals will ask for a short version to be included in the main manuscript and have the complete methods section in a separate resource (i.e., in the appendices, supplementary data, supporting information, online repository).

Complementary to the methods section, the results section should be a report of the results obtained. The main goal of the results section is to present quantitative arguments to the initial hypothesis. However, any interpretation of the results or reference to other studies should be reserved for the discussion. For example, results in a mantle convection model might show that dense material accumulates at the bottom of the domain (i.e., core-mantle boundary). The interpretation of these results is that they provide a mechanism to explain how LLSVPs (large low shear velocity provinces) have formed. Illustrations, including figures and tables, are the most efficient way to present results (Section 7). However, authors should only include material and information that is relevant to demonstrate the scientific arguments discussed in the next section. Therefore, to avoid distraction, writers should present additional data as supplementary materials, e.g., movies of the whole simulation, whereas only a few snapshots are provided in the main body of the manuscript.

The discussion section relates all the questions in the manuscript together: how do these results relate to the original questions or objectives outlined in the introduction section? Do the results support the hypothesis? Are the results consistent with observations and what other studies have reported? The modeller should discuss any simplifying assumptions, shortcomings of numerical methods and results, and their implications for the study. For example, the discussion in a specific modelling study (Section 5.2) should address how applicable the model results are to the specific problem/region, whereas the discussion in a generic modelling study should aim to understand the underlying factors in a given system. If the results are unexpected, authors should try to explain why. Are there other ways to interpret the results? What further research would be necessary to answer the questions raised by the results? What does it all mean? Many manuscripts are rejected because the discussion section is weak and the author does not clearly understand the existing literature. Writers should also put their results into a global context to demonstrate what makes those results significant or original.

At this point in preparing the manuscript, the authors have all the necessary elements to write the abstract and conclusions and come up with a descriptive title. Both the abstract and conclusion summarise the entire publication, but in a different way; one as a preview and one as an epilogue, respectively. It is crucial to focus a paper on a key message, intended for both specialist and non-specialist readership, which is communicated in the abstract and conclusions. Some journals also include a Plain Language Summary and/or Graphical Abstracts, as alternative ways to engage a broader audience.

In the end, every scientific manuscript has additional components such as the references, acknowledgements, supplementary material, software and data availability, and author contributions (Figure 11) that contain further information about how the study was funded, conducted, and shared with the community. Acknowledging the often substantial contributions of reviewers is a common courtesy.

In this section, we have primarily referred to scientific articles, but scientific manuscripts can also be reviews, editorials, and commentaries. The structure and contents of these manuscripts differ for each type. Each publisher and journal has their own style guidelines and preferences, so it is good practice to consult the publisher's guide for authors. Finally, even though scientific manuscripts may have a rigidly defined structure due to journal guidelines, there is still plenty of flexibility. In fact, the best manuscripts are creative, tell a story that communicates the science clearly, and encourage future work.

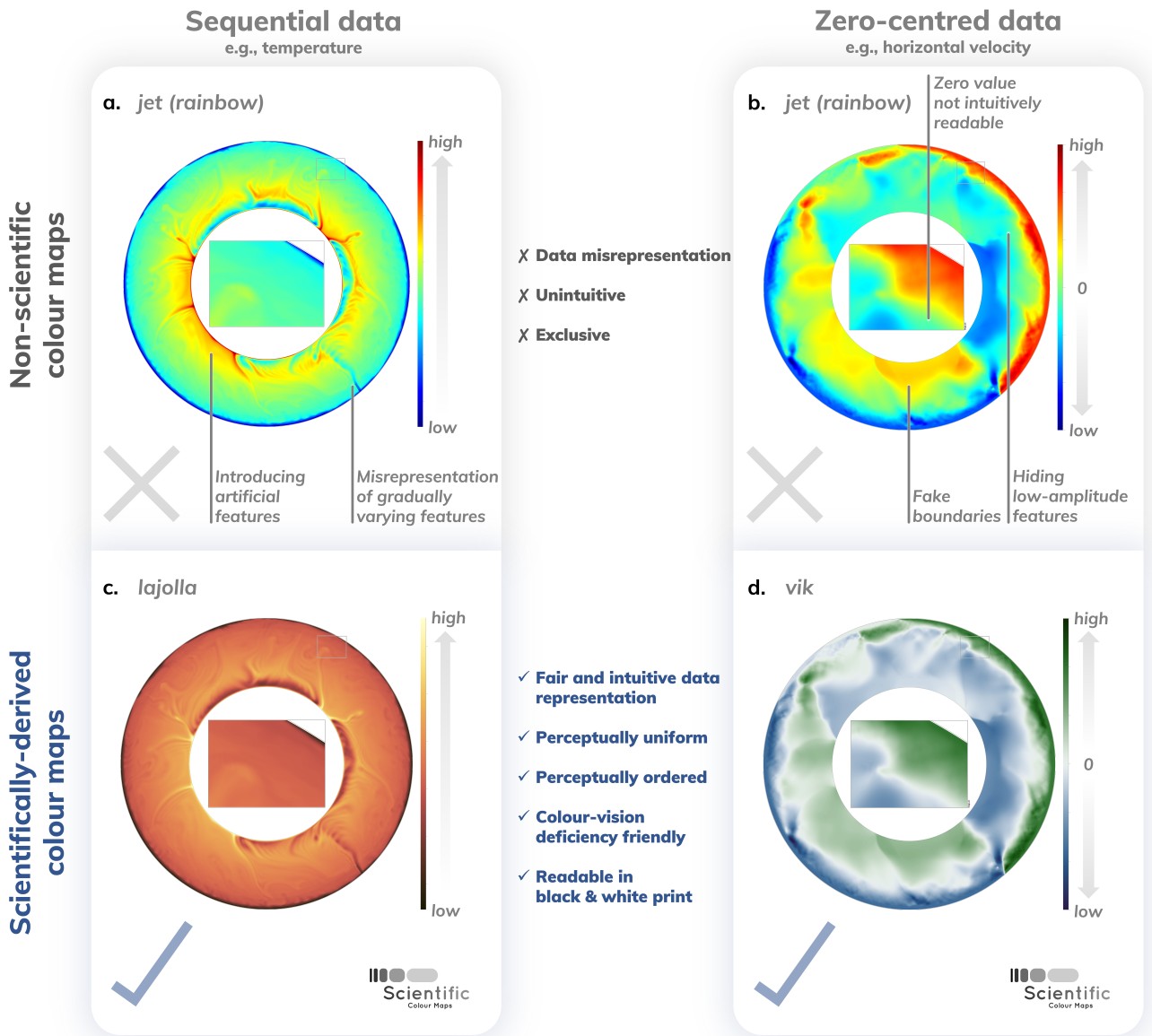

**Figure 12.** Effective visualisation through a scientific use of colours. Non-scientific colour maps (a,b) like rainbow always misrepresent data, are often not intuitive, and are inaccessible to a large portion of the readers, while scientific colour maps (c,d) like lajolla or vik (Crameri et al., 2020) ensure unbiased and intuitive data representation and are inclusive to readers with colour-vision deficiencies and even colour blindness.

## 8.2 Effective visualisation

There are many different ways to visualise geodynamic models and it is challenging to figure out how to do so most effectively. However, avoiding the most-common visualisation pitfalls is the best start for any modeller looking into visually communicating important results across the research community and possibly beyond. The key aspects to remember when creating figures, thereby preventing misleading visual impressions, are the following: (1) Scales, like graph axes and colour bars, must always be included to allow quantification of data values. (2) Bar plots must always have a zero baseline (or in the logarithmic case, have a baseline at one), to not mislead the reader with altered relative bar heights. (3) Pie diagrams should be avoided as angles and areas are not easily quantifiable by the human brain and therefore are not directly comparable to each other. These problems are exaggerated when pie charts are displayed as 3-D structures, which causes the values underlying the pieces closest to the viewer to appear artificially larger than the others. (4) Heatmaps (i.e., plots with differently coloured tiles) should have numbered tiles that include the data value, as surrounding colours heavily distort the impression of a given colour, which can mislead the viewer's perception of the underlying data values significantly (Crameri et al., 2020). (5) Colours must be applied in such a way that data is reflected correctly and is inclusive to all readers (Crameri et al., 2020). Scientifically-derived colour maps exist, like Colorbrewer, MPL, Cividis, CMOcean, CET, and Scientific colour maps (Kovesi, 2015; Thyng et al., 2016; Nuñez et al., 2018; Crameri, 2018b), and must be chosen over unscientific default colour maps like rainbow, jet, seis, and others (Figure 12). (6) Visualisation should be subject to the same scientific scrutiny as other research methods to communicate data and concepts truthfully, accessibly, and clearly.

All aspects of a figure need to be explained and understandable. While filtering, resampling or otherwise postprocessing model results instead of plotting raw data can improve the message purveyed by the figure, such actions should be mentioned in the figure caption. Some numerical codes work, for example, in dimensionless numbers (Section 7.2) and require scaling of the model output before they can be related to observations. However, too much information jammed in a figure can easily render the figure unusable to the reader. Again, the modeller has to simplify enough to reach the sweet spot, without oversimplifying the figure (compare with Section 5.1). Everything that can be removed without losing key information should be removed. Unnecessary and/or duplicated axes labels, e.g., those repeated across multiple panels, should be removed. The same applies to other figure aspects like colour bars. To make a figure intuitive to readers, colour bars in multiple figure panels applied to the same parameter should optimally maintain the same range (i.e., map the same colour to the same data value) and, if they do, displaying just one colour bar is sufficient.

Displaying 3-D models effectively is challenging, and somewhat arbitrary, as the third dimension is often difficult to convey in a still image. Given the current dominant occurrence of non-interactive, two-dimensional canvases (e.g., the pdf format), 2-D slices of parameter fields often represent the model more effectively than 3-D volumes. The combination of various data sets, like flow characteristics on top of a temperature field, can be effective, but is also challenging. Velocity arrows, for example, should not overlap, nor distract from the remaining important content of the figure. If the velocity in a 3-D visualisation is displayed using arrows, they should be coloured according to their magnitude because their lengths are distorted by the 3-D

perspective. Stream lines and a coloured contour plot of the velocity field often provide a more suitable solution to display the flow direction and patterns, and its velocity magnitudes, respectively.

An uninformed, unscientific use of colours does not only exclude a large portion of the readership, for example through hardly distinguishable colour combinations for readers with a colour-vision deficiency (like the most common red-green colour blindness), but also significantly distorts the underlying model data visually (e.g., Crameri, 2017a, 2018c, and Figure 12). In fact, the distortion can be more than 7% of the displayed data range (e.g., a temperature variation of 20 degrees could look like 24 degrees in one part along a colour axis that ranges between zero and a hundred degrees, while it looks like 17 degrees in another part), which in most cases easily amounts to the biggest error in a modelling study. Therefore, modellers should avoid using default colour maps blindly, but instead look for **scientific colour maps** (Crameri et al., 2020). If a software does not offer them, or at least allows to import them, users should contact the software developers. Hence, scientific colour maps should be the default in common geodynamic codes and visualisation programs.

Scientific colour maps are perceptually uniform to prevent data distortion, and they are perceptually ordered to make data intuitively readable, colour-vision deficiency friendly, and optimally readable in black and white to include all readers. Suitable scientific colour maps (e.g., Crameri, 2018b) that offer fair data representation and are fully inclusive have become readily usable (either as built-in options or after importing them) with all major modelling, visualisation, and graphics software, like MatLab, Python, R, GMT, QGIS, ParaView, VisIt, Mathematica, gnuplot, GIMP, and Inkscape. Indeed, the scientific colour maps suite (e.g., Crameri, 2018b) offers custom colour map classes for any data type. The sequential class (e.g., batlow) is best suited for monotonically increasing data like a temperature field. The diverging class (e.g., vik) is best suited for zero-centred data like a horizontal velocity field. A multi-sequential class (e.g., oleron) should be used with zero-centred, non-diverging data like a combination of ocean bathymetry and land topography and the cyclic class (e.g., romaO) is tuned for circular data like azimuthal direction. Moreover, these scientific colour maps are offered as a continuous type (e.g., batlow) to visualise small-scale data variation, a discrete type (e.g., batlow10) to highlight similar data values more clearly, and as a categorical type (e.g., batlowS) to clearly differentiate unordered data for line or scatter plots. In general, colour maps are an integral part of the colour axis (or colour bar) and their key purpose is to represent – and not interpret – data. As such, the local colour gradient, like the tick-spacing along an x-axis, should be the same all along the entire colour bar (or vary constantly in the case of a logarithmic axis). To interpret or highlight certain parts of the rendered data, other graphic tools like superimposed contour lines or visual indicators such as arrows should be used. More in-depth information, a clear user guide to using colour and colour maps, and a concise list of the major currently available scientific colour maps are provided in Crameri et al. (2020).

## 9 Software, data, and resource management

Just like any other study, numerical modelling studies should be **reproducible** (i.e., the same researcher doing the same computation on the same machine produces the same measurement at stated precision), and most importantly **replicable** (i.e., different researchers can obtain similar enough results from independent development) (Bollen et al., 2015; Goodman et al., 2016). Note that the actual terms used for these concepts vary across the sciences (Association for Computing Machinery,

2016; Karlsen, 2018; Plesser, 2018), with the terms reproducible and replicable for example used interchangeably. Reproducibility and replicability imply that the software used to conduct the study as well as the specific model setups, installation environment specifications, and post-processing workflow should be available (Perkel, 2020) to interested peers, and preferably, everybody. More and more, scientific journals are requesting or even requiring the publication of data and software along with the manuscript. Although the requirements vary per journal, it is good practice to adhere to these principles for every publication.

Before development starts, software developers and modellers involved in the development of the software they use should consider setting up a **software management plan** (SMP; see the SMP template by Jackson (2018) of the Software Sustainability Institute). This includes, but is not limited to, the following questions: what software will be produced? Who will use the software? How will the software be made available to the users? What support will be provided to the users? How will the software contribute to research and how can this contribution be measured? Where and how will the software be deposited to guarantee its lasting availability? Certain organisations that provide a platform for software packages state their own guidelines and requirements, such as the Computational Infrastructure for Geodynamics (CIG).

In the **hero codes** development strategy, one or only a few people are responsible for the development and maintenance of a modelling software package (Thieulot, 2017). **Community software** efforts are often managed through **version control software** like **git** and **svn** and corresponding platforms like GitHub, GitLab, and Bitbucket (e.g., ASPECT (2014) , Kronbichler et al. (2012); Heister et al. (2017); LaMEM, Kaus et al. (2016); pTatin3D, May et al. (2014); and PyLith, Aagaard et al. (2013)). These platforms provide open as well as private repositories for software development, issue tracking, automated testing, and project management, greatly simplifying the points addressed in the SMP. They also facilitate contributing the modeller's own development efforts to the main repository of the software such that they are available to other researchers. Moreover, extraction of statistics on the number of downloads, users, and contributors is made easy. However, these platforms themselves do not provide persistent identifiers and are not considered archiving facilities according to the FAIR Data Principles discussed below.

When archiving software developments or additions, one should take care to include instructions for installation and use as well as ample comments explaining the code. Software containers such as Docker containers (https://www.docker.com/) can help, as they bundle the program's complete run time environment into one package. The minting of a persistent identifier (PID) like a digital object identifier (**DOI**) for the repository should be a standard procedure and at the same time ensures the modeller can get credit for their coding efforts through citation of the source code. DOI minting can for example be done through Zenodo (https://zenodo.org/) for GitHub repositories. Writing code can take up a significant amount of time for modellers, although a modeller is not necessarily a developer. This time spent on something other than publishing papers should be acknowledged, as stated in the Software citations Principles established by the FORCE11 working group (Smith et al., 2016). Archiving data and code properly also facilitates others finding and reusing the work. Guidelines for writing reproducible code and research projects in general can for example be found here: Wilson et al. (2014b), Wilson et al. (2017), The Turing Way Community et al. (2019), and at the Software Sustainability Institute (https://www.software.ac.uk/resources/guides/guides-researchers).

Apart from software, modelling studies also use and produce data. For one, observational data can be provided as input to the model setup and usage of this data when created by others should be duly referenced. Then, simulations produce data – the

model results. Wilkinson et al. (2016) focused on the reuse of scientific data by providing **FAIR** Data Principles to improve the ability of machines to find, access, and use datasets. The FAIR Data Principles promote the Findability, Accessibility, Interoperability, and Reusability of data. Several initiatives are built upon these principles, like GO FAIR, the Coalition for Publishing Data in the Earth and Space Sciences (COPDESS), Enabling FAIR Data and the European Open Science Cloud. When input files, software, and machine specifications are properly described, identified, and made available, numerical data does not have to be archived, as the study can be reproduced from those elements. However, accessible model results can save the computational resources needed to recreate the model results, serve reviewer assessment and educational purposes, and post-processed results can be of interest to the general public (e.g. movies on YouTube or graphics on figshare; Pusok, 2020). Note that workflows themselves can be made public as well through platforms like WorkflowHub (https://workflowhub.eu).

To help modellers with the implementation of the FAIR Data Principles, publishers and data repositories formed the coalition COPDESS. The COPDESS website (http://www.copdess.org) explains why and how to archive and publish software and accompanying data and includes FAIR author guidelines, links to the guidelines of associated journals, and the Repository Finder from the Enabling FAIR Data project that searches re3data.org for FAIR-aligned repositories. An extensive explanation of the FAIR Data Principles as well as a starter kit for research data management for researchers can be obtained from the GO Fair initative. Proper research data management starts with addressing the following points (after Science Europe, 2018): (1) description of data, its collection/production and the reuse of existing data sets; (2) data metadata and management; (3) storage, backup and security during the project; (4) legal and ethical requirements, codes of conduct; (5) what, how and where will data be stored, accessed and identified?; (6) who will deposit and maintain the data?

Data repositories can be subdivided into institutional repositories, domain-specific repositories (e.g. EarthChem, IRIS, PANGAEA and HydroShare), thematic data repositories (which differ from domain-specific repositories by having to transform the data to the repository's format yourself, e.g. NASA's Planetary Data System) and general repositories, like figshare, Dryad and Zenodo (Stall et al., 2020). The library of a modeller's institute can explain what repositories they support and what workflows already exist for archiving data with persistent identifiers. In general, institutional and domain-specific repositories provide more support and quality control in submitting the data, while general repositories do not set specific requirements for the data. Also, by depositing data in a domain-specific repository, it is more likely to be found by the target audience. Useful repositories also provide you with copyright licensing options, which for research data are commonly **CC zero** and **CC BY** (Creative Commons). These licensing options standardise the granting of copyright permissions for research data and software, stating who can reuse creative work in what way, for example without making modifications, but with attribution to the original creator. Lastly, the dataset or software with its persistent identifier should be properly cited in the corresponding article which should clearly state how it can be accessed. Purely descriptive articles about a new dataset without any data interpretation can be published in, for example, Earth System Science Data, Geoscience Data Journal, Earth and Space Science and the Data Science Journal.

Not only data can (and should) have a persistent identifier, researchers can also create persistent digital identifiers, like an ORCID iD, for themselves. These identifiers can be connected to all types of research products, such as articles, grants, and peer reviews, providing acknowledgement of the work and preventing identity mix-ups. The article processing portals of many

journals already allow or require researchers to link their ORCID iD to their profile, and these iDs are then linked to in the published manuscript. Another development in article publication is the rise of reproducible articles, where not only the output, but the whole code used to produce the output is interactively included in the manuscript, using for example R Markdown or Jupyter Notebooks. For massively parallel geodynamic computations, such a reproducible article is currently not feasible, but we would like to invite both authors and reviewers to critically assess the reproducibility of their geodynamic modelling studies (Pusok, 2020).

One last thing modellers should consider is that numerical modelling does not come for free. As a community, we have to acknowledge the environmental impact, especially of high-performance computing and data storage. In a busy year (e.g., 1 million CPU hours), computations of one researcher can emit up to 6 tons of $CO_2$ on a modern high-performance machine (e.g., a power draw of 400 W per processor; see Lannelongue et al. (2020) and their computing emissions calculator). For comparison, a trans-Atlantic round-trip flight (e.g. Berlin-San Francisco) produces about 2 to 3 tons of $CO_2$ per person (myclimate $CO_2$ calculator, Ivanova et al. (2020)). A conscious effort should therefore be made to not waste computing power. For example, short, low-resolution models should be favoured to test new implementations and setups and weak and strong scaling test results (Section 3.5) should be used to determine the optimum amount of processes for large production runs. Furthermore, code developers should care about code optimisation from the start and could additionally provide both debug and release software versions, where less time-consuming internal checks are performed in release mode for production runs. Computing and data centre cooling systems can be improved and waste heat should be used for example in district heating or direct heating of the surrounding buildings (Wahlroos et al., 2018; Oltmanns et al., 2020), although these solutions are typically beyond the modeller's control.

In addition to environmental costs, there are non-negligible financial costs to modelling. Access to high-performance machines can be expensive and a heavy entry in a modeller's budget. Moreover, the often big data that results from running numerical models needs to be stored, diagnosed, visualised, and shared. Large local or remote storage solutions, software licenses, and powerful personal computers are expensive. These financial modelling costs need to be acknowledged not only by modellers themselves but also by others, such as funding agencies. With conscious management of resources, software and data, we can ensure a fairer, more efficient and greener geodynamic modelling community.

## 10 Conclusions & outlook

Geodynamic modelling studies provide a powerful tool for understanding processes in the Earth's crust, mantle, and core that are not directly observable. However, for geodynamic modelling studies to make significant contributions to advancing our understanding of the Earth, it is of paramount importance that the assumptions entering the modelling study and their effects on the results are accurately described and clearly communicated to the reader in order for them to be well understood. These assumptions are made at numerous stages of the numerical modelling process such as choosing the equations the code is solving, the numerical method used to solve them, and the boundary and initial conditions in the model setup.

Apart from acknowledging the assumptions made and their implications, it is important to view a modelling study in light of its intended philosophy. Generic modelling studies, usually characterised by extensive parameter variations, aim to understand the general physical behaviour of a system. Specific modelling studies on the other hand aim to reproduce a specific state of a specific geodynamic system and therefore rely more heavily on data comparisons.

In order to make the geodynamic modelling process transparent and less prone to errors, good software management is necessary with easily repeatable code verification to ensure that the equations are solved correctly. Additionally, it is important that the model is internally consistent with regards to thermodynamics and boundary conditions. Then, for individual models, the results need to be validated against observations.

When communicating the results of a geodynamic modelling study to peers, it is important to provide both quantitative and qualitative analyses of the model. Fair presentation of the results requires clear, unbiased, and inclusive visualisation. The results should first be objectively described and presented, after which they can be interpreted in the discussion.

In addition to outlining these best practices in geodynamic numerical modelling, we have shown how to apply them in a modelling study. Taking these best practices into account will lead to clearly communicated, unambiguous, reproducible geodynamic modelling studies. This will encourage an open, fair, and inclusive research community involving modellers, collaborators, and reviewers from diverse backgrounds. We hope to set a standard for the current state of geodynamic modelling that scientists can build upon as future research develops new methods, theories, and our understanding of the Earth. Geodynamic modelling is bound to increasingly link with a growing number of disciplines, and we trust that the perspective presented here will further facilitate this exchange.

*Data availability.* All figures presented in the manuscript, including light, dark, and transparent versions in various file formats, are available at s-ink.org/tag/101-modelling.

## Appendix A: Example: Discretising the heat equation

In this appendix, we provide an example of how to translate a physical model into a numerical model through discretisation, such that it can be coded up into an algorithm. More in-depth details on numerical modelling can be found in Zhong et al. (2015); Simpson (2017); Gerya (2019); Turcotte and Schubert (2014). For this example, we consider a simplified version of the energy conservation equation (6), i.e., the one-dimensional, transient (i.e., time-dependent) heat conduction equation without additional heat sources:

$$\rho C_p \frac{\partial T}{\partial t} = \frac{\partial}{\partial x} \left( k \frac{\partial T}{\partial x} \right), \tag{A1}$$

where $\rho$ is the density, $C_p$ the isobaric heat capacity, $k$ the thermal conductivity, and $T$ the temperature (Section 2.1.3). If we assume the thermal conductivity, density, and heat capacity to be constant over the model domain, the equation can be

simplified to the heat equation (16):

$$\frac{\partial T}{\partial t} = \kappa \frac{\partial^2 T}{\partial x^2}, \tag{A2}$$

where $\kappa = k/\rho C_p$ is the **thermal diffusivity** (Section 3.2). We want to solve this partial differential equation in time and 1-D space with appropriate boundary conditions using the finite difference method (Section 3.1) on a domain from $x = 0$ to $x = L_x$. Note that this appendix is a simple introduction to and example of the finite difference method and the reader is referred to

Gerya (2019) for a more thorough presentation of the method.

## A1    Taylor series of functions

Before discretising the heat equation, we need to approximate all its terms, after which we can discretise these approximations. Both first-order and second-order derivatives are present in the heat equation (A2), which we can approximate by Taylor series in the finite difference method. Here, we briefly show how to approximate a general function $f(x)$ that is continuous and

differentiable over the range of interest. We assume that we know the value $f(x_0)$ and all the derivatives $\partial^n f/\partial x^n$ at $x = x_0$. Then, in Section A2 and A3 we apply this to the specific terms of the heat equation.

### A1.1    First-order derivative

The forward Taylor-series expansion of $f(x)$, away from the point $x_0$ by a small amount $h$, is given by

$$f(x_0 + h) = f(x_0) + h\frac{\partial f}{\partial x}(x_0) + \frac{h^2}{2!}\frac{\partial^2 f}{\partial x^2}(x_0) + \cdots + \frac{h^n}{n!}\frac{\partial^n f}{\partial x^n}(x_0) + \mathcal{O}(h^{n+1}). \tag{A3}$$

This can be rewritten as

$$\frac{\partial f}{\partial x}(x_0) = \frac{f(x_0 + h) - f(x_0)}{h} + \mathcal{O}(h), \tag{A4}$$

where the truncation error $\mathcal{O}(h)$ indicates that the full solution would require additional terms of order $h$, $h^2$, and so on like in Eq. (A3). Hence, we have an approximation for a first-order derivative, which is formally defined as

$$\frac{\partial f}{\partial x}(x_0) = \lim_{h \to 0} \frac{f(x_0 + h) - f(x_0)}{h}. \tag{A5}$$

The fact that $h$ does not actually go to zero in Eq. (A4) and remains finite gives the finite difference method its name. Eq. (A4) is often called the forward derivative, as we can also expand the Taylor series backward (i.e. looking 'left' of $x_0$, at location $x_0 - h$) and the backward finite difference derivative then writes:

$$\frac{\partial f}{\partial x}(x_0) = \frac{f(x_0) - f(x_0 - h)}{h} + \mathcal{O}(h). \tag{A6}$$

### A1.2    Second-order derivative

To approximate second-order derivatives, we start with the Taylor expansions of function $f(x)$ at locations $x_0 + h$ and $x_0 - h$:

$$f(x_0 + h) = f(x_0) + h\frac{\partial f}{\partial x}(x_0) + \frac{h^2}{2!}\frac{\partial^2 f}{\partial x^2}(x_0) + \frac{h^3}{3!}\frac{\partial^3 f}{\partial x^3}(x_0) + \cdots + \frac{h^n}{n!}\frac{\partial^n f}{\partial x^n}(x_0) + \mathcal{O}(h^{n+1}), \tag{A7}$$

$$f(x_0 - h) = f(x_0) - h\frac{\partial f}{\partial x}(x_0) + \frac{h^2}{2!}\frac{\partial^2 f}{\partial x^2}(x_0) - \frac{h^3}{3!}\frac{\partial^3 f}{\partial x^3}(x_0) + \cdots + \frac{(-h)^n}{n!}\frac{\partial^n f}{\partial x^n}(x_0) + \mathcal{O}(h^{n+1}). \tag{A8}$$

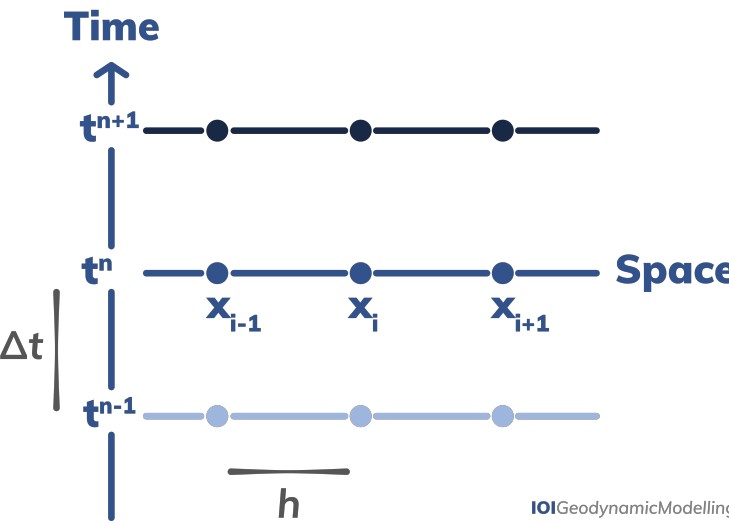

**Figure A1.** 1-D discretisation in space (horizontal axis) and time (vertical axis).

Adding these two equations together yields

$$\frac{\partial^2 f}{\partial x^2}(x_0) = \frac{f(x_0 + h) - 2f(x_0) + f(x_0 - h)}{h^2} + \mathcal{O}(h^2), \tag{A9}$$

which is an approximation of a second-order derivative.

### A2   Space discretisation

Now that we know how to approximate first-order and second-order derivatives, we can apply these approximations to Eq. A2. First, we start with the spatial discretisation for which we need the approximation of the second-order derivative. We want to solve the heat equation on a 1-D domain that is divided into separate parts (i.e., discretised). For simplicity, we will focus on
three consecutive, discretely spaced points (Figure A1). Using Eq. (A9), we can compute the second-order derivative of $T$ at point $x_i$ assuming we know the values of $T$ at $x_{i-1}$ and $x_{i+1}$:

$$\frac{\partial^2 T}{\partial x^2}(x_i) = \frac{T_{i+1} - 2T_i + T_{i-1}}{h^2} + \mathcal{O}(h^2), \tag{A10}$$

where $T_i = T(x_i)$, i.e., $T$ at $x = x_i$, and $x_{i+1} = x_i + h$ (i.e., $h = x_{i+1} - x_i$), where the node spacing, or resolution, $h$ is assumed to be constant.

### A3   Time discretisation

The next step is to discretise the first-order time derivative in Eq. A2, using the approximation of the first-order derivative (Section A1.1). To discretise time we divide it into discrete intervals of time, i.e., the time step $\Delta t$ is the time between two con-

secutive measurements (Figure A1). The time step is the equivalent of the grid size $h$ in the spatial discretisation (Section A2). In order to distinguish the indices relative to space and time, in what follows we adopt the convention that the subscript refers to space indices while the superscript refers to time indices. Using Eq. (A4), we can compute the first-order forward derivative of $T$ at $x = x + i$ and at time $t^n = n\Delta t$ as an approximation:

$$\frac{\partial T}{\partial t}(x_i, t^n) = \frac{T_i^{n+1} - T_i^n}{\Delta t} + \mathcal{O}(\Delta t). \tag{A11}$$

This forward finite difference derivative is called first-order accurate, which means that a very small $\Delta t$ is required for an accurate solution. The backward derivative from Eq. (A6) is then:

$$\frac{\partial T}{\partial t}(x_i, t^n) = \frac{T_i^n - T_i^{n-1}}{\Delta t} + \mathcal{O}(\Delta t). \tag{A12}$$

### A3.1 Explicit formulation

Both $n$ and $i$ are integers; $n$ varies from 0 to $nstep - 1$, where $nstep$ is the total number of time steps, and $i$ varies from 0 to $np - 1$, where $np$ is the total number of grid points in $x$-direction. When the forward derivative of Eq. (A11) is used for the time derivative and coupled with the spatial derivative of Eq. (A10), the following approximation of Eq. (A2) is obtained:

$$\frac{T_i^{n+1} - T_i^n}{\Delta t} = \kappa \frac{T_{i+1}^n - 2T_i^n + T_{i-1}^n}{h^2} \tag{A13}$$

which can be rearranged to

$$T_i^{n+1} = T_i^n + \Delta t \, \kappa \frac{T_{i+1}^n - 2T_i^n + T_{i-1}^n}{h^2}. \tag{A14}$$

Hence, we have found an expression to compute the temperature $T_i^{n+1}$ at point $x_i$ for the next time step $n + 1$ from all known values at the current time step $n$. Such a scheme is called an explicit finite difference method which we arrived at through our choice of evaluating the temporal first-order derivative with forward differences (Section A1.1).

### A3.2 Implicit formulation

An alternative approach to deal with the time discretisation is an implicit finite difference scheme, where we use the backward difference for the time derivative (Eq. A12). Together with the spatial derivative of Eq. A10, we then obtain

$$\frac{T_i^n - T_i^{n-1}}{\Delta t} = \kappa \frac{T_{i+1}^n - 2T_i^n + T_{i-1}^n}{h^2}. \tag{A15}$$

This is often rewritten as follows in order to deal with the unknowns of time step $n + 1$ instead of the known time step $n$:

$$\frac{T_i^{n+1} - T_i^n}{\Delta t} = \kappa \frac{T_{i+1}^{n+1} - 2T_i^{n+1} + T_{i-1}^{n+1}}{h^2}. \tag{A16}$$

The main advantage of implicit methods over their explicit counterpart is that there are no restrictions on the time step, since the fully implicit scheme is unconditionally stable. Therefore we will use the backward (implicit) scheme for the rest of this

example. This does not mean that it is accurate no matter what, as taking large time steps may result in an inaccurate solution for features with small spatial scales. For any application, it is therefore always a good idea to do a convergence test, i.e., to check the results by decreasing the time step, until the solution does not change anymore. Similarly, a spatial convergence check where the solution of the model is evaluated with changing spatial resolution is useful. Doing these convergence tests evaluates if and ensures that the method can deal with both small- and large-scale features robustly.

Eq. (A16) can be rearranged as follows:

$$-\frac{\kappa\,\Delta t}{h^2}\,T_{i+1}^{n+1} + \left(1 + 2\frac{\kappa\,\Delta t}{h^2}\right) T_i^{n+1} - \frac{\kappa\,\Delta t}{h^2}\,T_{i-1}^{n+1} = T_i^n. \tag{A17}$$

In contrast to the explicit formulation, we no longer have an explicit relationship which allows us to compute $T_i^{n+1}$ one by one by looping over index $i$. In other words, Eq. (A17) contains more than one unknown. Therefore, we need to combine these expressions for all unknown points in space and solve the resulting linear system of equations.

## A4   Obtaining the linear system of equations

We discretise the domain of length $L_x$ with 4 cells, i.e. $i = 0, \ldots 4$ ($np = 5$). Since we have a second-order derivative in space, we need to prescribe two boundary conditions. We choose $T(x=0) = T_0 = 0$ and $T(x=L_x) = T_4 = 100$. For simplicity, we assume that they do not change over time, so we omit the superscript. Finally, we assume that we know the initial temperature $T_i^0$ for all locations $i$, i.e. initial conditions have been provided. We want to compute the solution at time $t = \Delta t$, or $T_i^1$. To simplify notations we define the dimensionless parameter $s = (\kappa\,\Delta t)/h^2$ such that we get the following 5 equations, i.e., one for each point of the grid:

$$
\begin{aligned}
T_0^1 &= 0 \\
-sT_2^1 + (1+2s)T_1^1 - sT_0^1 &= T_1^0 \\
-sT_3^1 + (1+2s)T_2^1 - sT_1^1 &= T_2^0 \\
-sT_4^1 + (1+2s)T_3^1 - sT_2^1 &= T_3^0 \\
T_4^1 &= 100
\end{aligned}
\tag{A18}
$$

This system of equations can be rewritten in matrix and vector form, to obtain the general expression for the linear system of equations we are solving (Section 3.2):

$$
\underbrace{\begin{pmatrix}
1 & 0 & 0 & 0 & 0 \\
-s & 1+2s & -s & 0 & 0 \\
0 & -s & 1+2s & -s & 0 \\
0 & 0 & -s & 1+2s & -s \\
0 & 0 & 0 & 0 & 1
\end{pmatrix}}_{A}
\cdot
\underbrace{\begin{pmatrix}
T_0^1 \\ T_1^1 \\ T_2^1 \\ T_3^1 \\ T_4^1
\end{pmatrix}}_{X}
=
\underbrace{\begin{pmatrix}
0 \\ T_1^0 \\ T_2^0 \\ T_3^0 \\ 100
\end{pmatrix}}_{b}
\tag{A19}
$$

where $A$ is the coefficient matrix, $X$ is the unknown solution vector, and $b$ is the known right-hand-side vector. As opposed to the explicit approach, the linear system has a size given by the total number of nodes or points $np$. $A$ is a **sparse matrix** matrix and it is symmetric (i.e. $A = A^T$). In fact, even for very large values of $np$, only the diagonal and two off-diagonal lines contain non-zero values. These characteristics of the matrix can be exploited to efficiently solve the system of equations to eventually obtain a solution (Section 3.4).

*Author contributions.* This work is based on both the European Geosciences Union (EGU) Geodynamics blog and the annual EGU Geodynamics 101 short courses at the EGU General Assembly. IvZ and AP conceived the EGU geodynamics blog; IvZ, AP, JD, and AG conceived the EGU Geodynamics 101 short courses on geodynamic numerical modelling. In both instances, IvZ took the lead on realising these initiatives. All authors actively contributed to the EGU Geodynamics blog over the years and were involved in designing and teaching the EGU short courses on numerical methods. FC instigated this manuscript and designed the figures. All authors contributed equally to the content, writing, and revision of this manuscript.

*Competing interests.* The authors have no competing interests.

*Disclaimer.* We attempt to limit the total number of references presented in this work to increase readability. We acknowledge this does not represent the full extent of work done on any given topic. However, we refer to well-known review papers and textbooks with extensive explanations as well as exemplary papers from early career scientists from diverse backgrounds to further promote equality, diversity, and inclusion in geodynamics.

*Acknowledgements.* We would like to thank executive editor Susanne Buiter and topical editor Taras Gerya for supporting and encouraging the submission of an educational review paper like this. We would also like to thank our reviewers Paul Tackley, Boris Kaus, and Laurent Montési for extensive and detailed constructive reviews that greatly improved this manuscript. Similarly, we would like to thank everyone who provided additional comments during the open discussion on Solid Earth, particularly John Hernlund and Paul Pukite.

We warmly thank Antoine Rozel, who was an integral part of the original EGU GA Geodynamics 101 short courses and helped shape the format. We are deeply grateful to EGU - and in particular their communication officers Laura Roberts Artal, Olivia Trani, and Hazel Gibson - for the possibilities they provided us in the form of the EGU Geodynamics blog and the short courses as well as for their continued support. We thank the attendees of the short courses for their constructive feedback. Thanks to all our proofreaders for their valuable feedback: Ruth Amey, Molly Anderson, Kiran Chotalia, Tim Craig, Matthew Gaddes, Rene Gassmöller, Edwin Mahlo, Martina Monaco, Gilles Mercier, Arushi Saxena, and Jamie Ward.

IvZ was funded by the Royal Society (UK) through Research Fellows Enhancement Award RGF\EA\181084. IvZ also acknowledges the financial support and endorsement from the DLR Management Board Young Research Group Leader Program and the Executive Board Member for Space Research and Technology. FC acknowledges support from the Research Council of Norway through its Centres of

Excellence funding scheme, Project Number 223272. AG was supported by the Helmholtz Young Investigators Group CRYSTALS (grant no. VH-NG-1132). JD was partially supported by the National Science Foundation Award no. EAR-1925677.

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
