# Peer review of "Geodynamic modelling: How to design, interpret, and communicate numerical studies"

_Solid Earth, 2021_

## Referee Comment (RC1)

**Review of van Zelst et al. https://doi.org/10.5194/se-2021-14**
*Paul Tackley 12.05.2021*

This is a well-written and useful article that will serve as a broad introduction to newcomers in numerical modelling. It seems very long for an SE article but as a result fairly complete; there's always a balance between length and level of detail. All that is left for a reviewer to do is point out minor corrections and suggested additions.

As a general point, many points are supported by somewhat random citations (i.e. there are many articles that could be cited, they pick just one or two) while others are supported by exhaustive lists. The somewhat random citations could do with an (e.g. ) around them to show that they are not supposed to be complete lists.

l 35-37: Solomatov and Reese (2008) is a good reference to support the importance of grain dynamics, but not magma dynamics. How about (e.g. Keller et a. 2013) to support this.

l 109-110: For this citation an (e.g. …) is certainly needed because there are a huge number of 3D seismic tomography studies dating back to the pioneering works of Dziewonski (e.g. Dziewonski, 1984) - it's really random which ones are chosen here.

l 111-113. Adding some earlier references here might be appropriate, otherwise readers get the impression that these things were developed recently. Adjoint methods: (Liu and Gurnis, 2008; Burstedde et al., 2009). Data assimilation: (Bunge et al., 2003; Ismail-Zadeh et al., 2004; Hier-Majumder et al., 2005).

l 140: Update Ricard "Treatise on Geophysics" reference to the 2nd edition?

l 146: Give a more complete/rigorous definition of a continuum. For example something like what it says in Wikipedia "… certain physical phenomena can be modeled assuming the materials exist as a *continuum, meaning the matter in the body is continuously distributed and fills the entire region of space it occupies*. A continuum is a body that can be continually sub-divided into infinitesimal elements with properties being those of the bulk material."

l 183-185. Curbelo et al. (2019)'s relaxation time analysis doesn't apply to the mantle because they are considering an **ideal gas**, i.e. with the ideal gas equation of state (this is where their equation 4.4 comes from), which does not include elasticity. If you do the same analysis for a visco-elastic material like rock using an appropriate equation of state such as Birch-Murnaghan, then elasticity will be the dominant mechanism to even out pressure variations, i.e. if you apply a pressure perturbation in one place then it will propagate not via viscous relaxation but via elastic (seismic) waves, which are relatively fast. Of course even with seismic waves there is a component of viscous relaxation - this is why there is attenuation - so it would be possible to derive a time-scale for elastic waves to die out due to attenuation. A related point: compression of rock as it descends through the mantle does not occur by viscous relaxation, it occurs by elastic compression.

l 248: This simplification is only correct if density and Cp are constant. (as the authors note later, but it should be stated here as well).

l 264-265: Here their expression of $S_2$ is the general case of pressure varying in any direction whereas what they write is "…the dominant pressure variation…is the effect of the lithospheric pressure increase with depth". In this case the expression can be simplified: $\nabla p = \rho g \hat{\vec{z}}$ where z=vertical, positive downwards, hence $S_2 = \alpha T \rho g v_z$. This is the expression that is normally used in mantle convection codes.

l 289. When considering the various approximations, the order-of-magnitude fractional density error that is expected from ignoring thermally-related density variations is $\alpha \Delta T$, which is O(1%) ($\alpha$ ~10⁻⁵, dT~10³), so errors of this magnitude are considered "par for the course". This is the magnitude of error you get in thermal boundary layers / slabs / plumes with the Boussinesq approximation, and also with the Anelastic Liquid Approximations. It does not "invalidate" either

approximation, it is rather the magnitude of error that is accepted in making the approximation. (an aside: the Anelastic Approximation *is* invalidated by large T deviations from the adiabat, this is why the Anelastic *Liquid* Approximation is preferred for mantle dynamics).

l 294-295. In the convecting mantle, the magnitude of error in ignoring dynamic pressure in the density calculation, hence the resulting energy imbalance, is very small for realistic Earth parameters.
- The magnitude of dynamic pressure can be estimated by how much dynamic topography is generated by mantle flow: ~a few km. Compare this to the mantle depth of ~3000 km - the pressure error (dynamic/lithostatic) is thus in the range 0.1-1%.
- Leng and Zhong (2008) found resulting energy imbalances of up to ~few % because their experiments were at low Rayleigh numbers of $10^4$-$10^5$. Stress and dynamic pressure decrease with increasing Ra as roughly $Ra^{-1/3}$ (assuming that Ra increases because viscosity decreases) so extrapolated to Earth-like Ra of $10^7$-$10^8$, the expected energy imbalance is less than 1%.
- In deciding whether this small error is worth doing something about (i.e. using ALA instead of TALA), one must also consider the accuracy of the numerical pressure solution. For example, a well-documented issue with the finite volume (staggered-grid finite difference) discretisation is artificial pressure overshoots at viscosity jumps (e.g. Deubelbeiss and Kaus, 2008). If large, localised viscosity contrasts exist inside the modelled mantle, this "numerical" dynamic pressure is not something that one would want to use in calculating physical properties - it might result in larger errors than simply ignoring dynamic pressure.

l 308: For the extended Boussineq approximation it is not correct that "adiabatic heating leads to artificial generation of energy in the model".
(i) Adiabat heating removes energy from the system, it does not generate energy. "Heating" is a bit of a misnomer. This is because "heating" only applies to sinking material: rising material cools. Furthermore, rising material cools at a more rapid rate than sinking material is heated, because the adiabatic gradient is proportional to T, i.e. hot adiabats are steeper than cold adiabats. There is an equal amount of rising material and sinking material (mass conservation). The result of all this is that cooling of rising material exceeds heating of sinking material, so the AH term removes heat from the system. In equilibrium, heat loss due to adiabatic heating is exactly balanced by heat input due to viscous dissipation: the volume-integrals of the two terms are equal and opposite (e.g. Jarvis and McKenzie, 1980).
(ii) I did a quick test EB calculation using StagYY: 1x1 box, Ra=$10^5$ (based on total temperature drop), all properties constant=1, dissipation number=1. This is large Di - it means temperatures increase by a factor of exp(1.0)=2.7 from top to bottom - more than in the Earth. Despite this, integral(adiabat heating) = -integral(viscous dissipation), top heat flux = bottom heat flux, there is no energy imbalance.

```
Viscous dissipation: min =  3.52E-03 ; mean =  3.96E+00 ; max =  4.24E+01
Adiabatic heating  : min = -1.33E+02 ; mean = -3.96E+00 ; max =  9.63E+01
   Top flux and Nu =    4.860    4.860 ; Bot flux and Nu =    4.860    4.860
```

l 340: Use passive tense: "writes" -> "is written"

l 426: "strain rate increases" -> "strain or strain rate increases"

l 459 Either no comma or two commas: "…other variables like chemical composition…" or "other variables, like chemical composition, is…".

l 519-520. There is some confusion nowadays over the difference between finite difference and finite volume discretizations.
- The staggered-grid ("conservative") finite difference discretisation used in codes like StagYY, LaMem, I*VIS etc. is an example of a finite volume discretisation, and is normally referred to as such in the broader numerical simulation community and in many papers in our community (e.g. Ogawa et al. 1991; Trompert and Hansen, 1996; Shahnas et al. 2011). So, staggered-grid finite differences = finite volume, but
- it is also possible to have non-staggered grid finite differences that cannot be described as finite volume (e.g. several of the codes in Blankenbach et al. 1989),
- or unstructured-grid finite volume codes that cannot be described as finite difference (e.g. Hüttig and Stemmer, 2008).
- In conclusion, I suggest adding a clarification sentence, for example "We note that the commonly-used staggered grid finite difference discretisation is an example of a finite volume discretisation".

l 556-557: This explains the "difference" part of "finite difference"; why not also explain the "finite" part, which comes from the mathematical definition of a derivative as being a limit as h, the

difference in coordinate, tends to 0:
$$\frac{df}{dx} = \lim_{h \to 0} \frac{f(a+h) - f(a)}{h}$$

being replaced by a formula in which h is **finite**:
$$\frac{df}{dx} \approx \frac{f(a+h) - f(a)}{h}$$

l 584: $10^24$ -> $10^{24}$.

l 606-607: The most common iterative method used in geodynamic codes is the **multigrid method**. This is what is used in ASPECT, CitCom, StagYY, I3ELVIS, LaMEM, TERRA, etc.

l 620: "top 500" -> "TOP500 list (https://www.top500.org)"

Figure 5: A nice figure, but in the MPI part, bottom right, the 4 processors are all on the same node so actually MPI is not necessary - OpenMP could be used instead. It might be more illustrative to have/use only 1 CPU per node, so that the different nodes are communicating over the network.

l 664-667: The sentences on advection methods need to be rewritten/expanded/clarified.
(i) A distinction should be made between methods designed to treat discontinuities such as a free surface, and methods designed to treat smoothly-varying fields such as temperature.
(ii) The methods they mention (level set, marker chain, volume of fluid) are designed to treat discontinuities. Actually these work well and are widely used. I don't think there's anything "notoriously difficult" here. If all you need to track is one discontinuity, then using one of these methods makes more sense than placing particles everywhere in the domain.
(iii) Tracking temperature or other smoothly-varying fields is easier and many methods have been developed over many decades to advect fields while minimising artefacts such as numerical diffusion and dispersion (ripples). These methods work well and are in common use in a variety of fluid dynamics fields including mantle dynamics. For example, finite-element codes ASPECT and Citcom* use the streamline upwind/Petrov-Galerkin (SUPG) method. For finite-volume codes a variety of methods are available such as TVD (Total Variation Diminishing), FCT (Flux-Corrected Transport) and MPDATA (Multidimensional Positive Definite Advection Transport Algorithm), to name a few, and all of these are conservative. Again, if all you need to track is a smoothly-varying

field like temperature, it makes more sense to use one of these methods than to fill the domain with particles.

(iv) For compositional variations that exist everywhere in the domain, or multiple composition fields, it's best to use particles, although you can use one of the methods in (iii) - there isn't a single "compositional fields method" here - actually many possibilities. Some of the field-based methods are tested against particle methods in van Keken et al. (1997) and Tackley and King (2003).

l 670: Other disadvantages of particle-based methods are the introduction of artificial noise and the lack of conservation of advected quantities (when averaged to the grid).

l 670: Particle-based methods are not difficult to parallelise: each process holds the grid cells and particles in its subdomain and then each time step after advection, particles that have crossed to other subdomains are communicated to those subdomains. The only potential issue comes when the subdomains have different volumes, for example as a result of adaptive grid refinement - this can lead to load imbalance (i.e. different #particles in different subdomains).

l 726: Another useful example is Kramer et al. (2020).

l 745: Two more useful community benchmark papers: Travis et al. (1990), which was the US equivalent of Blankenbach et al (1989), and Busse et al. (1994), one of the few 3D benchmark papers.

l 746: Zhong et al. (2008) is not really a community benchmark because it is only testing one code; we don't get an idea of how different codes/methods compare. If you're going to include this why not include Tackley and King (2003) where we at least tested 2 codes (and several methods of treating composition).

l 907-916: Mention that periodic boundary conditions are the natural choice for global simulations.

l 993: Initial conditions do not always determine the model outcome: in long-term simulations or ones obtaining steady-state solutions it can often be that the initial condition is "forgotten". Therefore it would be more accurate to write "initial conditions **can often** determine the model outcome".

l 1026: add Duretz et al. (2011) - for example this is what I am using. It's basically like Kaus et al. (2010) but for the finite volume discretisation.

l 1068 "percentage" -> "fractional"

Section 9: There are a lot of underlined words in this section. I get the impression they are supposed to be hyperlinks, but when I click on them (in Adobe Acrobat Reader) nothing happens. Replace them with normal referencing.

Appendix A: Could list a few more references for numerical modelling, especially ones explaining finite element methods - for example the Zhong et al Treatise chapter or the book by Simpson (2017).

————–-

REFERENCES

Bunge, H.-P., C. R. Hagelberg, and B. J. Travis (2003), Mantle circulation models with variational data assimilation: inferring past mantle flow and structure from plate motion histories and seismic tomography, Geophysical Journal International, 152(2), 280-301.

Burstedde, C., O. Ghattas, G. Stadler, T. Tu, and L. C. Wilcox (2009), Parallel scalable adjoint-based adaptive solution of variable-viscosity Stokes flow problems, Computer Methods in Applied Mechanics and Engineering, 198(21-26), 1691-1700.

Busse, F. H., et al. (1994), 3d convection at infinite Prandtl number in Cartesian geometry - a benchmark comparison, Geophys. Astrophys. Fluid Dyn., 75(1), 39-59.

Deubelbeiss, Y., and B. J. P. Kaus (2008), A comparison of Eulerian and Lagrangian numerical techniques for the Stokes equation in the presence of strongly varying viscosity, Phys. Earth Planet. Int., 171(1-4), 92-111.

Duretz, T., D. A. May, T. V. Gerya, and P. J. Tackley (2011), Discretisation errors and free surface stabilization in the finite difference and marker-in-cell method in geodynamic applications: A numerical study, Geochem. Geophys. Geosyst., 12(Q07004), doi:10.1029/2011GC003567

Dziewonski, A. M. (1984), Mapping the Lower Mantle - Determination Of Lateral Heterogeneity In P-Velocity Up to Degree and Order-6, Journal Of Geophysical Research, 89(B7), 5929-.

Huettig, C., and K. Stemmer (2008), The spiral grid: A new approach to discretize the sphere and its application to mantle convection, Geochem. Geophys. Geosyst., 9(2), doi: 10.1029/2007GC001581.

Ismail-Zadeh, A., G. Schubert, I. Tsepelev, and A. Korotkii (2004), Inverse problem of thermal convection: numerical approach and application to mantle plume restoration, Physics of the Earth & Planetary Interiors, 145(1-4), 99-114.

Jarvis, G. T., and D. P. McKenzie (1980), Convection in a compressible fluid with infinite Prandtl number, J. Fluid Mech., 96, 515-583.

Keller, T., D. A. May, and B. J. P. Kaus (2013), Numerical modelling of magma dynamics coupled to tectonic deformation of lithosphere and crust, Geophysical Journal International, 195(3), 1406-1442, doi:10.1093/gji/ggt306.

Kramer, S. C., D. R. Davies, and C. R. Wilson (2020, in review), Analytical solutions for mantle flow in cylindrical and spherical shells, Geoscientific Model Development, doi:10.5194/gmd-2020-194.

Liu, L., and M. Gurnis (2008), Simultaneous inversion of mantle properties and initial conditions using an adjoint of mantle convection, Journal of Geophysical Research: Solid Earth, 113(B8), n/a-n/a, doi:10.1029/2008jb005594.

Ogawa, M., G. Schubert, and A. Zebib (1991), Numerical simulations of 3-dimensional thermal convection in a fluid with strongly temperature-dependent viscosity, J. Fluid Mech., 233, 299-328.

Shahnas, M. H., W. R. Peltier, Z. Wu, and R. Wentzcovitch (2011), The high-pressure electronic spin transition in iron: Potential impacts upon mantle mixing, Journal of Geophysical Research: Solid Earth, 116(B8), B08205, doi:10.1029/2010jb007965.

Simpson, Guy (2017) Practical Finite Element Modeling in Earth Science using Matlab, Wiley, ISBN: 978-1-119-24862-0

Tackley, P. J., and S. D. King (2003), Testing the tracer ratio method for modeling active compositional fields in mantle convection simulations, Geochem. Geophys. Geosyst., 4(4), doi: 10.1029/2001GC000214.

Travis, B. J., C. Anderson, J. Baumgardner, C. W. Gable, B. H. Hager, R. J. O'Connell, P. Olson, A. Raefsky, and G. Schubert (1990), A benchmark comparison of numerical methods for infinite Prandtl number thermal convection in two-dimensional Cartesian geometry, Geophysical & Astrophysical Fluid Dynamics, 55(3-4), 137-160, doi:10.1080/03091929008204111.

Trompert, R. A., and U. Hansen (1996), The application of a finite-volume multigrid method to 3-dimensional flow problems in a highly viscous fluid with a variable viscosity, Geophys. Astrophys. Fluid Dyn., 83(3-4), 261-291.

van Keken, P. E., S. D. King, H. Schmeling, U. R. Christensen, D. Neumeister, and M. P. Doin (1997), A comparison of methods for the modeling of thermochemical convection, J. Geophys. Res., 102(B10), 22477-22495.

---

## Referee Comment (RC2)

**Review of van Zelst et al. 101 Geodynamic modelling: How to design, carry out, and interpret numerical studies**
*Boris Kaus, Mainz 1.6.2021*

This is a well-written paper that is likely to be very useful for people that are new to geodynamic modelling. Putting all of that together in one place is challenging and I congratulate the authors for doing a good job with it. Unsurprisingly it is rather long and feels more like a book (which partly explains why it took me so long to get the review back to you - apologies!).
This certainly deserved to be published in SE. Yet, I do have a number of comments which I believe would be good to address first. This can mostly be done with some rewriting, so I don't think it should take you a lot of time (yet, it may help to point new geodynamicists in the right direction).

**1) Modelling tools**
You discuss quite a bit about the open source modelling packages like ASPECT, Underworld or LaMEM.  Yet, missing from this discussion are alternative approaches that are based on smaller (sometimes one-page) scripts. Those are not full-blown modelling packages but rather simpler scripts that solve a particular problem very well (and fast). The classical example of that in geodynamics is the MILAMIN code (http://milamin.org) which remains one of the fastest codes to solve the incompressible Stokes equations on 2D unstructured meshes, and may be very helpful for those interested to solve for example, problems with viscous inclusions or interacting crystals. Other examples are the M2Di scripts which are concise matlab and julia codes  to solve viscoelastic problems, including regularized plasticity, available from https://bitbucket.org/lraess/m2di/src/master/. The most recent development in this direction is the ParallelStencil julia package (https://github.com/omlins/ParallelStencil.jl#stencil-computations-with-math-close-notation) which comes with many geodynamic examples.

Geodynamicists that are interested to work on technical developments as well, may find such scripts much easier to understand than the big software packages that can do it all. In fact, with the ParallelStencil julia package it is possible to write a very compact code that scales to very large parallel GPU-based supercomputers with almost no effort (provided you use a pseudotransient iterative solver approach). Other efforts (under development) allow calling PETSc and its staggered grid interface from julia. In my opinion such approaches may become increasingly important in the future as it allows PhD students to go from writing an experimental solver to a fully blown (parallel) production code in a rather straightforward manner. This will help to address new multiphysics problems, such as the coupling between reactions & deformation. Given the informative nature of your current paper it would be good if you can discuss these topics as well in your manuscript (and give some of the links above).

**2) Parameter sensitivity/controlling parameters**
Typical forward models used in many geodynamic applications indeed have a large amount of parameters (as you discuss around lines 116 and lines 845). Usually, such

sensitivity studies are done 'by hand' by modifying input parameters, making a model run and checking the difference with respect to a reference model. Yet, part of this can be done automatically by computing scaling exponents which directly show which of the parameters control the velocity at a certain point (as discussed in Reuber et al. 2018 Tectonophysics and used in  Reuber et al. 2018, Front. Earth Science). In case adjoint methods are used to compute the gradients this is even computationally extremely efficient, and gives you the sensitivity to all model parameters at the same time. This would go a long way in determining which of those are of first order importance and which are not. This method only gives the sensitivity of the model results for a particular timestep/geometry but for many of the cases we looked at so far this sensitivity did not change drastically during a model simulation making this a quite powerful techniques (provided adjoints are available).

I suppose that the reason that this is not yet more widely applied is that it is not yet implemented in many of the codes currently in use, but I can well imagine that this may change soon. It would be good to highlight this as it is a very useful and automatic way to map and reduce the model parameter space (section 5.2.2) and will help to reduce the number of required simulations and thus the CO2 emissions of a study (line 1455).

**3) Section 2.1.1: Mass conservation**
Later in the manuscript (and around line 179/180 & lines 203-205) you discuss bulk compressibility, poison ratio etc., but you don't show how that should be added to the mass conservation equation in equations (same in line 452). It would clarify matters if you can add this.

**4) Solution methods**
You discuss different solution methods in section 3.4, but what is completely missing is a discussion about multigrid preconditioners. Users of any 3D geodynamic code will run into having to use multigrid at some point or the other and will wonder why it sometimes does not converge (and sometimes does). As people new to geodynamic modelling are the target audience of this paper, it would be good if you can add a paragraph to discuss this (and why, for many lithosphere dynamics problems, it is important to have a coarse grid that still "feels" the main viscosity structure of the model and is thus not too coarse).

What I believe is also important to discuss are pseudotransient solvers (there is much recent work by Raess et al.) as they result in compact solvers that scale particularly well on GPU's systems (see link to the ParallelStencil julia package for sample codes).

**5) Particle-in-cell methods**
I agree that for lithosphere dynamics codes, the particle-in-cell method is the most popular one (and I don't really understand why you say it is difficult to parallelize). Perhaps you can explain here why that as, which is in my opinion because it is the easiest method to take things like phase transitions, history variables like strain as well as large deformations in a simple manner. Many alternative approaches have been suggested over the years, and many of those are good if dealing with a limited amount of phases (e.g. bubbles interacting with crystals can be well approximated with level

sets). Yet, somehow none of these other methods withstood the test of time, perhaps because they are not general enough.

**6) Multiphysics**

Multiphysics is indeed an important avenue of future research (perhaps even one of the most important ones in geodynamics). Yet achieving this by "code coupling" of different, unrelated, codes really only works if there is only a loose coupling between the physics (section 3.8). This is perhaps the case when coupling models of surface processes and lithosphere deformation. Other problems, on the other hand, require a much stronger coupling on the solver level which implies that new solvers should be developed. An example are the two phase flow equations that describe magma migration, which roughly consists in a Stokes-like and a Darcy-like problem. Getting the solution in an efficient manner cannot be done by taking a Stokes code and a separate Darcy code and coupling that using batch scripts. Instead, the coupled set of equations needs to be solved in a tightly integrated manner. Our knowledge on the individual systems is still useful, as we can use multigrid preconditioners that work well for Stokes as part of this, for example. With this in mind, PETSc developed the multi-physics framework (see papers by Jed Brown) as well as the recently developed DMStag interface which allows you to add a Darcy-like code to an existing Stokes solver in a straightforward manner. I think it is important that you clarify this here, as there have been too many fruitless attempts in the past already to do loose coupling of different codes (which sounds intuitively easier but has very strong limitations for more tightly coupled problems).

**7) Creep rheologies & modelling manuscripts**

You spend quite some space describing how to write a modelling paper, which is certainly quite useful for people that are new to the community and/or to the paper-writing business. Yet, after having taught a class for over 10 years in Mainz in which students are supposed to reproduce published geodynamic modelling papers with a different code, I think the two most commonly made mistakes are not described here.

a) The first one is that in many papers, the model parameters are incorrectly listed in the tables, perhaps because the units of say dislocation creep laws are non-intuitive (the prefactor has units of the form of $MPa^{-n} s^{-1}$ where n is the powerlaw exponent; if n>1, transferring this to $Pa^{-n} s^{-1}$ takes a bit of work). In addition, creep law experiments are usually given in terms of principal stresses, whereas geodynamic models need to have this in a tensor format, which involves correction factors (see the textbook of Gerya for a nice derivation). All these issues add up to make it quite difficult to fully reproduce the experiments unless the input scripts are attached. In most cases these are obvious typos, as employing the stated parameters results in wide spread drip tectonics, rather than stable subduction (as many generations of students in mainz have experienced), so it seems quite clear that it is actually correctly implemented in the code. Yet, whether the correction factor is taken into account or not is often less clear as it has a smaller effect. To me this points to a deeper underlying problem: there is currently no central database that collects all experimental creep rheology data in a format that is directly implementable in different geodynamic simulations. If we would

have that it would eliminate quite a few of those mistakes.

b) A second issue is that often not all parameters are listed and some info is missing (nor just in terms of the material parameters employed but also with respect to the numerical convergence criteria etc.). I suppose that the reason for this is that there are often so many model parameters that it is quite easy to overlook some. A potential way around that would be to develop scripts that automatically generates tables with model parameters from input scripts which would eliminate another source of mistakes (which is that it seems that often some parameters are forgotten to be described in detail). Ofcourse publishing the input script used to perform the simulations, as you point out, helps as well, but in that case it is still not fully guaranteed that the creep law used in one code are implemented in the same manner in a different code (while using the same correction factors).

I think it would be good if you can discuss these topics in the manuscript. To minimize the chance of mistakes in the future, developing standardized databases and automatized ways to create the parameter tables could be very helpful.

**Additional comments**
line-wise (mostly typos, with some longer comments)

l. 55:  One of the reference textbooks on analytical solutions is Turcotte and Schubert, which would be good to list here.  Similarly, the textbook of Neil Ribe (theoretical mantle dynamics) has lots of useful info as well.

l. 170: highly viscous fluid

l. 259: please define deviatoric stress in equations, and not just in words.

l. 263: you did not explain what D/Dt means (or give equations)

l. 269: The shear heating term should only involve the non-recoverable deviatoric strain rate components (that is, non-elastic) and not the full strain rate. I realize that you define strain rate only later in the paper and that for incompressible viscous rheologies this is equivalent. For compressible viscoelastic materials it is however not the same, so it seems important to point that out here.

l. 268: thin phase transitions? Not sure what you mean by that.

l. 349: It would be good if you can give the mathematical definition of how to go from total to deviatoric strain rate (which indeed simplifies to what you write here in the incompressible case).

l. 368: fH2O looks as if it could be several symbols; perhaps better to write as f_{H_2O}

l. 368; please indicate the units of all variables. In fact that is an issue throughout the manuscript and without specifying the units of parameters you employed, modelling results will not be reproducible and replicable…

l. 368/369: Can you add the units of all parameters?

l. 402: Plasticity can also be used for pressure-independent yield criteria. Examples are Griffith's criteria (tensile failure) or implementing an ultimate yield stress in geodynamic models. It's therefore better to remove 'pressure-dependent'

l. 410: equation 14 is actually not identical to equation 13, unless cohesion C has a different meaning in eq. 14 compared to eq. 13. We actually had a discussion about this before in Solid Earth, so please have a look at https://se.copernicus.org/articles/11/1333/2020/se-11-1333-2020-discussion.html
to see an illustration of the difference. The reason that cosine and sine terms appear in eq. 13 instead of $\tan(\mu_f)$ is that this is the yield stress function.

l. 416/417: If lithostatic pressure is used to evaluate the yield criteria, the shear band orientation is always 45 degree and there is no difference between compression and extension. I realize that many large-scale convection codes use that assumption, and assuming lithostatic pressure is fine within the mantle. Yet, within the stronger lithosphere there can be quite strong deviations between lithostatic and dynamic P (up to factor 2 under compression). It's probably good to point that out to the readers.

l. 445. An easy-to-follow explanation of how to implement anisotropy in 2D codes is given in Kocher et al. Tectonophysics 421, 71–87.

l. 558: A big difference between FD and FE methods is that in FD, you solve the partial differential equation in a pointwise manner, whereas FE approximate the equations on average per element. It would be useful to add a comment on that here.

l. 562: Perhaps add a small remark on why finite elements should use a mixed formulation with higher order for velocity compared to pressure to get reliable results in the (near)-incompressible limit.

l. 584: $10^{24}$ Pas

l. 740: the method of manufactured solutions also works for nonlinear problems which is perhaps good to mention here.

l. 745: It is perhaps interesting to mention that community benchmarks is a process that typically takes several years…

l. 963: geomio is spelled "geomIO"

l. 1026: Implicit timestepping, in which advection becomes part of the nonlinear solution step, is an even better method to deal with the drunken sailor effect (was described in Popov & Sobolev 2008, even if only very briefly).

Fig. 10: Maybe it is good to mention that the reason the drunken sailor effect occurs is that the typical density difference between rocks and air is much much larger than the typical density difference within the Earth. Moving the Earth's surface by one meter therefore causes a much larger stress perturbation than moving the Moho by a meter and that is why the models have a tendency to become unstable.

Fig. 10b: Even when you can 'fix' the wrong pressure field in this case by smoothing there are other, more heterogeneous, setups where such smoothing does not fully remove the artefacts. It is thus clearly better to employ LBB stable elements (like $Q_2P_{-1}$) throughout, The other undesired side-effect of unstable elements such as $Q_1P_0$ is that they require more iterations for a higher resolutions, if combined with iterative solvers. Stable elements fix that.

Fig. 10c: I am not sure why you claim that the higher-resolution model is better here. Ity seems that both models are performed with non-regularized plasticity so both are actually numerically non-convergent. On one hand this manuscript argues about the importance of reproducible and numerically trustworthy results. On the other hand you show a key example in geodynamics where this is not the case. In my opinion there is no way around using a form of regularized plasticity (together with sufficient resolution such that the plasticity length scale is captured) and it would thus be better to use an example of that within the current manuscript. There are some recent papers by Duretz et al. showing that this can be done, for example by using viscoplastic regularization, so it seems more appropriate to show examples of such computations in this figure.

l. 1047: Or better avoid unstable elements altogether and use stable ones (which also scale better on parallel computers using multigrid solvers)..

l. 1184: probabilistic

l. 1311: In the geosciences it is common practice to acknowledge the work that reviewers put in going through the manuscript and making suggestions. Not everyone in computational  geodynamics follows this (unwritten) rule, however, which I know doesn't go well with many of my colleagues (including myself). So perhaps it is good to spell this out here.

*Online supplement:*

**Byerlee's law:** Byerlee's law was originally derived for small-scale laboratory samples and is in itself quite amazing in that it shows that the maximum stress that rocks can

withstand is nearly independent on the rock composition. What is even better is that it is nearly perfectly consistent with in-situ stress measurements in drill holes around the world (the classical reference would be Townend and Zoback, 2000). This shows that we can safely upscale Byerlee's law from small samples to the upper crust. For other geodynamically relevant parameters, such as the effective viscosity of rocks such upscaling does not appear to be that easy. It would be good to point that our here (or in the main manuscript).

---

## Referee Comment (RC3)

**101 Geodynamic Modelling: How to design, carry out, and interpret numerical studies**

Review by *Laurent Montesi*, University of Maryland.

This manuscript serves as a well-designed guide for modeling mantle and crustal-scale processes. As is necessarily the case with an exercise of the sort, it does in place reflect the personal experience and opinions of the authors, but it overall does a very nice job of remaining neutral, and even the more engaged sections are full of important information that will be useful for moving the field forward. I particularly appreciated the discussions of figure accessibility and Section 9. There are also some very important discussions of the objectives of modeling, in particular the difference between "specific" and "generic" modeling. Both are presented as equally valuable, which is an important message to pass to both modeling and non-modeling communities.

Although the paper is currently well organized and well written, I do have a few suggestions that could lead to significant rewriting. They are organized here as topics and followed by several more isolated comments. Most suggestions can be seen as a matter of personal preference and should not stand in the way of the publication of the paper. Maybe they are best seen as a discussion of the material presented. I apologize that as my review is already much overdue, I did not study the appendix or glossary in detail.

1. Although the title sets the scene for "Geodynamic modelling" in general, the authors focus on the dynamics of the mantle and the crust. In place, they contrast their discussion with the core, especially in their neglect of inertia and Coriolis forces. There are other geodynamical settings where these are also neglected: the hydrosphere (especially the oceans) and the atmosphere, to say nothing of giant planets. The effects of fluids (including but not restricted to magma) are noted in passing, even though they are a growing segment of geodynamical studies. I do not think that presenting specific equations would be necessary to discuss modeling strategies and philosophies but as the authors have chosen to focus on a specific system, I believe that the title must reflect that choice. Maybe specify "Geodynamic Modelling of the Mantle and Crust" (lithosphere instead of crust would be OK too) and make this focus (including the neglect of two-phase flow) clear at the outset. The occasional mention of core studies can be seen as choosing a related science but neglecting other applications. Minor points related to this topic:
   - Line 20: The mention of "spherical shells" implies a global focus. Local and even regional studies do not need to consider shells.
   - Line 37: "magma dynamics and grain dynamics" The citation to Solomatov and Reese shows that you have global mantle convection in mind. This is a good paper but please also mention studies about magma dynamics, and also other fluids ($CO_2$, water) that have been shown to matter in volcanic and seismogenic systems, and maybe also serpentinization (important in mid-ocean ridges).
   - Line 80: I don't have the statistics to evaluate if applications to Earth's crust and mantle are indeed "the most common applications". I expect that people studying climate and atmosphere may have something to say about that. The concepts presented (except for the specific equations) are also applicable to their disciplines. It may be better to just say that these are the disciplines you worked on and therefore are most familiar with.
   - I hate making titles longer (believe it or not) but it strikes me that the important communication steps that are in the paper are not included in the title.
2. Figure 2 is key to the entire paper. It is well-thought-out and I am sure the authors have spent much time discussing it. I do have several issues with it as it stands.
   - It is easy to assume, based on Figure 2, that geodynamic modeling is a linear process. I believe this is an oversimplification. This paper would be a great place to emphasize instead the need to constantly reevaluate foundational hypotheses. Results and analysis, especially following evaluation against observations (necessary whether the model is "specific" or "generic") can

lead to updates in any of the previous steps. "Nature" should feed directly into the validation (maybe) and analysis (certainly) steps.

- What is missing in Figure 2 and section 1.2 is the mention of a hypothesis. I strongly believe that the most important models are those that were designed at the outset to answer a specific scientific hypothesis. As the paper is currently organized, it would be possible to write a code and verifying it before a hypothesis is identified (see line 127). To me, this is backward: we should design codes that enable us to answer a question, therefore the question needs to come first, not in the "setup" stage.

- This may be a matter of semantic, but it bothers me that "physical models" are described only using equations (also Line 87). I believe that physical models can be discussed in terms of concepts (e.g. buoyancy, inertia, diffusion, or the rheologies that are illustrated without equations in Figure 3) and that by the time you have introduced an equation, you have already moved on to the next level of abstraction: a mathematical model. That model can be solved analytically or numerically. Therefore I believe that an intermediate step ("mathematical model"?) is missing in Figure 2.

- Somewhat related to the previous, don't forget that chemical processes and increasingly considered in geodynamics models. You might argue that they are included in "physical processes" as thermodynamics can be seen as either, but issues of trace element partitioning (important for evaluating the origin of magma or fluid interaction) are typically discussed only in geochemistry. Like physical models, chemical models need a mathematical description to be included in the later numerical modeling efforts.

- This is not an issue, but maybe consider distributing your figure as an open workflow (e.g. on https://workflowhub.eu/)

3. Model simplification
   - You often use "simplified" to describe your model. This is perhaps a matter of preference but I favor "idealized". Complexity is not necessarily your enemy (and you do a good job discussing the pitfalls over oversimplification) but whenever you settle on mathematical relations, especially in the constitutive equations that are needed to close your balance equations, you are looking at an idealized but not necessarily simpler view of reality.
   - Line 789: the concept of over-simplification is not shown in Figure. Another issue is that what is "too much" simplification depends really on the hypothesis to be tested (that's a reason why the hypothesis needs to be identified first). The "simple" convection model in Fig.7 is also quite complex to me. The triangle heart may be sufficiently realistic for some purpose. The link between oversimplification and hypothesis (line 791) should a major motivation for motivating the identification of the hypothesis before designing the model (see my comments about Figure 2) Here again, I would favor idealized over simple, and maybe realistic over complex.
   - Figure 8 could mention constitutive equations that coupled vary from non-linear to linear to fixed parameters. The degree of coupling in Multiphysics problems could also be mentioned.

4. Line 247 etc. High thermal conductivity does not mean that the curvature $\nabla \cdot \nabla T$ is steep. Just consider a simple steady-state solution with fixed temperatures at both ends of a 1D solid. You know the temperature varies linearly between the two ends. Curvature is 0, regardless of the conductivity. What the high conductivity does is allow for large heat flow through that material. It also makes it faster (everything else being the same) to obtain this solution, so that in general, the curvature of the temperature field is less in the case with high k, at least in this specific example.

5. The description of physical concepts emphasizes the momentum equations. While I can appreciate their importance, most models also need to include constitutive equations. These appear in section 2.2 but should be at least mentioned earlier, as part of the model design. The fact that these evolutions, including their parameters, can change over time should also be described in general with an evolution equation, and that aspect is buried in "more complex processes" (section 2.3). It goes beyond the idea of fault gouge development (line 422). I don't think this is any more complex

than the other kinds of equations described. Many constitutive parameters evolve following an ODE, so in a way, this is simpler.

- Figure 3: As much as I like the emphasis on rheology, it strikes me that it is only one of the many constitutive equations that can enter the model. Perhaps it is more fundamental because, without it, there would be no way to link stress and velocity. In that case, it has to make the distinction clear but highlight in the caption that other constitutive and evolution equations can (should?) be considered. Minor point: the diagram for "plastic" is brittle.
- Line 224: Stokes equations need not only eq. 2 and 5 but also viscous rheology. If you use elasticity you end up with a different set of equations.
- Line 229-231: Seismic waves do need inertia but they do not follow the viscous relations (Stokes or Navier-Stokes equations) that you describe here.
- In detail, Equation (10) doesn't work as it takes a tensor to a power that is not necessarily an integer. This equation should be written with a scalar measure of the strain rate and the stress tensor. We often use it in models with an effective viscosity approach. I think it would be important to explain this in the text. By the way, H2O should be styled as subscript (and not italicized) in $f_{H_2O}$.
- Line 374: In fact, grain size dependence is fairly well constrained. Grain size evolution is not (but you do not consider evolution at this stage). Also, dependence on porosity ($\alpha^*\phi$) is even less well constrained.
- I find the discussion of elastic deformation to be a distraction: it cannot be included as is with the other relations and the framework of Stokes flow. We can see that in line 471, it is written that "velocity … enter the stress tensor". That is incompatible with elasticity. I recommended either skipping section 2.2.3 or describing implementations of viscoelastic materials.
- Line 480: Why the specific focus on diffusive quantities? You can have ODEs, e.g. grain size evolution.
- Line 505: the addition of Eq. 15 is inconsistent with the statement of introducing "the general concepts of geodynamics". I would argue that constitutive relations are more fundamental and general.

6. Section 2.2.4 needs more discussion of the way brittle failure can be implemented.
- Line 392: brittle deformation takes place when rocks break, but this is not necessarily (in fact rarely) a rupture of the crystal lattice.
- Line 394: Actually, it is possible to represent discrete faults using split nodes (Melosh and Raefsky, 1981, A simple and efficient method for introducing faults into finite element computations, Bulletin of the Seismological Society of America 71 (5): 1391–1400), slippery nodes (Melosh and Williams, 1989, DOI 10.1029/JB094iB10p13961) and augmented FEM or X-FEM approaches (Ortiz et al., 1987, DOI 10.1016/0045-7825(87)90004-1; Dolbow et al., 2001, DOI 10.1016/S0045-7825(01)00260-2)
- Line 398: there is a difficulty, here again, of comparing and tensor (stress) and a scalar (yield stress). This difficulty should be mentioned. It is particularly important as the difference between criteria that are later mentioned (e.g. Tresca vs. Von Mises) stands in the way that this comparison is made (stress invariant for Tresca and Drucker-Prager, which leads to a smooth yield envelope, vs. resolved shear stress for Tresca and Mohr-Coulomb, which leads to a segmented yield envelope).
- Line 410: Byerlee's law has two branches. You describe here the friction coefficient of the high-pressure branch but that branch also has a cohesion term. The intermediate branch (important in the crust) does not have cohesion but its friction coefficient is 0.85.
- Line 415: Drucker-Prager and Mohr-Coulomb cannot be equivalent. One is smooth and depends only on the first and second stress invariants, the second is segmented. For an elastic-plastic deformation, localization arises spontaneously at the corners of the yield envelope, so the difference is significant.

7. Section 3 misses some of the fundamental differences between FEM and FDM. As the authors are well aware, FEM uses the original equations (in their weak form) while FDM modifies these equations. Conversely, FEM restricts the solution to a predefined space. Therefore, the statement in line 510 that we are considering approximate solutions to Eq. 2, 5, and 6 is only strictly valid for FEM (FDM approximates the equations, not necessarily the solution)
    - Line 520: there are even approaches where one equation (energy?) is solved with FDM and the others by FEM.
    - Line 560 etc. The notation of $Q_1 \times P_0$ etc. the concept of "bi/trilinear velocity" must all be described here or in a glossary. I don't think it's reasonable for non-initiated readers to know what this means.
    - Line 576: ALE is only one of several semi-Lagrangian methods.
8. Modeling philosophy
    - The analogy between car/model vs. engine/code did not work for me. To me, the code is more general than the model. The model takes the code and restricts it if using the setup. The code may have additional capabilities that are not used in a particular model. The car also uses many other components that are not the engine (steering wheel, headlamp, seats, doors). I am sure that every modeler can come up with their preferred analogy (and you may argue that your analogy best fits your vision) but I might suggest that the code is like a road network and the model is like an itinerary that uses these roads to read a destination.
    - It would be useful in Section 5.2 to emphasize the importance of "failed" models. It is often impossible to conclude that a certain phenomenon is responsible for observations but it can be shown which situation does not explain observations (this is valid for both specific and generic models). Similarly, these failed models would not verify the criterion in Line 1085 of being "consistent with our understanding of geodynamic processes" but are important to probe the limits of our understanding (this may not be relevant for Phanerozoic Earth, but what about other circumstances?)
    - Line 847: add "the PROPOSED control mechanism(s)". I am not convinced that assumptions and simplification are necessary for ALL control mechanisms. Some models may even be designed to find the necessary level of simplification.
    - Line 851: Other examples of regime diagrams can be found in Citron et al. (2020, DOI 10.1029/2019GC008895) and Gülcher et al. (2020, DOI https://doi.org/10.1038/s41561-020-0606-1)
    - Line 856. Spiegelman focuses more on magma transport than on magma dynamics (often linked to eruptions). It would be good to also cite the newer work by Sim et al. (2020, https://doi.org/10.1016/j.pepi.2020.106486).
9. Boundary and Initial conditions
    - Line 898: prescribed stress can also be used to mimic topographic loads.
    - Line 910, 924: periodic BC are also commonplace in mantle convection dynamics, not just lithosphere dynamics (for example Gurnis, 1988; 10.1038/332695a0; Lowman et al., 2001 10.1046/j.1365-246X.2001.00471.x).
    - Line 925: why restricting the criticism to specific model setups? Generic models of long-term mantle convection are more realistic if the core temperature is allowed to vary according to its thermal balance.
    - Line 926: I think you need a citation to support your statement that models of the other core are better off prescribing heat flux boundary conditions.
    - Line 955: Maybe highlight the issue of initial strain rate for models with non-linear viscosity (there is no solution if velocity is 0 everywhere).
    - Line 965-6: Why specify the mantle when mentioning chemical heterogeneities. The crust is also highly heterogeneous.

- Line 979 paragraph. Note that weak sees can also be implemented as a random field of initial values, which is less constraining than a single seed (e.g. Jammes and Lavier, 2016, 10.1002/2016GC006399).

10. Unphysical behavior vs. numerical problems (e.g. Line 1005 paragraph).
    - I am not certain what "nonphysical artefacts" you have in mind. An incorrect IC or BC is not sufficient to have nonphysical behavior but the resulting result could be irrelevant for the geological problem at hand (still being physical). Maybe you mean that the IC or BC is incorrectly implemented (the verification step should have taken care of that). In some cases, there could be issues of convergence, but not necessarily. I would imagine that the issues you identify (resolution, drunken sailor effect) would be identified at the verification stage (are you solving the problem correctly) not at the validation stage (are you solving the problem you think you are solving)
    - Lines 1035-1038. Stabilization by diffusion is helpful, but one should also remember that it potentially changes the set of physics included in the model (you are changing the equations). Same thing for mass scaling in FLAC codes.
    - Lines 1045: smoothing is indeed important for $Q_1 \times P_0$ elements but I think the recommended solution should be to use stable elements instead. I always refer to the list in the Bathe (2014). *Finite Element Procedures* textbook (Table 4.8 in the second edition). Certainly, there are other references)

11. Modelling manuscript. I found section 8.1 to be overly prescriptive. It presents one possible manuscript organization, which is indeed very common and effective. However, this is far from the only solution (this manuscript does not follow the prescribed structure, for example). I also notice that the general papers about manuscript structure and methods come from the clinical literature, which could have different conventions than geodynamics. At a minimum, the wording must be toned down (should or may instead of must). I would go so far as recommending you delete Figure 11 and much of section 8.1. I always prefer flexibility over rigidity when planning a publication. Scientific manuscripts DO NOT have a rigidly defined structure (Line 1316) although I can recognize that a structure can be useful especially for the first papers one might write. More specific comments follow.
    - Modern publications tend to deemphasize the method (especially in general journals) and emphasize results and insight instead. In my opinion, it is a good thing and helps broaden the reach of our papers. The IMRAD structure that you describe is appropriate for highly specialized publications but not for generic ones. Non-specialist will gloss over the technical aspects and focus on the take-home message of the paper. I would certainly emphasize more the message (you have just one sentence on telling a story in line 1317, yet that is what controls the impact of the paper).
    - As the editor of a journal, I strongly disagree that "it is always good practice to write a complete methods section for every manuscript" (line 1271). If the focus of the paper is on the method, that's true. Otherwise, it is better to refer to other papers that have developed the method and maybe summarize that method in supporting information documents. This separation can strengthen the take-home message of the paper (and remember we study geodynamics to gain new insight). Also, it avoids having identical (or nearly-identical) method sections in different papers, which violates dual-publication policies. Finally, modern open science strategies require sharing codes in FAIR-enabling platforms (as you describe very well), in which case it should be possible to include a citation to the code and its version. It is to be expected that code publications will have documented verification and maybe validation steps. The paper can then focus on the hypothesis, setup, and results.
    - On a related topic, line 1231 "The methods section is considered one of the most important parts of any scientific manuscript" is more true of social and clinical sciences (the Kallet reference you include), which struggle with reproducibility due to reliance on human subjects

and survey methods, than it is in geodynamics modeling. Note also that "who performed the experiment" (line 1241) should be irrelevant: the models should be reproducible by anyone. I think again that the context of the cited paper (here Annesley, 2010) makes it irrelevant for our discipline.

- My other problem with IMRAD as described is that it links discussion and conclusions. In our disciplines, these sections have different purposes: one puts the results in a broader perspective, and may even speculate on future hypotheses, while the other summarizes the paper.
- Line 1239-1240 I don't know what journal specifies "how many words can be used" to describe methods, but certainly not the one I edit.
- Line 1288: The results section should describe the model results. Answering the central question or hypothesis of the paper should only happen after these results are analyzed and evaluated, which is best done as a discussion. This is in fact what you prescribe in Line 1297.
- A couple of pitfalls should be mentioned in the Line 1306 paragraph: the abstract should give a preview of the results as well as the work. Too often I see abstracts that say what is done without saying what is learned (which is what will inspire a broader audience to read the paper). Second: the best titles remain succinct (I was told to limit titles to ten major words). I know I suggested additions to your title but in my opinion, the title should also stop at the column. Two-part titles are often cumbersome. You may also want to mention the use of Plain Language Summary and Graphical Abstracts, for publishers that allow them, as alternative ways to engage a broad readership (note that the purpose and mechanics of Plain Language Summaries are not the same as those of abstracts).

12. Not surprisingly considering who the authors are, the section on visualization is very strong. I have a couple of points to make, though
    - Line 1323: Bar plot should have a well-justified baseline, but it does not have to be 0. Imagine that the quantity reported varied over several orders of magnitude. I may best to use a log scale, and 0 may not be the best reference (e.g. grain size varying from micron to millimeters).
    - Line 1341: there are circumstances when it is necessary to change the range of a color scale. I agree it typically should be avoided and if the range changes, that needs to be emphasized in the caption. However, saying the range should "always" be the same is overly prescriptive.
    - Perceptually uniform color maps are certainly to be preferred for an unbiased reporting of results. However, figures should also inform the readership, and it may be useful to take advantage of non-uniformity to highlight a result (10.1109/TVCG.2018.2855742). This is in a sense what is done with a multi-sequential color map (oleron, highlighting the sea level) and there are also tools for interactively creating colormaps (https://sciviscolor.org/colormoves/overview/)

13. Section 9 is also very strong and useful. It contains many important resources.
    - I wonder if the underlined words should be links, though. In that case, they would best included as URL citations (or better permanent citations if a DOI is available).
    - You may also want to point to CIG's best practice documents at https://geodynamics.org/cig/dev/best-practices/.
    - Line 1418. Another argument for storing and sharing numerical data is to enable the R of FAIR (Reusability). Other scientists may be interested in new analyses of the model runs for different purposes or quantitative comparison with other studies. Sharing results saves on the time and computational cost needed to reproduce the results.
    - Line 1430: Mention the Planetary Data System?
    - Line 1442: Mention Earth and Space Science journal (https://agupubs.onlinelibrary.wiley.com/journal/23335084)?
    - Line 1443: here I think you are not strong enough: research not only can but must create and use persistent identifiers whenever possible (and you do include great pointers on how to do this).

- Line 1446: note that ORCID iD is required by some journals.
14. Isolated points
   - Line 24: replace hundreds by thousands of km (the scale of the largest plates, or the "penetration" of plate boundary deformation in Asia or North America).
   - Line 35: "THEY take place"? (subject surface processes)
   - Figure 1: 1) make the axes labels darker? 2) what about aseismic transients and creep processes?
   - Line 55: While I am proud that you included my paper here, I would recommending mentioning the seminal papers of Hager and O'Connell (1981, https://doi.org/10.1029/JB086iB06p04843), McKenzie (1969 https://doi.org/10.1111/j.1365-246X.1969.tb00259.x). There has been action between 1879 and 2006…
   - Line 63: What do you mean by "the physical properties of the variables" (variables can have value and they can represent physical properties, but they don't have physical properties).
   - Line 97: I would regard boundary conditions as a part of the model setup, not simulations.
   - Line 112 etc. It would be good to mention data assimilation.
   - Line 130: I would think that it's the model results, not the setup, that need to be compared to observations.
   - Line 133: add "openly" to "clearly and reproducibly" (BTW: reproducibly, not reproducible)
   - Line 136: elasticity can be important at long time scales too. See plate bending. Its role in long-term tectonics is still debated.
   - Line 141: 450 years assumes a certain viscosity. I am quite certain it is longer when you enter the lithosphere (still in the mantle). Do you assume the asthenospheric mantle?
   - Line 162: I would focus on the physical processes that are of interest (not just relevant).
   - Line 169 misses a space in "viscousFluid".
   - Line 255: Definitions may be different for different people, but I believe that shear heating is the more general term (it does not imply a shear mechanism) and that it can be divided into frictional and viscous dissipation. This is opposite to the relation you include here.
   - Line 270: Melting is much more complex than shown here, as the heat can be transported by melt flow. You also have the issue that the fraction that has melted (X) may be different for the (retained) melt fraction.
   - Line 283: density OF the …profile.
   - Line 297-309: It may be worth mentioning that many studies have used the EBA but called it Boussinesq.
   - Line 458: why do you assume the relation between density and temperature is linear?
   - Line 584: typo: $10^{24}$. I think the shear forces should be negligible, but it's not true that there are no shear forces.
   - Figure 4 gives the impression that triangular meshes are necessary for conforming to an interface. You can have quad meshes that do the same. Maybe include one on the bottom right panel? Anyway, as currently designed, the panel with the two fluids appears in the "mesh" row, so that's a little confusing. Define the fluids in the Field Method panel. I also find the grey fonts too light (I honestly didn't see the words "mesh" and "method" when I first looked at the figure).
   - Line 640: the viscosity "MAY DEPEND" on the velocity… (it's not a requirement)
   - Line 719: Explain what you mean by an "as simple hello world" test.
   - Line 725: I know it's impossible to be exhaustive here, but I would like to see mentioned example of corner flow, viscous folding, and half-space cooling. The latter is particularly important as otherwise there would be no example using the energy equation. Folding also presents interesting numerical effects (see Schmalholz and Podlachikov, 1999, DOI 10.1029/1999GL900412)

- Line 744: missing words "It is important…"
- Line 759: Laboratory experiments do not have "infinite resolution" especially in tectonic applications with granular media. Also, even if all possible physics is indeed included, the constitutive relations are not always fully understood (Katz et al., 2005 Tectonic microplates in a wax model of sea-floor spreading, New J. Phys. 7 37, Di Giuseppe et al., 2012. https://doi.org/10.1007/s00397-011-0611-9). These limitations are important to mention as it highlights the complementary between numerical and analog models (in numerical studies, only selected sets of physics are included, but that means they are well controlled).
- I would have preferred to see the non-dimensional numbers of lines 1130-1 introduced with the concept of the regime diagram (Line 851).

---

## Author Comment (AC2)

**Geodynamics 101 Revision - Review Paul Tackley**

This is the authors' response to the review of Paul Tackley. Our responses are indicated in **green**, with the original review presented in black.

This is an "educational" paper; it is not presenting new science or techniques, but instead is introducing newcomers to geodynamical modelling. As such, it is very good, and I have only some corrections and suggestions detailed in the attached report.

**Review of van Zelst et al. https://doi.org/10.5194/se-2021-14**
*Paul Tackley 12.05.2021*

This is a well-written and useful article that will serve as a broad introduction to newcomers in numerical modelling. It seems very long for an SE article but as a result fairly complete; there's always a balance between length and level of detail. All that is left for a reviewer to do is point out minor corrections and suggested additions.

As a general point, many points are supported by somewhat random citations (i.e. there are many articles that could be cited, they pick just one or two) while others are supported by exhaustive lists. The somewhat random citations could do with an (e.g. ) around them to show that they are not supposed to be complete lists.

l 35-37: Solomatov and Reese (2008) is a good reference to support the importance of grain dynamics, but not magma dynamics. How about (e.g. Keller et a. 2013) to support this.

We added this reference.

l 109-110: For this citation an (e.g. …) is certainly needed because there are a huge number of 3D seismic tomography studies dating back to the pioneering works of Dziewonski (e.g. Dziewonski, 1984) - it's really random which ones are chosen here.

We added 'e.g.,' and added the reference to Dziewonski, 1984.

l 111-113. Adding some earlier references here might be appropriate, otherwise readers get the impression that these things were developed recently. Adjoint methods: (Liu and Gurnis, 2008; Burstedde et al., 2009). Data assimilation: (Bunge et al., 2003; Ismail-Zadeh et al., 2004; Hier Majumder et al., 2005).

We added these references, as well as Bunge et al., 2002 on data assimilation.

l 140: Update Ricard "Treatise on Geophysics" reference to the 2nd edition?

We added this reference.

l 146: Give a more complete/rigorous definition of a continuum. For example something like what it says in Wikipedia "… certain physical phenomena can be modeled assuming the materials exist as a *continuum, meaning the matter in the body is continuously distributed and fills the entire region of space it occupies.* A continuum is a body that can be continually sub-divided into infinitesimal elements with properties being those of the bulk material.

We have changed the sentence to:
"We will also treat Earth materials as a continuum, i.e., we assume that the material is continuously distributed and completely fills the space it occupies, and that material properties are averaged over any unit volume. Thus, we ignore that the material is made up of individual molecules (Helena, 2017)."

l 183-185. Curbelo et al. (2019)'s relaxation time analysis doesn't apply to the mantle because they are considering an **ideal gas**, i.e. with the ideal gas equation of state (this is where their equation 4.4 comes from), which does not include elasticity. If you do the same analysis for a visco-elastic material like rock using an appropriate equation of state such as Birch-Murnaghan, then elasticity will be the dominant mechanism to even out pressure variations, i.e. if you apply a pressure perturbation in one place then it will propagate not via viscous relaxation but via elastic (seismic) waves, which are relatively fast. Of course even with seismic waves there is a component of viscous relaxation - this is why there is attenuation - so it would be possible to derive a time-scale for elastic waves to die out due to attenuation. A related point: compression of rock as it descends through the mantle does not occur by viscous relaxation, it occurs by elastic compression.

We changed the text to:

"Because the first term explicitly includes a time-dependence, it introduces a characteristic time scale into the model due to viscous (Curbelo et al., 2019) and elastic forces (Patocka et al., 2019). [...] When we consider the Earth as a visco-elastic body (see Section 2.2.1), this relaxation time is dominated by elastic forces and is on the order of a few hundred years for the upper mantle to a few tens of thousands of years for the lower mantle. "

l 248: This simplification is only correct if density and Cp are constant. (as the authors note later, but it should be stated here as well).

We have changed the sentence to

"[...] Eq. (6) can be simplified by dividing it by ρCp, assuming that they are constants."

l 264-265: Here their expression of S2 is the general case of pressure varying in any direction whereas what they write is "…the dominant pressure variation…is the effect of the lithospheric pressure increase with depth". In this case the expression can be simplified: where z=vertical, positive downwards, hence . This is the expression that is normally used in mantle convection codes.

We agree that this is the more common implementation (whereas the original version was what is implemented in ASPECT), and changed the text accordingly.

l 289. When considering the various approximations, the order-of-magnitude fractional density error that is expected from ignoring thermally-related density variations is , which is O(1%) ( ~10-5, dT~103), so errors of this magnitude are considered "par for the course". This is the magnitude of error you get in thermal boundary layers / slabs / plumes with the Boussinesq approximation, and also with the Anelastic Liquid Approximations. It does not "invalidate" either approximation, it is rather the magnitude of error that is accepted in making the approximation. (an aside: the Anelastic Approximation *is* invalidated by large T deviations from the adiabat, this is why the Anelastic *Liquid* Approximation is preferred for mantle dynamics).

This was indeed unclear; what we meant were processes that invalidate the assumption that deviations from the reference profile are small (not processes that invalidate the approximation). We have now reworded this.

In addition, we have also added a paragraph (l 312-322) that lists the estimated density variations caused by temperature, composition, lithostatic/dynamic pressure, and phase transitions, so that the reader gets a better understanding on when some of these approximations may or may not be accurate. This also relates to the next comment.

We would also like to point out that the error is usually on the order of 1% in many global convection models, which is why it is appropriate to use the Anelastic Liquid Approximation, but for some lithosphere scale models it may be much larger. Dehydration reactions in subducted slabs can have density changes of up to 10%, and if we consider the cooling of material at a mid-ocean ridge (assuming the thermal expansivity at the Earth's surface is ~4 * 10^-5 K-1 and material cools by 1000 K), the density difference would be about 4%, and the stresses caused by that contraction may be an important effect in the model.

l 294-295. In the convecting mantle, the magnitude of error in ignoring dynamic pressure in the density calculation, hence the resulting energy imbalance, is very small for realistic Earth parameters. - The magnitude of dynamic pressure can be estimated by how much dynamic topography is generated by mantle flow: ~a few km. Compare this to the mantle depth of ~3000 km - the pressure error (dynamic/lithostatic) is thus in the range 0.1-1%. - Leng and Zhong (2008) found resulting energy imbalances of up to ~few % because their experiments were at low Rayleigh numbers of 104-105. Stress and dynamic pressure decrease with increasing Ra as roughly Ra-1/3 (assuming that Ra increases because viscosity decreases) so extrapolated to Earth-like Ra of 107-108, the expected energy imbalance is less than 1%. - In deciding whether this small error is worth doing something about (i.e. using ALA instead of TALA), one must also consider the accuracy of the numerical pressure solution. For example, a well-documented issue with the finite volume (staggered-grid finite difference) discretisation is artificial pressure overshoots at viscosity jumps (e.g. Deubelbeiss and Kaus, 2008). If large, localised viscosity contrasts exist inside the modelled mantle, this "numerical" dynamic pressure is not something

that one would want to use in calculating physical properties - it might result in larger errors than simply ignoring dynamic pressure.

We added a sentence to make clear that the density variations caused by the dynamic pressure are very small compared to the ones caused by the temperature.

In addition, we added a sentence at (l 355-356) that explains that sometimes the numerical methods being used influence what approximation one may want to pick.

l 308: For the extended Boussinesq approximation it is not correct that "adiabatic heating leads to artificial generation of energy in the model". (i) Adiabat heating removes energy from the system, it does not generate energy. "Heating" is a bit of a misnomer. This is because "heating" only applies to sinking material: rising material cools. Furthermore, rising material cools at a more rapid rate than sinking material is heated, because the adiabatic gradient is proportional to T, i.e. hot adiabats are steeper than cold adiabats. There is an equal amount of rising material and sinking material (mass conservation). The result of all this is that cooling of rising material exceeds heating of sinking material, so the AH term removes heat from the system. In equilibrium, heat loss due to adiabatic heating is exactly balanced by heat input due to viscous dissipation: the volume-integrals of the two terms are equal and opposite (e.g. Jarvis and McKenzie, 1980). (ii) I did a quick test EB calculation using StagYY: 1x1 box, Ra=105 (based on total temperature drop), all properties constant=1, dissipation number=1. This is large Di - it means temperatures increase by a factor of exp(1.0)=2.7 from top to bottom - more than in the Earth. Despite this, integral(adiabat heating) = -integral(viscous dissipation), top heat flux = bottom heat flux, there is no energy imbalance. Viscous dissipation: min = 3.52E-03 ; **mean = 3.96E+00** ; max = 4.24E+01 Adiabatic heating : min = -1.33E+02 ; **mean = -3.96E+00** ; max = 9.63E+01 Top flux and Nu = 4.860 4.860 ; Bot flux and Nu = 4.860 4.860

[Figure]

What we wanted to say here is that in reality, adiabatic heating requires work being done (and the material being compressed) for an increase in its temperature. But in models that use the EBA, material is sinking down and therefore heating up adiabatically, but the mechanical work

that would lead to this heating is not being done. So the temperature increases, but the material is not compressed. This means that energy is created out of nowhere.

This of course doesn't say anything about the net energy change that is introduced by this process, since the opposite happens when material is rising and cooling down. We didn't want to make any statements about what the net energy change would be, and we wanted to keep this overview of the approximations short and concise, so we changed the sentence to:

"Since it includes adiabatic heating, but not the associated volume and density changes, adiabatic heating can lead to artificial changes of energy in the model, i.e., material is being heated or cooled based on the assumption that it is compressed or it expands, but the mechanical work that causes compression or expansion is not done."

Of course, in a fully compressible model, adiabatic heating should not remove energy from or add energy to the system at all. When the equations are formulated in terms of, for example, entropy, then the term vanishes completely.

l 340: Use passive tense: "writes" -> "is written"

We changed it to "can be written as".

l 426: "strain rate increases" -> "strain or strain rate increases"

We adjusted this sentence accordingly.

l 459 Either no comma or two commas: "…other variables like chemical composition…" or "other variables, like chemical composition, is…".

We adjusted this sentence.

l 519-520. There is some confusion nowadays over the difference between finite difference and finite volume discretizations. - The staggered-grid ("conservative") finite difference discretisation used in codes like StagYY, LaMem, I*VIS etc. is an example of a finite volume discretisation, and is normally referred to as such in the broader numerical simulation community and in many papers in our community (e.g. Ogawa et al. 1991; Trompert and Hansen, 1996; Shahnas et al. 2011). So, staggered-grid finite differences = finite volume, but - it is also possible to have non-staggered grid finite differences that cannot be described as finite volume (e.g. several of the codes in Blankenbach et al. 1989), - or unstructured-grid finite volume codes that cannot be described as finite difference (e.g. Hüttig and Stemmer, 2008). - In conclusion, I suggest adding a clarification sentence, for example "We note that the commonly-used staggered grid finite difference discretisation is an example of a finite volume discretisation".

We added:

"The last two are equivalent in some instances, such as in the case of the commonly-used staggered grid finite difference discretisation in geodynamic codes."

l 556-557: This explains the "difference" part of "finite difference"; why not also explain the "finite" part, which comes from the mathematical definition of a derivative as being a limit as h, the difference in coordinate, tends to 0: being replaced by a formula in which h is **finite**:

We added the following sentence to explain the 'finite' part of 'finite differences':

"In addition, `finite' refers to the mathematical definition of a derivative as a limit where h --> 0 is replaced by a formula in which h remains finite (see Appendix A)."

l 584: 1024 -> 1024.

We fixed this.

l 606-607: The most common iterative method used in geodynamic codes is the **multigrid method**. This is what is used in ASPECT, CitCom, StagYY, I3ELVIS, LaMEM, TERRA, etc.

We clarified this, by modifying the sentence:

"Common iterative methods in geodynamic codes are the Conjugate Gradient method and the GMRES (Generalized Minimal RESidual) method [...] which are used in conjunction with multigrid methods to accelerate their convergence."

l 620: "top 500" -> "TOP500 list (https://www.top500.org)"

We fixed this.

Figure 5: A nice figure, but in the MPI part, bottom right, the 4 processors are all on the same node so actually MPI is not necessary - OpenMP could be used instead. It might be more illustrative to have/use only 1 CPU per node, so that the different nodes are communicating over the network.

We updated the 'computing' figure to reflect only 1 CPU per node for the MPI example, which makes the communication between the nodes more illustrative.

l 664-667: The sentences on advection methods need to be rewritten/expanded/clarified. (i) A distinction should be made between methods designed to treat discontinuities such as a free surface, and methods designed to treat smoothly-varying fields such as temperature. (ii) The methods they mention (level set, marker chain, volume of fluid) are designed to treat discontinuities. Actually these work well and are widely used. I don't think there's anything "notoriously difficult" here. If all you need to track is one discontinuity, then using one of these methods makes more sense than placing particles everywhere in the domain. (iii) Tracking

temperature or other smoothly-varying fields is easier and many methods have been developed over many decades to advect fields while minimising artefacts such as numerical diffusion and dispersion (ripples). These methods work well and are in common use in a variety of fluid dynamics fields including mantle dynamics. For example, finite-element codes ASPECT and Citcom* use the streamline upwind/Petrov-Galerkin (SUPG) method. For finite-volume codes a variety of methods are available such as TVD (Total Variation Diminishing), FCT (Flux-Corrected Transport) and MPDATA (Multidimensional Positive Definite Advection Transport Algorithm), to name a few, and all of these are conservative. Again, if all you need to track is a smoothly-varying $df\ dx = \lim h{\rightarrow}0\ f\,(a+h)-f\,(a)\ h\ df\ dx \approx f\,(a+h)-f\,(a)\ h$ field like temperature, it makes more sense to use one of these methods than to fill the domain with particles. (iv) For compositional variations that exist everywhere in the domain, or multiple composition fields, it's best to use particles, although you can use one of the methods in (iii) - there isn't a single "compositional fields method" here - actually many possibilities. Some of the field-based methods are tested against particle methods in van Keken et al. (1997) and Tackley and King (2003).

Thank you for the clarification. We rewrote section 3.7 (l 759 - 785) and added the mentioned advection methods designed to treat discontinuities or smoothly varying fields. We also nuanced our discussion on the marker-in-cell technique and added an example of weighing the pros and cons of methods depending on the property that needs to be tracked.

l 670: Other disadvantages of particle-based methods are the introduction of artificial noise and the lack of conservation of advected quantities (when averaged to the grid).

We added these disadvantages to the text, see revised text in the above comment for lines 781-782.

l 670: Particle-based methods are not difficult to parallelise: each process holds the grid cells and particles in its subdomain and then each time step after advection, particles that have crossed to other subdomains are communicated to those subdomains. The only potential issue comes when the subdomains have different volumes, for example as a result of adaptive grid refinement - this can lead to load imbalance (i.e. different #particles in different subdomains).

We agree and adjusted the sentence accordingly. See comment for lines 759-785 for all changes.

l 726: Another useful example is Kramer et al. (2020).

We added this reference.

l 745: Two more useful community benchmark papers: Travis et al. (1990), which was the US equivalent of Blankenbach et al (1989), and Busse et al. (1994), one of the few 3D benchmark papers.

We added these references.

l 746: Zhong et al. (2008) is not really a community benchmark because it is only testing one code; we don't get an idea of how different codes/methods compare. If you're going to include this why not include Tackley and King (2003) where we at least tested 2 codes (and several methods of treating composition).

We added this reference.

l 907-916: Mention that periodic boundary conditions are the natural choice for global simulations.

We added this.

l 993: Initial conditions do not always determine the model outcome: in long-term simulations or ones obtaining steady-state solutions it can often be that the initial condition is "forgotten". Therefore it would be more accurate to write "initial conditions **can often** determine the model outcome".

We added this.

l 1026: add Duretz et al. (2011) - for example this is what I am using. It's basically like Kaus et al. (2010) but for the finite volume discretisation.

We added this reference.

l 1068 "percentage" -> "fractional"

We modified this.

Section 9: There are a lot of underlined words in this section. I get the impression they are supposed to be hyperlinks, but when I click on them (in Adobe Acrobat Reader) nothing happens. Replace them with normal referencing.

We apologise for the inconvenience of the broken links. Apparently this is an issue that Solid Earth has in the discussion phase of the review process. We will make sure that the links are working in the final version of the manuscript together with the copyeditor.

Appendix A: Could list a few more references for numerical modelling, especially ones explaining finite element methods - for example the Zhong et al Treatise chapter or the book by Simpson (2017).

We added these references.
* * *
REFERENCES

Bunge, H.-P., C. R. Hagelberg, and B. J. Travis (2003), Mantle circulation models with variational data assimilation: inferring past mantle flow and structure from plate motion histories and seismic tomography, Geophysical Journal International, 152(2), 280-301.

Burstedde, C., O. Ghattas, G. Stadler, T. Tu, and L. C. Wilcox (2009), Parallel scalable adjoint-based adaptive solution of variable-viscosity Stokes flow problems, Computer Methods in Applied Mechanics and Engineering, 198(21-26), 1691-1700.

Deubelbeiss, Y., and B. J. P. Kaus (2008), A comparison of Eulerian and Lagrangian numerical techniques for the Stokes equation in the presence of strongly varying viscosity, Phys. Earth Planet. Int., 171(1-4), 92-111.

Duretz, T., D. A. May, T. V. Gerya, and P. J. Tackley (2011), Discretisation errors and free surface stabilization in the finite difference and marker-in-cell method in geodynamic applications: A numerical study, Geochem. Geophys. Geosyst., 12(Q07004), doi:10.1029/2011GC003567

Dziewonski, A. M. (1984), Mapping the Lower Mantle - Determination Of Lateral Heterogeneity In P-Velocity Up to Degree and Order-6, Journal Of Geophysical Research, 89(B7), 5929-.

Huettig, C., and K. Stemmer (2008), The spiral grid: A new approach to discretize the sphere and its application to mantle convection, Geochem. Geophys. Geosyst., 9(2), doi: 10.1029/2007GC001581.

Ismail-Zadeh, A., G. Schubert, I. Tsepelev, and A. Korotkii (2004), Inverse problem of thermal convection: numerical approach and application to mantle plume restoration, Physics of the Earth & Planetary Interiors, 145(1-4), 99-114.

Jarvis, G. T., and D. P. McKenzie (1980), Convection in a compressible fluid with infinite Prandtl number, J. Fluid Mech., 96, 515-583.

Keller, T., D. A. May, and B. J. P. Kaus (2013), Numerical modelling of magma dynamics coupled to tectonic deformation of lithosphere and crust, Geophysical Journal International, 195(3), 1406-1442, doi:10.1093/gji/ggt306.

Liu, L., and M. Gurnis (2008), Simultaneous inversion of mantle properties and initial conditions using an adjoint of mantle convection, Journal of Geophysical Research: Solid Earth, 113(B8), n/ a-n/a, doi:10.1029/2008jb005594.

Ogawa, M., G. Schubert, and A. Zebib (1991), Numerical simulations of 3-dimensional thermal convection in a fluid with strongly temperature-dependent viscosity, J. Fluid Mech., 233, 299-328.

Shahnas, M. H., W. R. Peltier, Z. Wu, and R. Wentzcovitch (2011), The high-pressure electronic spin transition in iron: Potential impacts upon mantle mixing, Journal of Geophysical Research: Solid Earth, 116(B8), B08205, doi:10.1029/2010jb007965.

Simpson, Guy (2017) Practical Finite Element Modeling in Earth Science using Matlab, Wiley, ISBN: 978-1-119-24862-0

Tackley, P. J., and S. D. King (2003), Testing the tracer ratio method for modeling active compositional fields in mantle convection simulations, Geochem. Geophys. Geosyst., 4(4), doi: 10.1029/2001GC000214.

Trompert, R. A., and U. Hansen (1996), The application of a finite-volume multigrid method to 3-dimensional flow problems in a highly viscous fluid with a variable viscosity, Geophys. Astrophys. Fluid Dyn., 83(3-4), 261-291.

van Keken, P. E., S. D. King, H. Schmeling, U. R. Christensen, D. Neumeister, and M. P. Doin (1997), A comparison of methods for the modeling of thermochemical convection, J. Geophys. Res., 102(B10), 22477-22495.

---

## Author Comment (AC3)

**Geodynamics 101 Revision - Review Boris Kaus**

This is the authors' response to the review of Boris Kaus. Our responses are indicated in **green**, with the original review presented in black.

This is a lengthy, but well written and useful manuscript that highlights many aspects of geodynamic modelling. I have a number of minor suggestions to further strengthen the manuscript in the attached pdf.

**Review of van Zelst et al. 101 Geodynamic modelling: How to design, carry out, and interpret numerical studies**
*Boris Kaus, Mainz 1.6.2021*

This is a well-written paper that is likely to be very useful for people that are new to geodynamic modelling. Putting all of that together in one place is challenging and I congratulate the authors for doing a good job with it. Unsurprisingly it is rather long and feels more like a book (which partly explains why it took me so long to get the review back to you - apologies!).
This certainly deserved to be published in SE. Yet, I do have a number of comments which I believe would be good to address first. This can mostly be done with some rewriting, so I don't think it should take you a lot of time (yet, it may help to point new geodynamicists in the right direction).

**1) Modelling tools**
You discuss quite a bit about the open source modelling packages like ASPECT, Underworld or LaMEM. Yet, missing from this discussion are alternative approaches that are based on smaller (sometimes one-page) scripts. Those are not full-blown modelling packages but rather simpler scripts that solve a particular problem very well (and fast). The classical example of that in geodynamics is the MILAMIN code (http://milamin.org) which remains one of the fastest codes to solve the incompressible Stokes equations on 2D unstructured meshes, and may be very helpful for those interested to solve for example, problems with viscous inclusions or interacting crystals. Other examples are the M2Di scripts which are concise matlab and julia codes to solve viscoelastic problems, including regularized plasticity, available from https://bitbucket.org/lraess/m2di/src/master/. The most recent development in this direction is the ParallelStencil julia package (https://github.com/omlins/ParallelStencil.jl#stencilcomputations-with-math-close-notation) which comes with many geodynamic examples.

Geodynamicists that are interested to work on technical developments as well, may find such scripts much easier to understand than the big software packages that can do it all. In fact, with the ParallelStencil julia package it is possible to write a very compact code that scales to very large parallel GPU-based supercomputers with almost no effort (provided you use a pseudotransient iterative solver approach). Other efforts (under

development) allow calling PETSc and its staggered grid interface from julia. In my opinion such approaches may become increasingly important in the future as it allows PhD students to go from writing an experimental solver to a fully blown (parallel) production code in a rather straightforward manner. This will help to address new multiphysics problems, such as the coupling between reactions & deformation. Given the informative nature of your current paper it would be good if you can discuss these topics as well in your manuscript (and give some of the links above).

We added a paragraph at the end of section 3.8 (l 801-814) about these recent approaches in building multi-physics applications. In this paragraph, we also addressed the comment mentioned later on doing multiphysics beyond the coupling of codes through boundary conditions.

**2) Parameter sensitivity/controlling parameters**

Typical forward models used in many geodynamic applications indeed have a large amount of parameters (as you discuss around lines 116 and lines 845). Usually, such sensitivity studies are done 'by hand' by modifying input parameters, making a model run and checking the difference with respect to a reference model. Yet, part of this can be done automatically by computing scaling exponents which directly show which of the parameters control the velocity at a certain point (as discussed in Reuber et al. 2018 Tectonophysics and used in Reuber et al. 2018, Front. Earth Science). In case adjoint methods are used to compute the gradients this is even computationally extremely efficient, and gives you the sensitivity to all model parameters at the same time. This would go a long way in determining which of those are of first order importance and which are not. This method only gives the sensitivity of the model results for a particular timestep/geometry but for many of the cases we looked at so far this sensitivity did not change drastically during a model simulation making this a quite powerful techniques (provided adjoints are available).

I suppose that the reason that this is not yet more widely applied is that it is not yet implemented in many of the codes currently in use, but I can well imagine that this may change soon. It would be good to highlight this as it is a very useful and automatic way to map and reduce the model parameter space (section 5.2.2) and will help to reduce the number of required simulations and thus the CO2 emissions of a study (line 1455).

This is indeed an exciting direction for future modelling. To include this into the manuscript, we added a sentence to this approach in the introduction (section 1.1):

"An alternative approach is to incorporate automatic parameter scaling routines or use adjoint methods to test parameter sensitivities in models (Reuber et al., 2018a,c), which could considerably reduce the amount of models required."

To emphasise how adjoint methods will make scanning the parameter space more efficient - particularly in the case of general modelling studies where large parameter sweeps are common, we added a further explanation of adjoint methods at the end of paragraph 5.2.2:

"The mapping of a parameter space is often done through manual variation of a single model parameter and comparison of the resulting model predictions. However, recent developments allow for scaling laws between the model solution and the model parameters to be computed automatically through adjoint methods. Besides solving inverse problems (e.g. Ismail-Zadeh et al., 2003; Ghelichkhan and Bunge, 2016; Colli et al., 2018), adjoint methods can efficiently compute the scaling exponent for all model parameters with one linear solve (for a specific model time step) (Reuber et al., 2018c). These scaling exponents (that are based on the derivative of the solution parameter to model parameter) indicate which parameters control the model solution and which have a lesser effect (e.g. Reuber et al., 2018a; Crawford et al., 2018). Knowledge of the relative importance of each model parameter can help decrease the parameter space that is to be investigated (see also Section 7.3)."

**3) Section 2.1.1: Mass conservation**

Later in the manuscript (and around line 179/180 & lines 203-205) you discuss bulk compressibility, poison ratio etc., but you don't show how that should be added to the mass conservation equation in equations (same in line 452). It would clarify matters if you can add this.

We incorporated this comment in various places in the manuscript:

Section 2.2.5: We now include an explicit equation for the density in our section on the equation of state (the simplest case of rho = rho0 (1 - alpha \Delta T). We realise that there are many more complex equations of state, but there are many different ways for how, for example, the compressibility can be included in the equation of state, and many of them can not even be expressed as an equation anymore, for example a look-up table (we also added a sentence to highlight such efforts). In addition, choosing a different equation of state does not change the equations (2)-(7) beyond them being compressible or incompressible or using a specific approximation, which is something we already discuss. The only part that changes is the values of rho (and how it depends on the solution variables). Because of this, we have decided to not add any other specific examples for equations of state.

Equation 2: We now also include an explicit reference to the equation of state section from the section about mass conservation.

After eq 4.: We also added the definition of Poisson's ratio - defining it in terms of Lame's first parameter and shear modulus for a homogeneous isotropic linear elastic material.

**4) Solution methods**

You discuss different solution methods in section 3.4, but what is completely missing is a discussion about multigrid preconditioners. Users of any 3D geodynamic code will run into having to use multigrid at some point or the other and will wonder why it sometimes does not converge (and sometimes does). As people new to geodynamic modelling are the target audience of this paper, it would be good if you can add a paragraph to

discuss this (and why, for many lithosphere dynamics problems, it is important to have a coarse grid that still "feels" the main viscosity structure of the model and is thus not too coarse).

What I believe is also important to discuss are pseudotransient solvers (there is much recent work by Raess et al.) as they result in compact solvers that scale particularly well on GPU's systems (see link to the ParallelStencil julia package for sample codes).

We clarified and added that the solver methods described can be augmented with multigrid for optimized performance, also in response to a comment from Reviewer #1 (section 3.4). We have also added the following information on pseudo-transient solvers (lines 711-714):

"Recently, iterative pseudo-transient solvers have been used to solve coupled sets of equations (Räss et al., 2019;Reuben et al. 2020). These methods introduce a physics-based transient-term (a time derivative) to a steady-state equation, in order to iterate towards the steady-state solution. The matrix-free, finite difference pseudo-transient schemes of Räss et al. (2019) are well-suited for GPU (Graphical Processing Unit) accelerated systems (Räss et al. 2020)."

**5) Particle-in-cell methods**

I agree that for lithosphere dynamics codes, the particle-in-cell method is the most popular one (and I don't really understand why you say it is difficult to parallelize). Perhaps you can explain here why that as, which is in my opinion because it is the easiest method to take things like phase transitions, history variables like strain as well as large deformations in a simple manner. Many alternative approaches have been suggested over the years, and many of those are good if dealing with a limited amount of phases (e.g. bubbles interacting with crystals can be well approximated with level sets). Yet, somehow none of these other methods withstood the test of time, perhaps because they are not general enough.

We removed the sentence that PIC methods are difficult to parallelize and significantly improved our section on tracking materials (section 3.7) (l 759-785; also in response to a comment from Reviewer #1).

**6) Multiphysics**

Multiphysics is indeed an important avenue of future research (perhaps even one of the most important ones in geodynamics). Yet achieving this by "code coupling" of different, unrelated, codes really only works if there is only a loose coupling between the physics (section 3.8). This is perhaps the case when coupling models of surface processes and lithosphere deformation. Other problems, on the other hand, require a much stronger coupling on the solver level which implies that new solvers should be developed. An example are the two phase flow equations that describe magma migration, which roughly consists in a Stokes-like and a Darcy-like problem. Getting the solution in an efficient manner cannot be done by taking a Stokes code and a separate Darcy code and coupling that using batch scripts. Instead, the coupled set of equations needs to be

solved in a tightly integrated manner. Our knowledge on the individual systems is still useful, as we can use multigrid preconditioners that work well for Stokes as part of this, for example. With this in mind, PETSc developed the multi-physics framework (see papers by Jed Brown) as well as the recently developed DMStag interface which allows you to add a Darcy-like code to an existing Stokes solver in a straightforward manner. I think it is important that you clarify this here, as there have been too many fruitless attempts in the past already to do loose coupling of different codes (which sounds intuitively easier but has very strong limitations for more tightly coupled problems).

That is a good point. We have added a paragraph at the end of the section, saying:

"However, some multi-physics problems are so closely intertwined that solving the coupled system of equations requires different numerical methods than solving the problems individually (for example, coupled magma/mantle dynamics). In these cases, coupling requires the development of new code that tightly integrates the different physical processes."

This fits well with comment 1 on APIs, because using them could overcome these limitations on coupling codes that are incompatible at a fundamental level (i.e. defining a coupled system of equations).

**7) Creep rheologies & modelling manuscripts**
You spend quite some space describing how to write a modelling paper, which is certainly quite useful for people that are new to the community and/or to the paperwriting business. Yet, after having taught a class for over 10 years in Mainz in which students are supposed to reproduce published geodynamic modelling papers with a different code, I think the two most commonly made mistakes are not described here.

a) The first one is that in many papers, the model parameters are incorrectly listed in the tables, perhaps because the units of say dislocation creep laws are non-intuitive (the prefactor has units of the form of $MPa^{-n} s^{-1}$ where n is the powerlaw exponent; if n>1, transferring this to $Pa^{-n} s^{-1}$ takes a bit of work). In addition, creep law experiments are usually given in terms of principal stresses, whereas geodynamic models need to have this in a tensor format, which involves correction factors (see the textbook of Gerya for a nice derivation). All these issues add up to make it quite difficult to fully reproduce the experiments unless the input scripts are attached. In most cases these are obvious typos, as employing the stated parameters results in wide spread drip tectonics, rather than stable subduction (as many generations of students in mainz have experienced), so it seems quite clear that it is actually correctly implemented in the code. Yet, whether the correction factor is taken into account or not is often less clear as it has a smaller effect. To me this points to a deeper underlying problem: there is currently no central database that collects all experimental creep rheology data in a format that is directly implementable in different geodynamic simulations. If we would have that it would eliminate quite a few of those mistakes.

b) A second issue is that often not all parameters are listed and some info is missing (nor just in terms of the material parameters employed but also with respect to the numerical convergence criteria etc.). I suppose that the reason for this is that there are often so many model parameters that it is quite easy to overlook some. A potential way around that would be to develop scripts that automatically generates tables with model parameters from input scripts which would eliminate another source of mistakes (which is that it seems that often some parameters are forgotten to be described in detail). Ofcourse publishing the input script used to perform the simulations, as you point out, helps as well, but in that case it is still not fully guaranteed that the creep law used in one code are implemented in the same manner in a different code (while using the same correction factors).

To address both comments a,b, we added a paragraph in section 8.1 on how information should be given in the methods section (l 1416 - 1430).

I think it would be good if you can discuss these topics in the manuscript. To minimize the chance of mistakes in the future, developing standardized databases and automatized ways to create the parameter tables could be very helpful.

These are great ideas, but we think they should be a community effort and are therefore unfortunately beyond the scope of this manuscript. Summarised potential ideas for the community:
1. Central database that collects all experimental creep rheology data in a format that is directly implementable in geodynamic models.
2. Database with standardised model inputs/parameters tables: create automatised ways to print input parameters as tables ready for publications, to avoid editing mistakes.

**Additional comments**
line-wise (mostly typos, with some longer comments)

l. 55: One of the reference textbooks on analytical solutions is Turcotte and Schubert, which would be good to list here. Similarly, the textbook of Neil Ribe (theoretical mantle dynamics) has lots of useful info as well.

We added these additional references.

l. 170: highly viscous fluid

We corrected this typo.

l. 259: please define deviatoric stress in equations, and not just in words.

We include the equations in Section 2.2.1, and we now refer to that section.

l. 263: you did not explain what D/Dt means (or give equations)

We have added the definition (including the equation).

l. 269: The shear heating term should only involve the non-recoverable deviatoric strain rate components (that is, non-elastic) and not the full strain rate. I realize that you define strain rate only later in the paper and that for incompressible viscous rheologies this is equivalent. For compressible viscoelastic materials it is however not the same, so it seems important to point that out here.

We clarified the text.

l. 268: thin phase transitions? Not sure what you mean by that.

We changed it to "For phase transitions that occur over a narrow pressure/temperature range"

l. 349: It would be good if you can give the mathematical definition of how to go from total to deviatoric strain rate (which indeed simplifies to what you write here in the incompressible case).

We added an explanation after eq. 9.

l. 368: fH2O looks as if it could be several symbols; perhaps better to write as f_{H_2O}

Fixed equation 10.

l. 368; please indicate the units of all variables. In fact that is an issue throughout the manuscript and without specifying the units of parameters you employed, modelling results will not be reproducible and replicable…

Our equations are formulated independently of the units; i.e. the pressure could be given in Pa, or bar or any other unit, as long as the material parameters are adjusted accordingly. Consequently, we have decided to not indicate any units here (and throughout the manuscript). To emphasise the importance of units in normal studies of geodynamic modelling, we added sentences on how to properly include parameter tables (including units). Also see our response to comment 7b above and our response to the next comment.

l. 368/369: Can you add the units of all parameters?

We specifically do not include the units of the parameters in the manuscript, for multiple reasons:
  - This would needlessly clutter the (already long) manuscript and reduce the readability of the manuscript in our opinion

- The units of some parameters (specifically constants even if we would stick to SI units) can differ in different formulations of the constitutive equations, making it an inherently incomplete and potentially confusing exercise to include the units.
- As mentioned above, as long as no values for specific parameters are given, the variables can be in any units as long as they are consistent with each other within the equation, as they are formulated independent of the units.

However, we agree that the consistent use of units is important for writing geodynamic modeling manuscripts and have added a sentence about this in Section 8.1 (also see our response to comment 7b).

l. 402: Plasticity can also be used for pressure-independent yield criteria. Examples are Griffith's criteria (tensile failure) or implementing an ultimate yield stress in geodynamic models. It's therefore better to remove 'pressure-dependent'

We removed 'pressure-dependent'.

l. 410: equation 14 is actually not identical to equation 13, unless cohesion C has a different meaning in eq. 14 compared to eq. 13. We actually had a discussion about this before in Solid Earth, so please have a look at https://se.copernicus.org/articles/11/1333/2020/ to see an illustration of the difference. The reason that cosine and sine terms appear in eq. 13 instead of tan(\mu_f) is that this is the yield stress function.

We agree that equation 13 and 14 indeed are not the same. We significantly rewrote section 2.2.4 to better introduce the various yield criteria.

l. 416/417: If lithostatic pressure is used to evaluate the yield criteria, the shear band orientation is always 45 degree and there is no difference between compression and extension. I realize that many large-scale convection codes use that assumption, and assuming lithostatic pressure is fine within the mantle. Yet, within the stronger lithosphere there can be quite strong deviations between lithostatic and dynamic P (up to factor 2 under compression). It's probably good to point that out to the readers.

We added this further explanation:

"If lithostatic pressure is used in the yield criterion, shear bands are always orientated 45° under both extension and compression. This assumption is allowed for the mantle, where the total pressure is close to the lithostatic pressure. However, pressure can deviate strongly from the lithostatic pressure in the lithosphere, which can have major effects on the results."

l. 445. An easy-to-follow explanation of how to implement anisotropy in 2D codes is given in Kocher et al. Tectonophysics 421, 71–87.

We added this reference.

l. 558: A big difference between FD and FE methods is that in FD, you solve the partial differential equation in a pointwise manner, whereas FE approximate the equations on average per element. It would be useful to add a comment on that here.

We added:

"This also controls how the partial differential equations are solved on the grid, with finite difference methods solving the equations pointwise and finite element methods averaging the equations per element."

l. 562: Perhaps add a small remark on why finite elements should use a mixed formulation with higher order for velocity compared to pressure to get reliable results in the (near)-incompressible limit.

We added this as "Stable elements are typically characterised by m>n." and made a link with figure 10.

l. 584: 10^{24} Pas

We fixed this.

l. 740: the method of manufactured solutions also works for nonlinear problems which is perhaps good to mention here.

We changed the sentence to reflect this point.

l. 745: It is perhaps interesting to mention that community benchmarks is a process that typically takes several years…

We added a sentence to highlight this.

l. 963: geomio is spelled "geomIO"

We fixed this.

l. 1026: Implicit timestepping, in which advection becomes part of the nonlinear solution step, is an even better method to deal with the drunken sailor effect (was described in Popov & Sobolev 2008, even if only very briefly).

The Popov & Sobolev (2008) paper does not make a clear statement about implicit time stepping avoiding the drunken sailor effect, so we refrained from adding this reference and remedy.

Fig. 10: Maybe it is good to mention that the reason the drunken sailor effect occurs is that the typical density difference between rocks and air is much much larger than the typical density difference within the Earth. Moving the Earth's surface by one meter therefore causes a much larger stress perturbation than moving the Moho by a meter and that is why the models have a tendency to become unstable.

We have clarified this in the main text (l. 1168-1172) as well as in the caption of Figure 10. We now say:

"The stark density contrasts (approx. 1.2~$\mathrm{kgm}^{-3}$ versus 2830~$\mathrm{kgm}^{-3}$) lead to much larger stress perturbations from topographic changes compared to similar topography variations at a typical density contrast inside the Earth (e.g. the density jump at the continental crust-mantle boundary is $\sim$280~$\mathrm{kgm}^{-3}$ \citep{Martinec1994})."

Fig. 10b: Even when you can 'fix' the wrong pressure field in this case by smoothing there are other, more heterogeneous, setups where such smoothing does not fully remove the artefacts. It is thus clearly better to employ LBB stable elements (like Q_2P_{-1}) throughout, The other undesired side-effect of unstable elements such as Q_1P_0 is that they require more iterations for a higher resolutions, if combined with iterative solvers. Stable elements fix that.

We added:

"Using an LBB stable element, like $Q_2\times Q_1$ or $Q_2\times P_{-1}$, avoids this problem (Donea & Huerta, 2003)."

to the figure caption, and a more extensive explanation to the main text. See our response to the reviewer's comment about line 1047 (below).

Fig. 10c: I am not sure why you claim that the higher-resolution model is better here. It seems that both models are performed with non-regularized plasticity so both are actually numerically non-convergent. On one hand this manuscript argues about the importance of reproducible and numerically trustworthy results. On the other hand you show a key example in geodynamics where this is not the case. In my opinion there is no way around using a form of regularized plasticity (together with sufficient resolution such that the plasticity length scale is captured) and it would thus be better to use an example of that within the current manuscript. There are some recent papers by Duretz et al. showing that this can be done, for example by using viscoplastic regularization, so it seems more appropriate to show examples of such computations in this figure.

In the main text (l 1154-1164), we state that common plasticity implementations are non-convergent with resolution, as illustrated in the figure. The figure however indeed suggests that the higher-resolution solution is better, but this was only meant to indicate that the fault

angle at higher resolution is closer to the theoretical angle. To avoid ambiguity, we have removed the check mark symbol from the higher resolution figure and updated the caption of Fig 10 to:

"The angle and thickness of the shear bands is dependent on the mesh resolution. Regularised plasticity implementations and sufficient resolution are required to achieve convergence with resolution (e.g. Duretz et al., 2020)".

We also added the Duretz et al. 2018 and 2019 and DeBorst et al. 2020 references to the main text.

l. 1047: Or better avoid unstable elements altogether and use stable ones (which also scale better on parallel computers using multigrid solvers)..

We have included the following at line 1194 to point this out: "Stable elements, which fulfil the Ladyzhenskaya- Babuska-Brezzi compatibility condition (LBB or inf-sup condition), do not exhibit pressure artefacts (Donea and Huerta, 2003). Moreover, the required number of outer iterations does not increase significantly with mesh resolution compared to the Q1×P0 element (Thieulot and Bangerth, 2021)."

l. 1184: probabilistic

We fixed the typo.

l. 1311: In the geosciences it is common practice to acknowledge the work that reviewers put in going through the manuscript and making suggestions. Not everyone in computational geodynamics follows this (unwritten) rule, however, which I know doesn't go well with many of my colleagues (including myself). So perhaps it is good to spell this out here.

We agree it is good practice to thank the reviewers for their work and have therefore added:

"Acknowledging the often substantial contributions of reviewers is a common courtesy."

**Online supplement**

Byerlee's law: Byerlee's law was originally derived for small-scale laboratory samples and is in itself quite amazing in that it shows that the maximum stress that rocks can withstand is nearly independent on the rock composition. What is even better is that it is nearly perfectly consistent with in-situ stress measurements in drill holes around the world (the classical reference would be Townend and Zoback, 2000). This shows that we can safely upscale Byerlee's law from small samples to the upper crust. For other geodynamically relevant parameters, such as the effective viscosity of rocks such upscaling does not appear to be that easy. It would be good to point that our here (or in

the main manuscript).

We added the following sentence to the glossary entry for Byerlee's law:

"Originally derived from small-scale laboratory samples, it has been shown that Byerlee's law can be safely upscaled to crustal conditions (Townend & Zoback 2000); something which is not straightforward for most geodynamically relevant parameters."

---

## Author Comment (AC4)

**Geodynamics 101 Revision - Review Laurent Montesi**

This is the authors' response to the review of Laurent Montesi. Our responses are indicated in **green**, with the original review presented in black.

This manuscript serves as a well-designed guide for modeling mantle and crustal-scale processes. As is necessarily the case with an exercise of the sort, it does in place reflect the personal experience and opinions of the authors, but it overall does a very nice job of remaining neutral, and even the more engaged sections are full of important information that will be useful for moving the field forward. I particularly appreciated the discussions of figure accessibility and Section 9 "Software, data, and resource management". There are also some very important discussions of the objectives of modeling, in particular the difference between "specific" and "generic" modeling. Both are presented as equally valuable, which is an important message to pass to both modeling and non-modeling communities.

Although the paper is already well organized and well written, I do have a few suggestions that could lead to significant rewriting, as detailed in the attached PDF. These include 1) make clearer that the focus of the paper is on geodynamic modeling of the mantle and the crust. 2) emphasize the importance of hypotheses in the model design process. 5) emphasize the role of constitutive relations 6) explain how brittle failure can be implemented. 7) better contrast FEM and FDM. 10) clarify what an "unphysical behavior" is. My biggest concern, though is with section 8.1 "Structure of a geodynamic modelling manuscript" that I find overly prescriptive. Other comments can be seen as minor.

Please note that most of my suggestions can be seen as a matter of personal preference and should not stand in the way of the publication of the paper (even 8.1!). The authors have produced an important manuscript that will many gain a better understanding of geodynamic modeling. That said, there is room in our discipline for personal preferences, and I welcome continued discussion of the topic. More of us should take an occasional pause from the pace of research to reflect on the higher objectives of our work, in this case geodynamic modeling. This paper is a great way to get the conversation started. Thank you for putting it together.

**101 Geodynamic Modelling: How to design, carry out, and interpret numerical studies**
*Review by Laurent Montesi, University of Maryland.*

This manuscript serves as a well-designed guide for modeling mantle and crustal-scale processes. As is necessarily the case with an exercise of the sort, it does in place reflect the personal experience and opinions of the authors, but it overall does a very nice job of remaining neutral, and even the more engaged sections are full of important information that will be useful for moving the field forward. I particularly appreciated the discussions of figure accessibility and Section 9. There are also some very important discussions of the objectives of modeling, in particular the difference between "specific" and "generic" modeling. Both are presented as equally valuable, which is an important message to pass to both modeling and non-modeling Communities.

Although the paper is currently well organized and well written, I do have a few suggestions that could lead to significant rewriting. They are organized here as topics and followed by several more isolated comments. Most suggestions can be seen as a matter of personal preference and should not stand in the way of the publication of the paper. Maybe they are best seen as a discussion of the material presented. I apologize that as my review is already much overdue, I did not study the appendix or glossary in detail.

**1. Although the title sets the scene for "Geodynamic modelling" in general,** the authors focus on the dynamics of the mantle and the crust. In place, they contrast their discussion with the core, especially in their neglect of inertia and Coriolis forces. There are other geodynamical settings where these are also neglected: the hydrosphere (especially the oceans) and the atmosphere, to say nothing of giant planets. The effects of fluids (including but not restricted to magma) are noted in passing, even though they are a growing segment of geodynamical studies. I do not think that presenting specific equations would be necessary to discuss modeling strategies and philosophies but as the authors have chosen to focus on a specific system, I believe that the title must reflect that choice. Maybe specify "Geodynamic Modelling of the Mantle and Crust" (lithosphere instead of crust would be OK too) and make this focus (including the neglect of two-phase flow) clear at the outset. The occasional mention of core studies can be seen as choosing a related science but neglecting other applications.

We agree that the paper focuses on the dynamics of the mantle and crust, although the same equations described here can be applied to the solid inner core of the Earth. To emphasise this focus, we immediately mention that we focus on the solid Earth in line 5 of the abstract and we mention that we use examples from mantle and lithosphere dynamics. To drive this point further home, we adjusted a sentence in the abstract:

"We provide ample examples, from lithosphere and mantle dynamics SPECIFICALLY, and point out synergies with related fields such as seismology, tectonophysics, geology, mineral physics, planetary science and geodesy."

Similarly, we highlight in the introduction that this paper focuses on mantle and lithosphere dynamics (l 67-69).  Hence, we think we have made it clear to our readers that this paper relates mostly to the part of the Earth between the atmosphere and the core. Moreover, we see our paper first and foremost as a guide to numerical models in general that heavily leans on specific examples that mainly involve modelling crust or mantle aspects. Extending the title as suggested appears therefore to make it too specific and might, for example, unnecessarily prevent non-modelling co-authors of core studies from reading and understanding the basics behind the general - indeed also to them very relevant - aspects of modelling. Therefore, we have not changed the title in this respect, although we have changed the title to better reflect the content of the paper by including 'communication' as a key word. We think this title emphasises the focus on numerical modelling (studies) in general and highlights the relevance of the paper for anyone interested in numerical modelling studies regardless of the specific application.

To highlight that other (geodynamic) disciplines use similar equations and assumptions, we added a line in the introduction citing atmospheric dynamics and hydrology as examples:

"Other disciplines, such as atmospheric dynamics and hydrology, use similar equations and assumptions, but are not discussed in detail here."

Minor points related to this topic:
• Line 20: The mention of "spherical shells" implies a global focus. Local and even regional studies do not need to consider shells.

We were not clear enough, which led to misunderstanding about the spherical shells (i.e., spheres, as in Atmosphere, Lithosphere, Asthenosphere). Every geodynamic subdiscipline is part of a sphere (or spherical shell), whether its focus is global or local only. We clarified the sentence to: "The broad definition of geodynamics typically results in a subdivision of disciplines relating to one of the Earth's spherical shells and specific spatial and temporal scales."

• Line 37: "magma dynamics and grain dynamics" The citation to Solomatov and Reese shows that you have global mantle convection in mind. This is a good paper but please also mention studies about magma dynamics, and also other fluids (CO2, water) that have been shown to matter in volcanic and seismogenic systems, and maybe also serpentinization (important in mid-ocean ridges).

We have added Keller et al 2013 in response to comments from Reviewer 1. To include studies about fluids and volatiles as well as serpentinization, we also included:
- McKenzie, D., 1984. The generation and compaction of partially molten rock, J. Petrol., 25,713–765
- Ruepke, L., Phipps Morgan, J., *et al.* (2004) 'Serpentine and the subduction zone water cycle', *Earth and Planetary Science Letters*, 223(1–2), pp. 17–34. doi: 10.1016/j.epsl.2004.04.018.
- Bercovici, David and Ricard, Yanick (2012), Mechanisms for the generation of plate tectonics by two-phase grain-damage and pinning. Physics of the Earth and Planetary Interiors, 202, 27--55.
- Katz, Richard F (2008), Magma dynamics with the enthalpy method: Benchmark solutions and magmatic focusing at mid-ocean ridges. Journal of Petrology, 12, 49, 2099-2121.
- Montesi, Laurent GJ and Behn, Mark D (2007), Mantle flow and melting underneath oblique and ultraslow mid-ocean ridges. Geophysical Research Letters, 24, 34.

• Line 80: I don't have the statistics to evaluate if applications to Earth's crust and mantle are indeed "the most common applications". I expect that people studying climate and atmosphere may have something to say about that. The concepts presented (except for the specific equations) are also applicable to their disciplines. It may be better to just say that these are the disciplines you worked on and therefore are most familiar with.

Fair point. We rewrote to:

"We will focus on the Earth's crust and mantle, although the methods outlined here are similarly valid for other solid planetary bodies such as terrestrial planets and icy satellites, and the Earth's solid inner core."

• I hate making titles longer (believe it or not) but it strikes me that the important communication steps that are in the paper are not included in the title.

A useful suggestion; we adjusted the title to: "101 Geodynamic modelling: How to design, interpret, and communicate numerical studies".

**2. Figure 2 is key to the entire paper.**
It is well-thought-out and I am sure the authors have spent much time discussing it. I do have several issues with it as it stands.

• It is easy to assume, based on Figure 2, that geodynamic modeling is a linear process. I believe this is an oversimplification. This paper would be a great place to emphasize instead the need to constantly reevaluate foundational hypotheses. Results and analysis, especially following evaluation against observations (necessary whether the model is "specific" or "generic") can lead to updates in any of the previous steps. "Nature" should feed directly into the validation (maybe) and analysis (certainly) steps.

The figure was designed so that "Nature" indeed appears to feed directly into all individual steps. We now better highlighted this aspect with more transparency of the individual panels. We also agree that Geodynamics should constantly reevaluate foundational hypotheses. While a graphical representation might be too complicated and counterproductive, we have made it clear now through the figure caption, with:

"Note that constant (re-)evaluation and potential subsequent adjustments of previous steps are key, and indeed necessary, throughout this process."

We also adjusted the text with:

"It is important to note that geodynamic modelling is not a linear process as depicted in Figure 2. For example, modelling does not necessarily start with the formulation of a hypothesis, as this can also arise from the encounter of interesting, unexpected modelling results, resulting in the formulation of a hypothesis later on in the process. Similarly, observations from nature feed directly into all of the above-mentioned steps as illustrated by the dark grey arrow through all modelling steps (Figure 2). Indeed, it is important to evaluate and adjust the numerical modelling study throughout the entire process"

• What is missing in Figure 2 and section 1.2 is the mention of a hypothesis. I strongly believe

that the most important models are those that were designed at the outset to answer a specific scientific hypothesis. As the paper is currently organized, it would be possible to write a code and verifying it before a hypothesis is identified (see line 127). To me, this is backward: we should design codes that enable us to answer a question, therefore the question needs to come first, not in the "setup" stage.

Good point! We added an additional panel to account for the hypothesis, which is indeed an important first step of geodynamic modelling.
We do, however, disagree to some extent with the comment about the most important models being based on a linear process from hypothesis to model design for the exact same reason as brought forward by the reviewer himself in just the previous comment. In fact, our schematic figure depicting numerical modelling study as a strictly linear process is inherently flawed as numerical studies are rarely linear and the hypothesis and ideas are adjusted and gained throughout the process. We added clarifying sentences on this in the figure caption and text (see comment above).

• This may be a matter of semantic, but it bothers me that "physical models" are described only using equations (also Line 87). I believe that physical models can be discussed in terms of concepts (e.g. buoyancy, inertia, diffusion, or the rheologies that are illustrated without equations in Figure 3) and that by the time you have introduced an equation, you have already moved on to the next level of abstraction: a mathematical model. That model can be solved analytically or numerically. Therefore I believe that an intermediate step ("mathematical model"?) is missing in Figure 2.

Yes, we had a long discussion on whether to use 'physical' or 'mathematical' model. In the end we agreed that all mathematical equations we use have a physical meaning. Besides that, the discretisation step of the equations can be considered a mathematical step as well (i.e., converting the physical model into a mathematical / numerical model), So, in order to avoid making things more confusing, we use 'physical' model consistently throughout to refer to the equations and 'numerical' model to refer to the discretised, numerical formulation of the equations.

• Somewhat related to the previous, don't forget that chemical processes and increasingly considered in geodynamics models. You might argue that they are included in "physical processes" as thermodynamics can be seen as either, but issues of trace element partitioning (important for evaluating the origin of magma or fluid interaction) are typically discussed only in geochemistry. Like physical models, chemical models need a mathematical description to be included in the later numerical modeling efforts.

We added '[... physical] or chemical process'.

• This is not an issue, but maybe consider distributing your figure as an open workflow (e.g. on https://workflowhub.eu/)

Thank you for bringing this website to our attention; we didn't know about it, but it is a helpful tool. Since it is specifically designed for storing specific, actual workflows, we unfortunately cannot register our schematic workflow figure here. However, we do refer to this website now in the paper to bring it to the attention of others (I 1593).

**3. Model simplification**

• You often use "simplified" to describe your model. This is perhaps a matter of preference but I favor "idealized". Complexity is not necessarily your enemy (and you do a good job discussing the pitfalls over oversimplification) but whenever you settle on mathematical relations, especially in the constitutive equations that are needed to close your balance equations, you are looking at an idealized but not necessarily simpler view of reality.

We agree that 'idealised' is also a good description of models in general, but we keep the terminology 'simplified' to indicate that all models are ultimately simplified (/idealised) versions of reality. Using this terminology then allows it to be used together with the terminology 'complex', highlighting that all models are simplified, but can have various levels of complexity, while never attaining reality. Related to that, we hope that we imply that we are solving a set of idealised equations (governing and constitutive; we indeed mention the term 'idealised' in this sense in the manuscript).

• Line 789: the concept of over-simplification is not shown in Figure. Another issue is that what is "too much" simplification depends really on the hypothesis to be tested (that's a reason why the hypothesis needs to be identified first). The "simple" convection model in Fig.7 is also quite complex to me. The triangle heart may be sufficiently realistic for some purpose. The link between oversimplification and hypothesis (line 791) should a major motivation for motivating the identification of the hypothesis before designing the model (see my comments about Figure 2) Here again, I would favor idealized over simple, and maybe realistic over complex.

We believe it depends on the problem at hand whether a model is oversimplified or not. For most people, triangles bear no direct relation to a heart (or love, for that matter) and hence are over-simplified in that the original goal is no longer reached. We did not find a way to improve on the figure regarding this aspect. We also think that the figure caption of figure 7 is already clear enough regarding the subjective meaning of "oversimplification" and we did not take any measures here.
Additionally, we find the current mentions of the link between simplification and hypothesis in the text already clear enough, which is why we did not take any measures here.

Given that the term "realistic" is harder to quantify and has an intrinsic positive meaning regarding modelling and could perhaps hint at 'the one true solution' (as it is realistic after all), we chose to stick with "complex" instead. Similarly, we keep "simplified" instead of "idealised" (also see our response to the comment above) and indeed regard these differences in formulation as a matter of personal preferences to a degree.

• Figure 8 could mention constitutive equations that coupled vary from non-linear to linear to

fixed parameters. The degree of coupling in Multiphysics problems could also be mentioned.

We agree with this addition and have added the following rows to the figure:

| Constitutive equations | non-linear | linearised | constant |
|---|---|---|---|
| Multiphysics1 | coupling | one-way coupling | none |

In caption: 1. Note that we mean 'multiphysics' beyond the already coupled system described in Section 2 (see Section 3.8).

**4. Line 247 etc.**

High thermal conductivity does not mean that the curvature $\nabla \cdot \nabla T$ is steep. Just consider a simple steady-state solution with fixed temperatures at both ends of a 1D solid. You know the temperature varies linearly between the two ends. Curvature is 0, regardless of the conductivity. What the high conductivity does is allow for large heat flow through that material. It also makes it faster (everything else being the same) to obtain this solution, so that in general, the curvature of the temperature field is less in the case with high k, at least in this specific example.

This is of course correct, thank you for pointing that out. What we meant was

"When the thermal conductivity k is larger, OR the temperature variation as expressed by the gradient nabla T becomes steeper, more heat is diffused."

We changed the manuscript accordingly.

**5. The description of physical concepts emphasizes the momentum equations.**

While I can appreciate their importance, most models also need to include constitutive equations. These appear in section 2.2 but should be at least mentioned earlier, as part of the model design. The fact that these evolutions, including their parameters, can change over time should also be described in general with an evolution equation, and that aspect is buried in "more complex processes" (section 2.3). It goes beyond the idea of fault gouge development (line 422). I don't think this is any more complex than the other kinds of equations described. Many constitutive parameters evolve following an ODE, so in a way, this is simpler.

We now mention that constitutive relations can be time-dependent in the section about constitutive relations (2.2), just before the start of the subsection about rheology (l 371-373). We also point out that this would require solving additional equations and refer to Section 2.3. In addition, we made the statement about fault gouges in line 507-508 more general.

We now also mention the constitutive equations already in the introduction of Section 2, just before the section on the mass conservation equation.

• Figure 3: As much as I like the emphasis on rheology, it strikes me that it is only one of the

many constitutive equations that can enter the model. Perhaps it is more fundamental because, without it, there would be no way to link stress and velocity. In that case, it has to make the distinction clear but highlight in the caption that other constitutive and evolution equations can (should?) be considered. Minor point: the diagram for "plastic" is brittle.

We now note in the figure caption that many geodynamic models include additional constitutive and evolution equations. We also point out in the figure caption that the plastic rheology depicted in the figure is the geodynamic approximation of brittle failure.

• Line 224: Stokes equations need not only eq. 2 and 5 but also viscous rheology. If you use elasticity you end up with a different set of equations.

We have added the line "Under the assumption that deformation is dominantly viscous, …"

• Line 229-231: Seismic waves do need inertia but they do not follow the viscous relations (Stokes or Navier-Stokes equations) that you describe here.

We have changed the sentence accordingly.

• In detail, Equation (10) doesn't work as it takes a tensor to a power that is not necessarily an integer. This equation should be written with a scalar measure of the strain rate and the stress tensor. We often use it in models with an effective viscosity approach. I think it would be important to explain this in the text. By the way, H2O should be styled as subscript (and not italicized) in $f$!!".

Indeed. We have changed the equation to use scalar measures and added this sentence after the equation:

"where $\dot{\varepsilon}$ and $\tau$ are a scalar strain rate and shear stress respectively. In geodynamic modelling they are usually the square root of the second invariants of the respective tensors."

We have made H2O a subscript.

• Line 374: In fact, grain size dependence is fairly well constrained. Grain size evolution is not (but you do not consider evolution at this stage). Also, dependence on porosity (? $*$? ) less well constrained.
This is a good point. We rephrased the sentence, which now says that grain size is not well-constrained in the mantle. We have also added a sentence about melt.

• I find the discussion of elastic deformation to be a distraction: it cannot be included as is with the other relations and the framework of Stokes flow. We can see that in line 471, it is written that "velocity … enter the stress tensor". That is incompatible with elasticity. I recommended either skipping section 2.2.3 or describing implementations of viscoelastic materials.

We now briefly describe how elasticity is included in codes that solve for Stokes flow:

"Elastic deformation is often included in geodynamic codes that solve for Stokes equations by taking the time derivative of Eq. [..], which introduces the velocity and strain-rate. The term sigma is then approximated by a first order Taylor expansion (see Appendix A) which ultimately amounts to adding terms to the right hand side of the momentum equation (Eq. [...]; see for example [...])."

• Line 480: Why the specific focus on diffusive quantities? You can have ODEs, e.g. grain size evolution.

We did list in the following paragraph more parameters that require more pdes/odes/evolution like grain size. We do not go into further details what kind of equations these processes are. In addition, the transport equation (15) can be used for grain size evolution, as long as additional terms for grain growth and reduction are added, as pointed out at the end of the paragraph.

• Line 505: the addition of Eq. 15 is inconsistent with the statement of introducing "the general concepts of geodynamics". I would argue that constitutive relations are more fundamental and General.

In our opinion, the advection equation 15 is the most common additional equation in geodynamic models. It is sometimes solved using particle methods instead of field methods, but being able to track the rock type or chemical composition is fundamental to many geodynamic applications. Sometimes it also includes additional terms (for reactions, or grain size evolution or other processes), and this is also something we point out in the text. But we agree that constitutive equations are important too, and have changed the text to "the partial differential equations (2), (5), (6), and potentially (15), as well as the constitutive equations"

**6. Section 2.2.4 needs more discussion of the way brittle failure can be implemented.**
• Line 392: brittle deformation takes place when rocks break, but this is not necessarily (in fact rarely) a rupture of the crystal lattice.

We removed "the crystal lattice breaks, and"

• Line 394: Actually, it is possible to represent discrete faults using split nodes (Melosh and Raefsky, 1981, A simple and efficient method for introducing faults into finite element computations, Bulletin of the Seismological Society of America 71 (5): 1391–1400), slippery nodes (Melosh and Williams, 1989, DOI 10.1029/JB094iB10p13961) and augmented FEM or X-FEM approaches (Ortiz et al., 1987, DOI 10.1016/0045-7825(87)90004-1; Dolbow et al., 2001, DOI 10.1016/S0045-7825(01)00260-2)

Since we do not want to introduce numerical terminology at this point of the manuscript, we have instead nuanced the formulation here to:

"However, in a continuum formulation, discontinuous faults cannot be represented and hence the deformation in geodynamics models typically localises in so-called shear bands of finite width, which can be interpreted as faults on crustal and lithospheric scales [...]. "

The short description of FEM in the paper unfortunately does not allow us to add the proposed references, as we think that explaining split nodes would be beyond the scope of this paper.

• Line 398: there is a difficulty, here again, of comparing and tensor (stress) and a scalar (yield stress). This difficulty should be mentioned. It is particularly important as the difference between criteria that are later mentioned (e.g. Tresca vs. Von Mises) stands in the way that this comparison is made (stress invariant for Tresca and Drucker-Prager, which leads to a smooth yield envelope, vs. resolved shear stress for Tresca and Mohr-Coulomb, which leads to a segmented yield envelope).

We have added a sentence that conveys the point of the reviewer:

"Note that it is difficult to compare a tensor (the stress) and a scalar (the yield stress). This is particularly important as the difference between the standard yield criteria is partly based on the way that this comparison is made, i.e., the stress invariant for von Mises and Drucker-Prager, which leads to a smooth yield envelope, versus resolved shear stress for Tresca and Mohr-Coulomb, which leads to a segmented yield envelope"

• Line 410: Byerlee's law has two branches. You describe here the friction coefficient of the high-pressure branch but that branch also has a cohesion term. The intermediate branch (important in the crust) does not have cohesion but its friction coefficient is 0.85.

We now explain Byerlee's law in more detail and describe both components (l 489-491).

• Line 415: Drucker-Prager and Mohr-Coulomb cannot be equivalent. One is smooth and depends only on the first and second stress invariants, the second is segmented. For an elasticplastic deformation, localization arises spontaneously at the corners of the yield envelope, so the difference is significant.

In general the two are not equivalent. However, for the specific case outlined in the manuscript (i.e. incompressible, 2D, plane strain) they are the same. We think we have described this with sufficient nuance.

**7. Section 3 misses some of the fundamental differences between FEM and FDM.**
As the authors are well aware, FEM uses the original equations (in their weak form) while FDM modifies these equations. Conversely, FEM restricts the solution to a predefined space. Therefore, the statement in line 510 that we are considering approximate solutions to Eq. 2, 5, and 6 is only strictly valid for FEM (FDM approximates the equations, not necessarily the solution).

We modified the comparison between FEM/FDM also in response to reviewer #2 comments:

"This also controls how the partial differential equations are solved on the grid, with finite difference methods solving the equations pointwise and finite element methods averaging the equations per element."

• Line 520: there are even approaches where one equation (energy?) is solved with FDM and the others by FEM.

Yes, we hope we explain that different methods can be used in different places and different combinations:

 "Often combinations of these methods are used to deal with time and space discretisation separately. [...] None is intrinsically better than the other, although there are differences that make a certain method more suitable for certain types of science questions. "

• Line 560 etc. The notation of $Q$1xP0 etc. the concept of "bi/trilinear velocity" must all be described here or in a glossary. I don't think it's reasonable for non-initiated readers to know what this means.

We have added a clearer definition of what these spaces mean:

"For quadrilaterals/hexahedra, the designation Q_m x Q_n means that each component of the velocity is approximated by a continuous piecewise polynomial of degree m in each direction on the element and likewise for pressure, except that the polynomial is of degree n. Again for the same families, Q_m x P_-n indicates the same velocity approximation with a pressure approximation that is a discontinuous complete piecewise polynomial of degree n (not of degree n in each direction ) [...]. Stable elements are typically characterised by m>n."

• Line 576: ALE is only one of several semi-Lagrangian methods.

We reformulated to:

"Finally, as its name implies, the Arbitrary Lagrangian-Eulerian (ALE), part of the semi-Lagrangian class of methods, is a kinematical description that combines features of both the Lagrangian and Eulerian formulations."

**8. Modeling philosophy**
• The analogy between car/model vs. engine/code did not work for me. To me, the code is more general than the model. The model takes the code and restricts it if using the setup. The code may have additional capabilities that are not used in a particular model. The car also uses many other components that are not the engine (steering wheel, headlamp, seats, doors). I am sure that every modeler can come up with their preferred analogy (and you may argue that your

analogy best fits your vision) but I might suggest that the code is like a road network and the model is like an itinerary that uses these roads to read a destination.

We agree that the car analogy was not fully waterproof and thank the reviewer for his constructive suggestion. We replaced the relevant paragraph with an analogy-free alternative that, we think, follows on the reviewer's suggestion.

• It would be useful in Section 5.2 to emphasize the importance of "failed" models. It is often impossible to conclude that a certain phenomenon is responsible for observations but it can be shown which situation does not explain observations (this is valid for both specific and generic models). Similarly, these failed models would not verify the criterion in Line 1085 of being "consistent with our understanding of geodynamic processes" but are important to probe the limits of our understanding (this may not be relevant for Phanerozoic Earth, but what about other circumstances?)

We agree that so-called "failed" models that do not fulfil a hypothesis are valuable too and have added a paragraph emphasizing this aspect to Section 5.2 as suggested:

"Both overarching modelling philosophies can either fulfil or reject a hypothesis. Most results published to date fulfil a hypothesis, even though positive modelling results only hint at a certain phenomenon being responsible for an observation. Modelling results that reject a hypothesis (often called `failed models') are of course more abundant, but, also, much clearer as they indeed serve as proof that a certain situation does not lead to a specific observation."

• Line 847: add "the PROPOSED control mechanism(s)". I am not convinced that assumptions and simplification are necessary for ALL control mechanisms. Some models may even be designed to find the necessary level of simplification.

We have added the word 'proposed'.

• Line 851: Other examples of regime diagrams can be found in Citron et al. (2020, DOI 10.1029/2019GC008895) and Gülcher et al. (2020, DOI https://doi.org/10.1038/s41561-020-0606-1)

We have added both suggestions as well as two other recent papers on more lithosphere/upper mantle scale.

• Line 856. Spiegelman focuses more on magma transport than on magma dynamics (often linked to eruptions). It would be good to also cite the newer work by Sim et al. (2020, https://doi.org/10.1016/j.pepi.2020.106486).

We added McKenzie 1984 and Sim et al. 2020. Additionally, we think the work in Spiegelman 1993 does add further understanding into magma dynamics.

**9. Boundary and Initial conditions**

• Line 898: prescribed stress can also be used to mimic topographic loads.

We added this.

• Line 910, 924: periodic BC are also commonplace in mantle convection dynamics, not just lithosphere dynamics (for example Gurnis, 1988; 10.1038/332695a0; Lowman et al., 2001 10.1046/j.1365-246X.2001.00471.x).

This was also pointed out by other reviewers. We added this fact and the references above.

• Line 925: why restricting the criticism to specific model setups? Generic models of long-term mantle convection are more realistic if the core temperature is allowed to vary according to its thermal balance.

We did not mean to restrict this comment to specific model setups and have rephrased the sentence as follows:

"However, this might not always be applicable."

• Line 926: I think you need a citation to support your statement that models of the outer core are better off prescribing heat flux boundary conditions.

We added a reference to the review paper of Wicht & Sanchez (2019) and changed the wording to "more appropriate".

• Line 955: Maybe highlight the issue of initial strain rate for models with non-linear viscosity (there is no solution if velocity is 0 everywhere).

We have added ", and is critical in cases where all deformation mechanisms included in a model are strain-rate-dependent (such as pure dislocation creep)."

• Line 965-6: Why specify the mantle when mentioning chemical heterogeneities. The crust is also highly heterogeneous.

We added the crust.

• Line 979 paragraph. Note that weak seeds can also be implemented as a random field of initial values, which is less constraining than a single seed (e.g. Jammes and Lavier, 2016, 10.1002/2016GC006399).

We have added this reference and the random field approach in the paragraph.

**10. Unphysical behavior vs. numerical problems (e.g. Line 1005 paragraph).**

• I am not certain what "nonphysical artefacts" you have in mind. An incorrect IC or BC is not sufficient to have nonphysical behavior but the resulting result could be irrelevant for the geological problem at hand (still being physical). Maybe you mean that the IC or BC is incorrectly implemented (the verification step should have taken care of that). In some cases, there could be issues of convergence, but not necessarily. I would imagine that the issues you identify (resolution, drunken sailor effect) would be identified at the verification stage (are you solving the problem correctly) not at the validation stage (are you solving the problem you think you are solving)

We see the code verification and model validation steps as follows: During code verification, correct implementation of IC and BC is checked (code development stage). During the first step of model validation, we check whether the particular chosen combination of IC and BC (i.e. the model setup) is solved correctly. For example, the way prescribed velocity boundaries are implemented by the developer can be correct, but if the net sum of the prescribed velocities over the boundaries as set by the user is non-zero for an incompressible model without a free surface, this will still lead to solver issues and model failure. The second component of model validation concerns the internal consistency of the chosen setup. In the last step of model validation, we check whether the particular model setup is representative of and relevant for the problem/hypothesis under investigation.

We have rephrased the first sentences of section 6.1 Common numerical problems without 'nonphysical artefacts':

"The construction of a specific model setup to investigate a particular problem or hypothesis can give rise to numerical issues, despite successful code verification. During the model validation process, these issues are identified and addressed."

• Lines 1035-1038. Stabilization by diffusion is helpful, but one should also remember that it potentially changes the set of physics included in the model (you are changing the equations). Same thing for mass scaling in FLAC codes.

We have added the sentence: "However, adding diffusion could possibly change the physics included in the model."

• Lines 1045: smoothing is indeed important for $Q$1xP0 elements but I think the recommended solution should be to use stable elements instead. I always refer to the list in the Bathe (2014). Finite Element Procedures textbook (Table 4.8 in the second edition). Certainly, there are other references)

Agreed. Also in light of the comments of the other reviewers, we have added:

"Stable elements, which fulfil the Ladyzhenskaya-Babuska-Brezzi compatibility condition (LBB or inf-sup condition), do not exhibit pressure artefacts (Donea and Huerta, 2003) and are preferable (see Bathe (2014) for examples of such elements). Moreover, the required number of

outer iterations does not increase significantly with mesh resolution compared to the Q1×P0 element (Thieulot and Bangerth, 2021)."

**11. Modelling manuscript.**

I found section 8.1 to be overly prescriptive. It presents one possible manuscript organization, which is indeed very common and effective. However, this is far from the only solution (this manuscript does not follow the prescribed structure, for example). I also notice that the general papers about manuscript structure and methods come from the clinical literature, which could have different conventions than geodynamics. At a minimum, the wording must be toned down (should or may instead of must). I would go so far as recommending you delete Figure 11 and much of section 8.1. I always prefer flexibility over rigidity when planning a publication. Scientific manuscripts DO NOT have a rigidly defined structure (Line 1316) although I can recognize that a structure can be useful especially for the first papers one might write. More specific comments follow.

We agree that there are different ways to report scientific results, but the section and the IMRAD structure of a manuscript are intended to be a starting point for students and scientists new to geodynamics. In our experience, having a structure is very helpful when writing the first papers. Equally, for non-geodynamicists, it is important to highlight the importance of the methods section, which determines whether the results are relevant for the investigation. Moreover, writing clear and comprehensive manuscripts is a critical skill in a scientist's training, which usually is not formally taught. We hope to provide some help here.

We do not think we are overly prescriptive, and repeatedly mention that journal guidelines may specify differently. We also found 3 instances of 'must' in the section, which we now changed to 'should'.

• Modern publications tend to deemphasize the method (especially in general journals) and emphasize results and insight instead. In my opinion, it is a good thing and helps broaden the reach of our papers. The IMRAD structure that you describe is appropriate for highly specialized publications but not for generic ones. Non-specialist will gloss over the technical aspects and focus on the take-home message of the paper. I would certainly emphasize more the message (you have just one sentence on telling a story in line 1317, yet that is what controls the impact of the paper).

We are not aware of journals that do not require a methods section, even if it is added as a supplementary section or material. We consider the methods section an important part of a manuscript, which allows verification and validation of the experimental setup and results. In numerical geodynamics, the "recipe" dictates the results.

We have added a sentence that advises to focus on a key message for both specialist and non-specialist readership. We now also mention that different journals place different emphasis on the IMRAD components:

"In geodynamics, the general structure of a manuscript follows the IMRAD structure (Figure 11), although journals can place different emphasis on the individual components through reordering and formatting."

• As the editor of a journal, I strongly disagree that "it is always good practice to write a complete methods section for every manuscript" (line 1271). If the focus of the paper is on the method, that's true. Otherwise, it is better to refer to other papers that have developed the method and maybe summarize that method in supporting information documents. This separation can strengthen the take-home message of the paper (and remember we study geodynamics to gain new insight). Also, it avoids having identical (or nearly-identical) method sections in different papers, which violates dual-publication policies. Finally, modern open science strategies require sharing codes in FAIR-enabling platforms (as you describe very well), in which case it should be possible to include a citation to the code and its version. It is to be expected that code publications will have documented verification and maybe validation steps. The paper can then focus on the hypothesis, setup, and results.

As reviewers of papers and scientists trying to replicate published results, we disagree with this comment, because we have encountered papers where the authors did not fully describe the methods section, the model was irreproducible, and it turned out the authors didn't understand the implementation of different features, which invalidated the whole experimental setup and results.

Even if, for example, a scientist uses a community code/method already developed, they might want to vary the parameter space or use some processes in combination with others that produce irrelevant results. A clear methods section describing the specific model is therefore still important (even if it is just included as supplementary material).

To nuance our formulation, we have removed the word 'always'.

• On a related topic, line 1231 "The methods section is considered one of the most important parts of any scientific manuscript" is more true of social and clinical sciences (the Kallet reference you include), which struggle with reproducibility due to reliance on human subjects and survey methods, than it is in geodynamics modeling. Note also that "who performed the experiment" (line 1241) should be irrelevant: the models should be reproducible by anyone. I think again that the context of the cited paper (here Annesley, 2010) makes it irrelevant for our discipline.

We note the lack of publications on this topic in geodynamics, so we had to borrow from other, larger fields (i.e. medical sciences), and adapt to our field from our experience (and we do mention in the text that imrad and content may vary from field to field). We hope that with this manuscript we can provide some best practices for our field.

We agree the explanation on the different questions relating to the methods was unclear. We have now reformulated it, such that each question has a clear geodynamic example. For

example, for geodynamic modeling 'who performed the experiment' translates to on what cluster and with what hardware the simulation was performed. When the same model setup is run on a different system with a different installation, dependencies and number of cores, the results of the (same) model could be different. We added geodynamic-specific explanations for the relevant questions in lines 1394-1400.

• My other problem with IMRAD as described is that it links discussion and conclusions. In our disciplines, these sections have different purposes: one puts the results in a broader perspective, and may even speculate on future hypotheses, while the other summarizes the paper.

We agree that the discussion and conclusions serve different purposes (although some journals nowadays merge the discussion and conclusion) and we discuss them as such in the current document.

• Line 1239-1240 I don't know what journal specifies "how many words can be used" to describe methods, but certainly not the one I edit.

There are a few instances where the number of words for the methods is specified. For example, Nature Communications write in their guidelines: "Methods should be written as concisely as possible and typically do not exceed 3,000 words but may be longer if necessary." However, it is more common that there is a general word limit, which then implicitly limits the amount of words available for the methods. For example, from Geology: "Geology manuscripts must fit the following size constraints: (1) A character count with spaces (in Microsoft Word) of no more than 18,500 characters for the title, author names, affiliations, abstract, main text, Acknowledgments, and figure captions (the References Cited list does not need to be counted)." We do agree with the reviewer that the word limit specification for methods is not common, so we removed this.

• Line 1288: The results section should describe the model results. Answering the central question or hypothesis of the paper should only happen after these results are analyzed and evaluated, which is best done as a discussion. This is in fact what you prescribe in Line 1297.

We changed the sentence to: The main goal of the results section is to present quantitative arguments to the initial hypothesis.

• A couple of pitfalls should be mentioned in the Line 1306 paragraph: the abstract should give a preview of the results as well as the work. Too often I see abstracts that say what is done without saying what is learned (which is what will inspire a broader audience to read the paper). Second: the best titles remain succinct (I was told to limit titles to ten major words). I know I suggested additions to your title but in my opinion, the title should also stop at the column. Two-part titles are often cumbersome. You may also want to mention the use of Plain Language Summary and Graphical Abstracts, for publishers that allow them, as alternative ways to

engage a broad readership (note that the purpose and mechanics of Plain Language Summaries are not the same as those of abstracts).

We have added these suggestions.

**12. Not surprisingly considering who the authors are, the section on visualization is very strong.**
I have a couple of points to make, though
• Line 1323: Bar plot should have a well-justified baseline, but it does not have to be 0. Imagine that the quantity reported varied over several orders of magnitude. I may best to use a log scale, and 0 may not be the best reference (e.g. grain size varying from micron to millimeters).

A logarithmic bar plot is indeed a rare case, where the base level must not be 0 (but 1). We added the statement: "(or in the logarithmic case, have a baseline at one)", to clarify this.

• Line 1341: there are circumstances when it is necessary to change the range of a color scale. I agree it typically should be avoided and if the range changes, that needs to be emphasized in the caption. However, saying the range should "always" be the same is overly prescriptive.

This is a good point and we agree. We rewrote accordingly.

• Perceptually uniform color maps are certainly to be preferred for an unbiased reporting of results. However, figures should also inform the readership, and it may be useful to take advantage of non-uniformity to highlight a result (10.1109/TVCG.2018.2855742). This is in a sense what is done with a multi-sequential color map (oleron, highlighting the sea level) and there are also tools for interactively creating colormaps (https://sciviscolor.org/colormoves/overview/)

We disagree with this statement and the effort represented by the people behind *sciviscolor.org* (see also the related discussion in Crameri et al., 2020, *The misuse of colour in science communication*).
Axes (colorbars included) are not a tool to highlight or interpret data. Likewise, it is ill-practice to suggest squeezing any y-axes between certain tick values to highlight some parts of the data to inform the readership, while it is factually a distortion and misleading the readers. Instead, axes (colorbars included) are a tool to represent data, so that others (not only during peer-review) can actually judge the interpretation(s) made by the authors. Community feedback and checks are fundamental pillars of science. If others cannot judge the interpretation of a study, then the study presented becomes just a fact-less opinion piece by the authors. With distorted figure axes, there is no way to judge the interpretation made based on the underlying data without the redoing of the entire study - because it can't be seen. To highlight data, an entire variety of alternative options exist, like superposed contour lines or graphical indicators like arrows. Perceptual uniformity, however, is a must for scientists; no way around it – and really no need for a way around it either.
We clarified that further in the manuscript.

Multi-sequential, perceptually uniform colour maps like oleron do not distort the data, because they basically represent two datasets (for surface elevation this would be ocean bathymetry and land topography), just using one single colour axis. The same also applies to diverging colour maps.

We do not mention tools for interactively creating colour maps, since hardly any scientist today is equipped to personally create a colour map that fulfills critical aspects like perceptual uniformity or colour-vision deficiency friendliness.

**13. Section 9 is also very strong and useful. It contains many important resources.**
• I wonder if the underlined words should be links, though. In that case, they would best included as URL citations (or better permanent citations if a DOI is available).

Indeed the underlined words are links. As Solid Earth informed us, in the Discussion stage links are unfortunately disabled, but not in the final publication. Nevertheless, we have reviewed all link occurrences, and where applicable replaced them with a permanent citation.

• You may also want to point to CIG's best practice documents at https://geodynamics.org/cig/dev/best-practices/.

We have included the CIG's guidelines as follows, with a link to the given webpage:

"Certain organisations that provide a platform for software packages state their own  guidelines and requirements, for example those of the Computational Infrastructure for Geodynamics (CIG)."

• Line 1418. Another argument for storing and sharing numerical data is to enable the R of FAIR (Reusability). Other scientists may be interested in new analyses of the model runs for different purposes or quantitative comparison with other studies. Sharing results saves on the time and computational cost needed to reproduce the results.

Agreed. We think the sentence "However, accessible model results can save the computational resources needed to recreate the model results, " (l 1590-1592) already covers this aspect of FAIR.

• Line 1430: Mention the Planetary Data System?

We have added PDS, thank you for the suggestion.

• Line 1442: Mention Earth and Space Science journal (https://agupubs.onlinelibrary.wiley.com/journal/23335084)?

We have added the journal.

• Line 1443: here I think you are not strong enough: research not only can but must create and use persistent identifiers whenever possible (and you do include great pointers on how to do this).

We have made the statement stronger by writing "Not only data can (and should) have a persistent identifier".

• Line 1446: note that ORCID iD is required by some journals.

Indeed. We have updated the sentence to "The article processing portals of many journals already allow or require researchers to link their ORCID iD to their profile, ...".

**14. Isolated points**
• Line 24: replace hundreds by thousands of km (the scale of the largest plates, or the "penetration" of plate boundary deformation in Asia or North America).

We modified this.

• Line 35: "THEY take place"? (subject surface processes)

Fixed

• Figure 1: 1)make the axes labels darker? 2) what about aseismic transients and creep Processes?

We have made the numeric tick labels a shade darker and improved the overall readability of the figure. In order to avoid cluttering the figure, we chose to stay with the overarching processes already shown. Aseismic transients and creep processes are rather complex terms that need not appear on such an early figure in the manuscript and would require additional substantial explanation we do not want to focus on.

• Line 55: While I am proud that you included my paper here, I would recommending mentioning the seminal papers of Hager and O'Connell (1981, https://doi.org/10.1029/JB086iB06p04843), McKenzie (1969 https://doi.org/10.1111/j.1365-246X.1969.tb00259.x). There has been action between 1879 and 2006…

We have added both references.

• Line 63: What do you mean by "the physical properties of the variables" (variables can have value and they can represent physical properties, but they don't have physical properties).

We clarified the sentence and we now write:

"This is due to the fact that numerical models are increasingly more difficult to solve computationally when large contrasts are present in the physical properties (e.g., in space or time)"

• Line 97: I would regard boundary conditions as a part of the model setup, not simulations.

We agree with this and have changed the sentence to reflect this:

"Each numerical model can have a different model setup with different dimensions, geometries, and initial and boundary conditions. For a specific model setup, the numerical model can then be executed multiple times by varying different aspects of the model setup to constitute multiple model simulations (also often called model runs)."

• Line 112 etc. It would be good to mention data assimilation.

This was included in the sentence already.

• Line 130: I would think that it's the model results, not the setup, that need to be compared to observations.

We agree and changed the sentence to clarify:

"After the model has successfully run, the model results should be validated in terms of robustness and against observations (Section 6). Once the model has been validated, the results of the simulation need to be interpreted, visualised, and diagnosed while taking into account the assumptions and limitations that entered the modelling study in previous steps."

• Line 133: add "openly" to "clearly and reproducibly" (BTW: reproducibly, not reproducible)

We changed this.

• Line 136: elasticity can be important at long time scales too. See plate bending. Its role in longterm tectonics is still debated.

We adjusted the sentence to make it sound less like there is just one single deformation mechanism at play for a given condition. In addition, we added a more in-depth discussion about how well geodynamic models approximate the real rheology in Section 2.2.1.

• Line 141: 450 years assumes a certain viscosity. I am quite certain it is longer when you enter the lithosphere (still in the mantle). Do you assume the asthenospheric mantle?

We took this value from the reference we give (Schubert, Turcotte and Olson; Mantle Convection in the Earth and Planets) which assume a viscosity of $10^{21}$. But this of course excludes the lithosphere, so we changed our wording to "sublithospheric mantle". The viscosity

is of course still a bit low for the lower mantle, but on the other hand the viscosity may go down to even smaller values in the asthenosphere, so we take this value as a compromise (and after all, we point out that this is just an order-of-magnitude estimate).

• Line 162: I would focus on the physical processes that are of interest (not just relevant).

We changed this to:

"Depending on which physical processes are relevant and/or of interest,"

• Line 169 misses a space in "viscousFluid".

We corrected this.

• Line 255: Definitions may be different for different people, but I believe that shear heating is the more general term (it does not imply a shear mechanism) and that it can be divided into frictional and viscous dissipation. This is opposite to the relation you include here.

We follow Ismael-Zadeh and Tackley (2010, section 10.3.4), and call the term viscous dissipation, which is sometimes also referred to as shear heating. We changed the sentence to:

"Viscous dissipation, also called shear heating, describes the amount of energy that is released as heat when material is deformed viscously and/or plastically."

• Line 270: Melting is much more complex than shown here, as the heat can be transported by melt flow. You also have the issue that the fraction that has melted (X) may be different for the (retained) melt fraction.

We agree that including melt generation and transport in a model is more complex than just including the latent heat term for melting, and we hope that this becomes clear in other parts of the manuscript (for example, in the section on more complex processes and in the rheology section where we discuss the dependence of viscosity on melt fraction). In line 270 we only discuss the latent heat terms (which is also all that is needed if one wants to include **melting**), and therefore we think it is not necessary to include additional details on what other terms would be needed to model **melt transport**.

• Line 283: density OF the …profile.

We fixed this.

• Line 297-309: It may be worth mentioning that many studies have used the EBA but called it Boussinesq.

To avoid confusion, we have refrained from adding this point.

• Line 458: why do you assume the relation between density and temperature is linear?

The sentence is actually more nuanced than that as we specifically start it with "In the simplest case" and mention that one of the options then *may be* a linear relationship.

• Line 584: typo: 10&'.

We fixed this.

I think the shear forces should be negligible, but it's not true that there are no shear forces.

We have changed the text to say the forces are negligible.

• Figure 4 gives the impression that triangular meshes are necessary for conforming to an interface. You can have quad meshes that do the same. Maybe include one on the bottom right panel? Anyway, as currently designed, the panel with the two fluids appears in the "mesh" row, so that's a little confusing. Define the fluids in the Field Method panel. I also find the grey fonts too light (I honestly didn't see the words "mesh" and "method" when I first looked at the figure).

We added in the caption that quadrilateral meshes can also conform to interfaces. We did not see a way to add it to the figure itself without introducing more complexity. Also, triangles are more commonly used to conform to interfaces than quads.

We moved the definition of the fluids (now called Materials) to the Field Method panel, and made the gray words a shade darker.

• Line 640: the viscosity "MAY DEPEND" on the velocity… (it's not a requirement)

We changed this.

• Line 719: Explain what you mean by an "as simple hello world" test.

This is explained in the glossary as we didn't want to clutter the text.

• Line 725: I know it's impossible to be exhaustive here, but I would like to see mentioned example of corner flow, viscous folding, and half-space cooling. The latter is particularly important as otherwise there would be no example using the energy equation. Folding also presents interesting numerical effects (see Schmalholz and Podlachikov, 1999, DOI 10.1029/1999GL900412)

We have added the corner flow, viscous folding, and half-space cooling to the list of examples, as well as the proposed reference.

• Line 744: missing words "It is important…"

We changed this.

• Line 759: Laboratory experiments do not have "infinite resolution" especially in tectonic applications with granular media. Also, even if all possible physics is indeed included, the constitutive relations are not always fully understood (Katz et al., 2005 Tectonic microplates in a wax model of sea-floor spreading, New J. Phys. 7 37, Di Giuseppe et al., 2012. https://doi.org/10.1007/s00397-011-0611-9). These limitations are important to mention as it highlights the complementary between numerical and analog models (in numerical studies, only selected sets of physics are included, but that means they are well controlled).

We agree and we have removed this part of the sentence. Differences between numerical and analogue models are briefly discussed in the introduction (also elaborated upon with respect to the previous version of the manuscript in response to the comment of Paul Pukite in the open discussion).

• I would have preferred to see the non-dimensional numbers of lines 1130-1 introduced with the concept of the regime diagram (Line 851).

We added a sentence on this:

"Non-dimensional numbers often make up axes and boundaries in regime diagrams, although other diagnostic quantities can also be used (Section 7.2)."

---

## Author Response (AR2)

Dear Editor, dear Reviewers,

Please find below our revised detailed responses in green to the minor revisions required by the reviewers. The modified manuscript can be find in the attachments. We also added a link to an online repository where our figures can be downloaded in various formats and in various styles. We would like to thank you all once again for your detailed constructive feedback, which greatly improved the manuscript.

Yours sincerely,

Iris van Zelst (corresponding author),
Fabio Crameri, Adina E. Pusok, Anne Glerum, Juliane Dannberg, Cedric Thieulot

**Boris Kaus**

Thanks, you did a great job with correcting the manuscript which I believe is significantly clarified now.

My only minor comment is on line 812, where I believe part of the sentence is missing. I guess it should be: … experimental numerical applications "and scale those" to large-parallelized production code (text between apostrophes added). I guess this can be added during typesetting.

Thanks for pointing that out! We rectified this.

**Paul Tackley**

I have no further revisions to recommend, except to point out that rock is not made of molecules - it has a crystalline structure:
l163 "we ignore that the material is made up of individual molecules"
Change "molecules" -> "atoms"

Good point. We changed this.

**Laurent Montesi**

Once again, it was a pleasure to read this important and well-organized manuscript. The authors have generally changed the manuscript where appropriate in response to my comments. In a few cases, they disagree with my suggestions and gave clear explanations why. As I wrote in my original review most comments express personal opinions or preferences, and so I am satisfied with the authors' response. I do have a couple of further requests for clarification and wish to return to some of our discussions.

- The title has been updated to mention communication, which is a great addition and emphasizes a particularly original part of the paper. However, the authors declined my suggestion to mention the mantle, crust, or lithosphere in the paper, as that distinction is made in the abstract. While that is true, I still feel that readers from different geodynamic subdisciplines may be frustrated by the emphasis on convection and could use a clarification of the context of the paper when they first meet it, i.e., when they read the title. Elements of discussion (e.g., writing, study organization) are universal, but a large part of the paper discusses specific equations. Someone working on magma dynamics (two-phase flow), groundwater flow (Darcy's law), or even earthquake mechanics (elastodynamics) would be frustrated with the appearance that geodynamics is assumed to be equivalent to mantle convection. You refer to Schubert et al. (2001) for the derivation of the equations, but please note that this book, although very important, is focused on mantle convection, not geodynamics in general. Your response to my concern of Eq. 15 is that it is the most common additional

equation in geodynamics models. This ignores every problem based on elasticity or fluid transport. It is your paper, and you have the right to frame it as you wish. It just is not as general as you might think.

We recognise the point you are making. In order to reflect this within the paper, we modified our title to "101 Geodynamic modelling: How to design, interpret, and communicate numerical studies **of the solid Earth**", which we hope sufficiently emphasises that we are predominantly concerned with the geodynamics of the solid Earth (i.e., lithosphere and mantle dynamics).

- In response to my comment to line 415, you say that the two criteria are equivalent for an "incompressible, 2D, plane strain model". That does not make sense to me. The failure criteria only involve stress. Incompressibility and plane strain assumptions involve strains, so unless you assume a rheology, there is no general link between the conditions you express and the failure criteria. Even in 2D, Mohr-Coulomb features corners, as it is based on the maximum differential stress, while Drucker-Prager is smooth, as it is based on invariants (as you added to section 2.2.4).

The two failure criteria are equivalent under these sets of circumstances. For a full derivation of the equivalence of these two failure criteria under these assumptions, we refer you to Section 3.26.13 in the Fieldstone code manual, which can be found here: http://cedricthieulot.net/manual.pdf

• The following should not lead to a change in the paper, but I would like to push you a little on the topic of non-uniform color scales. Note that I am a fan of your color scale and try to use them in all my recent work, so, fundamentally, I agree with you. However, the effect of using a color scale that highlights a certain value is not very different from one of the "alternatives" you mention, superposing a contour. The key is that the person using a non-perceptually uniform scale should do it with a purpose (most use those scales without giving it a thought, and that's bad). It is not easy to use these scales properly, so I agree that there is no need for going through the trouble, but I personally prefer to leave options open.

We appreciate the reviewer's thoughts and that the topic is being discussed. The difference between using an even colour map (i.e., a correct scale) with a superposed contour (i.e., a superposed interpretation) and solely using an uneven colour map (i.e., a faulty scale) is that in the latter approach, the data itself is not represented, but only the author's interpretation. This makes an independent evaluation of the author's interpretation/conclusion impossible and the scientific procedure fails.
If the author decides to interpret the data with a distorted scale (according to the reviewer's suggestion), then that is fine if – and only if – the data is additionally separately presented with an accurate scale.

Minor comments on the updated text:
- Line 54: there have been approaches for gravity scaling using centrifuges (e.g., Ramberg, H., (1967) Model Experimentation of the Effect of Gravity on Tectonic Processes, Geophysical Journal International, 14, 307–329, doi:10.1111/j.1365-246X.1967.tb06247.x )

This is a good point. We changed the formulation of our sentence by mentioning that it is difficult (and not impossible) to scale gravity in a lab setting and we added several references (Ramberg et al., 1967; Mart et al., 2005; Noble and Dixon, 2011).

- Figure 3: I just noticed that the drawing for a plastic material is actually brittle. We use a plastic approximation for frictional sliding, where, unlike that snapping bar, the materials stay in contact after failure.

We modified the figure by writing 'brittle' instead of 'plastic' and we added a sentence to the caption to clarify that brittle rheology in geodynamics is approximated by plasticity, as depicted by the dark grey plastic slider in the figure. We also refer the reader to the relevant section in the caption. We hope this sufficiently clarifies this.

- Lines 208-210: I am confused by the discussion of a "viscous isentropic relaxation time scale". If you discuss "viscous relaxation", how can "elastic forces" dominate the time scale? A quick look at Curbelo et al. (2019) paper shows that the time scale does involve viscosity but no elastic moduli. Please include the reference to back up these statements for a viscoelastic body and the time scales shown on line 210. By the way, it may be worth quickly mentioning that fluid transport can lead to additional volume changes (viscous compaction; see McKenzie 1984, Ricard et al. 2001, Wilson et al., 2014a).

The reviewer is right that this is formulated in a way that is misleading. We have changed the text to the following:

"Because the first term explicitly includes a time-dependence, it introduces a characteristic time scale into the model due to viscous (Curbelo et al., 2019) and elastic forces (Patocka et al., 2019). For viscous forces, this is called the viscous isentropic relaxation time scale and is on the order of a few hundred years for the upper mantle to a few tens of thousands of years for the lower mantle (Curbelo et al., 2019). For visco-elastic deformation, this time scale is the Maxwell time (see above). When we consider the Earth as a visco-elastic body (see Section 2.2.1), the relaxation time is dominated by elastic forces."

The viscous relaxation time scales with viscosity over pressure, so using Earth-like values of 1e20 Pa s (upper mantle viscosity) to 1e22 Pa s (lower mantle viscosity) and 10GPa (pressure), we get 1e10-1e12 s, or 300 to 30 000 years using equation 4.5 from Curbelo et al. (2019).

- Line 235: Without a rheology, the stress tensor does not contain information on the deformation of the material. That would make it a deformation vector.

We removed our explanation of the stress tensor containing information on deformation and instead only mention that sigma represents the stress tensor.

- Line 381: Very confusing sentence. "Rocks can deform elastically, by brittle failure" implies that "brittle failure" enables elastic deformation, and of course, it's not the case.

We rephrased this sentence.